# Manifold Approximation leads to Robust Kernel Alignment

### Abstract

Centered kernel alignment (CKA) is a popular metric for comparing representations, determining equivalence of networks, and neuroscience research. However, CKA does not account for the underlying manifold and relies on numerous heuristics that cause it to behave differently at different scales of data. In this work, we propose Manifold approximated Kernel Alignment (MKA), which incorporates manifold geometry into the alignment task. We derive a theoretical framework for MKA. We perform empirical evaluations on synthetic datasets and real-world examples to characterize and compare MKA to its contemporaries. Our findings suggest that manifold-aware kernel alignment provides a more robust foundation for measuring representations, with potential applications in representation learning.

## 1 Introduction

Centered Kernel Alignment (CKA) (Cortes et al., 2010; Kornblith et al., 2019) is a statistical method used to compare the similarity between representations of data, often in the form of feature maps or embeddings. It works by aligning kernels, which capture pairwise relationships within datasets, and measuring their agreement. CKA is widely used in studies to compare layers of neural networks, analyze representational similarity, and study how models process information (Ramasesh et al., 2021; Nguyen et al., 2022; Ciernik et al., 2024). Its ability to handle datasets of different sizes and dimensions makes it a powerful tool to understand complex models and evaluate their performance. However, very few studies have characterized CKA under known representations/topologies. Moreover, the reliability of the CKA measure has been under scrutiny numerous times (Davari et al., 2023; Murphy et al., 2024).

To address this, we propose Manifold-approximated Kernel Alignment (MKA). Manifold approximation is a way of understanding and simplifying complex data. In many real-world problems, data with many dimensions - like x-rays, medical records, and neuroimaging data - actually lie on a much smaller, curved structure called a "manifold" within the high-dimensional space. Known as the "manifold hypothesis", this concept is integral to modern statistics and learning algorithms (Fefferman et al., 2016). Manifold approximation uncovers and represents this underlying structure within the high-dimensional data by exploiting the relationships between data points.

We use manifold approximation to define a non-linear and non-Mercer kernel. Using this kernel function, we provide a theoretical framework for MKA. With extensive characterization on synthetic datasets, we show that MKA is more consistent under varying dimensionality and shapes that preserve topology. We also discovered that MKA captures the underlying topology better and is less sensitive to hyperparameters than CKA, its variants, and many contemporary methods. To achieve this, we performed experiments using various known shapes and topologies, taking into consideration distributions and their behavior that mimics real work settings. We also perform large-scale benchmarks on multiple tasks (vision, natural language, and graph) and datasets to assess the quality of the algorithm. Overall, this work will pave the way for applying manifold approximation in diverse applications.

## 2 Related Works

The recent interest in alignment metrics stems from the desire to understand how neural network works and how the intermediate layers of neural networks are related. To compare learned features, we need

metrics that measure alignment between two representations. Earlier studies assessed representational similarity with correlation- and mutual-information–based measures (Li et al., 2016) and with linear-classifier probes (Alain & Bengio, 2017). Next progress came from Raghu et al. (2017), who modeled the problem as one of dimensionality reduction and used singular value decomposition (SVD) to remove noise from the representations, followed by canonical correlation for alignment, namely SVCCA. Later, Morcos et al. (2018) proposed PWCCA, which extends SVCCA by weighting the canonical directions according to their contribution to the original representations, making the similarity measure more robust to noisy or unimportant dimensions. This dimensionality reduction approach is also followed by a few other studies (Sussillo & Barak, 2013; Maheswaranathan et al., 2019). Other approaches include, revisiting classifier probes (Graziani et al., 2019; Davari et al., 2022), exploring multiple approaches together Ding et al. (2021), Procrustes analysis (Williams et al., 2021), graphs (Chen et al., 2021), and exploring effect of transformations (Lenc & Vedaldi, 2015).

However, Kornblith et al. (2019)'s exploration of representations through kernel methods has sparked renewed attention and discoveries in this area. Known as centered kernel alignment (CKA) (Cortes et al., 2010; Kornblith et al., 2019), this approach compares two different kernel matrices obtained from the representations. The initial studies (Kornblith et al., 2019; Nguyen et al., 2021; Raghu et al., 2021) explored the feature similarity of nearby layers (the famous block structure). Another concurrent observation is that the kernel structure is relatively robust when low-variance components are removed Ding et al. (2021). Later, Nguyen et al. (2022) discovered that the block structure is primarily due to a few dominant data points. Davari et al. (2023) formalized these observations theoretically. For a comparison of CKA to its contemporaries (representation similarity analysis and CCA), see Williams (2024).

Another avenue is to explore the nearest neighbor structure, which, in our opinion, is a natural extension of the CKA philosophy. Huh et al. (2024) proposed a mutual nearest neighbor-based extension of CKA. Tsitsulin et al. (2020) proposed Intrinsic Multi-scale Distance (IMD), which uses the heat kernel to estimate the manifold. Recently, topological data analysis has been applied to propose Representational Topology Divergence (RTD) (Barannikov et al., 2022; Tulchinskii et al., 2025).

The kernel approach also connects to manifold approximation, a cornerstone of non-linear dimensionality reduction. Methods such as stochastic neighbor embedding (Hinton & Roweis, 2002; Van der Maaten & Hinton, 2008), Uniform Manifold Approximation and Projection (UMAP) (McInnes et al., 2018), related variants (Wang et al., 2021; Damrich et al., 2023; Islam & Fleischer, 2022), and legacy methods (Belkin & Niyogi, 2002; Coifman & Lafon, 2006) rely on efficient sampling of the manifold followed by optimization of a low-dimensional embedding. In particular, the use of k-nearest neighbor graphs and parameter-tuned local neighborhoods has proven to be an effective tool for this class of methods. While k-nearest neighbors show usefulness in some recently proposed alignment metrics (Tsitsulin et al., 2020; Huh et al., 2024) and topology (Damrich et al., 2024), the kernels arising from manifold approximation lack wide adoption here and in other kernel-based algorithms.

## 3 Centered Kernel Alignment (CKA)

Let $X \in \mathbb{R}^{n \times d_1}$ and $Y \in \mathbb{R}^{n \times d_2}$ be feature sets from $n$ samples each with $d_1$ and $d_2$ features, respectively. The corresponding symmetric kernel matrices are $K$ and $L$ with $K_{ij} = k(x_i, x_j)$ and $L_{ij} = l(y_i, y_j)$, respectively. The CKA measure between the two feature sets is given by

$$\text{CKA}(K, L) = \frac{\text{HSIC}(K, L)}{\sqrt{\text{HSIC}(K, K) \text{HSIC}(L, L)}}, \tag{1}$$

where $\text{HSIC}(\cdot, \cdot)$ is the Hilbert-Schmidt independent criterion given by $\text{HSIC}(K, L) = \frac{1}{(n-1)^2} \text{trace}(KHLH)$. Here, $H = I - \frac{1}{n}\mathbf{1}\mathbf{1}^T$ is a centering matrix that mitigates bias in the kernel. There are other debiasing techniques (Song et al., 2007; Sucholutsky et al., 2023), however, we will consider the simplest and most widely used one in practice. HSIC computes the similarity between the two kernel matrices of the same size, while the CKA measure normalizes this similarity within $[0, 1]$.

Various options exist for the kernel. The common ones include the linear kernel (LIN) given by $k(x_i, x_j) = x_i^T x_j$ and the radial basis function (RBF) kernel given by $k(x_i, x_j) = \exp(-||x_i - x_j||/(2\sigma^2))$, where $\sigma$ is the

bandwidth of the Gaussian. The following theorem establishes an equivalence relation between CKA with linear and RBF kernel:

**Theorem 3.1** (Alvarez (2022)). $\mathrm{CKA}(K_{\mathrm{RBF}}, L) = \mathrm{CKA}(K_{\mathrm{LIN}}, L) + O(1/\sigma^2)$ *as* $\sigma \to \infty$. *Here,* $K_{\mathrm{RBF}}$ *is the RBF kernel matrix with bandwidth* $\sigma$, $K_{\mathrm{LIN}}$ *is the linear kernel matrix, and* $L$ *is any positive definite symmetric kernel matrix.*

Softly, it states that at higher values of $\sigma$, CKA with linear and RBF kernels behave equivalently. Various studies have reported this in empirical settings (e.g., in Kornblith et al. (2019) and Fig. 4(a) of Davari et al. (2023)). Thus, most researchers use the linear kernel, effectively capturing linear relationships alone. And by Theorem 3.1, even results with an RBF kernel (without properly tuning the bandwidth, $\sigma$) potentially suffer from the same pitfalls of the linear one. In the rest of the paper, unless explicitly mentioned, we use CKA only with RBF kernel.

## 4 Manifold-approximated Kernel Alignment (MKA)

Manifold approximation is a method for defining a graph that quantifies the pairwise relations within the data. A manifold-approximated kernel is often sparse and typically obtained by the k-nearest neighbor (kNN) algorithm. These kernels are generally not positive semidefinite; rather are of non-mercer (Ong et al., 2004) type. Moreover, we will use a kernel function that is non-symmetric (i.e., $k(x_i, x_j) \neq k(x_j, x_i)$). We impose additional conditions for computational advantage, explainability and robustness. Overall, for MKA the design properties are:

A1 (**directed** kNN **locality; non-symmetric**) $K_{ii} = 1$ and $K_{ij} = 0$ unless $x_j \in \mathrm{kNN}(x_i)$.

A2 (**constant row-sum**) $K\mathbf{1} = D\mathbf{1}$, where $D$ is a constant.

A3 (**stability**) There exists a parameter sequence $(\theta_k)$ and $(\varepsilon_k)$ such that for all $k$,

$$\max_{i \in [n]} \|K(\theta_{k+1})_{i,:} - K(\theta_k)_{i,:}\|_1 \leq \varepsilon_k. \tag{2}$$

Here, $\theta_k$ values represent the hyperparameters for each value $k$.

The first property is trivial, while the second one provides a different explainability from that of CKA (in the following subsection), and the third ensures that the alignment metric is stable under hyperparameter variation. With this kernel, we define Manifold-approximated Kernel Alignment (MKA) as:

$$\mathrm{MKA}(K, L) = \frac{\langle KH, LH \rangle}{\sqrt{\langle KH, KH \rangle \langle LH, LH \rangle}}. \tag{3}$$

Despite using non-symmetric kernels, the measure MKA is symmetric ($\mathrm{MKA}(K, L) = \mathrm{MKA}(L, K)$). However, unlike CKA, which performs both row- and column-wise centering, we opted for only row-wise centering. This leaves additional bias terms in the estimation, however, we empirically show in Appendix E that this slight oversight does not make MKA less meaningful. Exploiting the properties of the kernel matrix, we can simplify and characterize MKA by

**Theorem 4.1.** *If the kernels* $K$ *and* $L$ *both follow constant row sum property (A2) with* $K\mathbf{1} = L\mathbf{1} = D\mathbf{1}$, *then* MKA *reduces to*

$$\mathrm{MKA}(K, L) = \frac{\langle K, L \rangle - D^2}{\sqrt{(\langle K, K \rangle - D^2)(\langle L, L \rangle - D^2)}}. \tag{4}$$

**Corollary 4.2.** *If the kernels* $K$ *and* $L$ *follow properties (A1) and (A2) with* $K\mathbf{1} = L\mathbf{1} = D\mathbf{1}$ *and* $D < \sqrt{n}$, *then* $0 \leq \mathrm{MKA}(K, L) \leq 1$.

Theorem 4.1 enables fast computation of MKA, making it more scalable (especially when combined with approximate nearest neighbor search algorithms). Few works (Chen et al., 2021; Huh et al., 2024) have considered sparsifying the kernel matrix of CKA by taking the top-k values in rows/columns. However, these works do not consider constraining the rows/columns of the kernel matrix.

## 4.1 Comparison to CKA

To establish a conceptual relation between MKA and CKA, we provide a Markov kernel interpretation of the algorithms in the following theorems:

**Theorem 4.3** (Markov interpretation of nonlinear CKA)**.** CKA *with RBF Kernel is the cosine similarity (in the Frobenius inner-product space) between left centered degree-weighted grand-mean centered Markov transition matrices, i.e.,*

$$\text{CKA}(K, L) = \frac{\langle HS(P - U), HT(Q - U)\rangle_F}{||HS(P - U)||_F ||HT(Q - U)||_F},\tag{5}$$

*where $S$ and $T$ are diagonal degree matrices with $s_{ii} = \sum_j K_{ij}$ and $t_{ii} = \sum_j L_{ij}$, $P = S^{-1}K$ and $Q = T^{-1}L$ are row-stochastic Markov kernels of the feature sets, and $U = \frac{1}{n}\mathbf{1}\mathbf{1}^T$ is the uniform distribution.*

**Theorem 4.4** (Markov interpretation of MKA)**.** MKA *is the cosine similarity between grand-mean centered Markov transition matrices, i.e.,*

$$\text{MKA}(K, L) = \frac{\langle P - U, Q - U\rangle_F}{||P - U||_F ||Q - U||_F}.\tag{6}$$

Since $P$ and $Q$ are row-stochastic Markov kernels, each row $P_i$ and $Q_i$ has a natural probabilistic interpretation as a neighborhood distribution on the manifold; both algorithms measure the distance from the trivial uniform baseline, $U$. However, CKA is dependent on the row sum of the kernel matrices, which in turn depends on the pairwise distances and the value of $\sigma$. MKA, on the other hand, is independent of the row sum (by design) and therefore it compares how the two representations deviate from the uniform distribution in their local transition structure, without requiring the kernels to be symmetric or positive semidefinite. In particular, if the two representations induce identical neighborhood distributions, $P = Q \neq U$, then $\text{MKA}(K, L) = 1$. Thus, the maximum alignment score corresponds exactly to perfect agreement of the directed neighborhood structure on the manifold.

## 4.2 Implementation

We define a manifold-approximated kernel using a pairwise relationship by

$$K_{ij} = \begin{cases} 1, & \text{if } i = j \\ \exp\left(-\frac{d(x_i, x_j) - \rho_i}{\sigma_i}\right) & \text{if } x_j \in \text{kNN}(x_i, k), \\ 0 & \text{otherwise} \end{cases}\tag{7}$$

where $\text{kNN}(x_i, k)$ contains the $k$-nearest neighbors of $x_i$, $d(\cdot, \cdot)$ is a distance metric, $\rho_i = \min_{x_j \in \text{KNN}(x_i, k)} d(x_i, x_j)$ is the minimum distance from the nearest neighbor and $\sigma_i$ is the row-wise bandwidth parameter. $\sigma_i$ is analogous to the $\sigma$ parameter of the RBF kernel; however, $\sigma_i$ operates only on each row of the kernel matrix, while *sigma* works globally. This bandwidth is computed such that $\sum_j K_{ij} = 1 + \log_2(k)$. This constraint fixes the row of the kernel matrix to a constant, i.e., (A2), and makes the kernel less sensitive to lone outliers. Additionally, this imposes a rank order within the row. The kNN imposes a stricter constraint on the number of points that are considered related compared to CKA, which allows for a softer, more global measure of similarity. This manifold approximation method is similar to that of UMAP, but unlike UMAP, we avoid the symmetrization step to ensure it follows property (A2) and provides some computational efficiency.

With this specific manifold-approximated kernel, we can now discuss the stability of MKA. Overall, $K$ is a graph on the data that depends on only one hyperparameter: $k$ (i.e., $\theta_k = \{k\}$). As $k$ increases, the bandwidth parameter $\sigma_i$ increases slowly, which in turn conforms to property (A3) and thus, MKA becomes a robust metric. To establish this formally, we provide the following theorem:

**Theorem 4.5.** *Let $K^{(k)}$ denote the kernel in Eq. (7) constructed using $k$ nearest neighbors. Assume $d(x_i, x_{(2)}) - \rho_i \geq \delta > 0$, where $x_{(2)}$ is the second nearest neighbor of $x_i$ (ordered by increasing distance).*

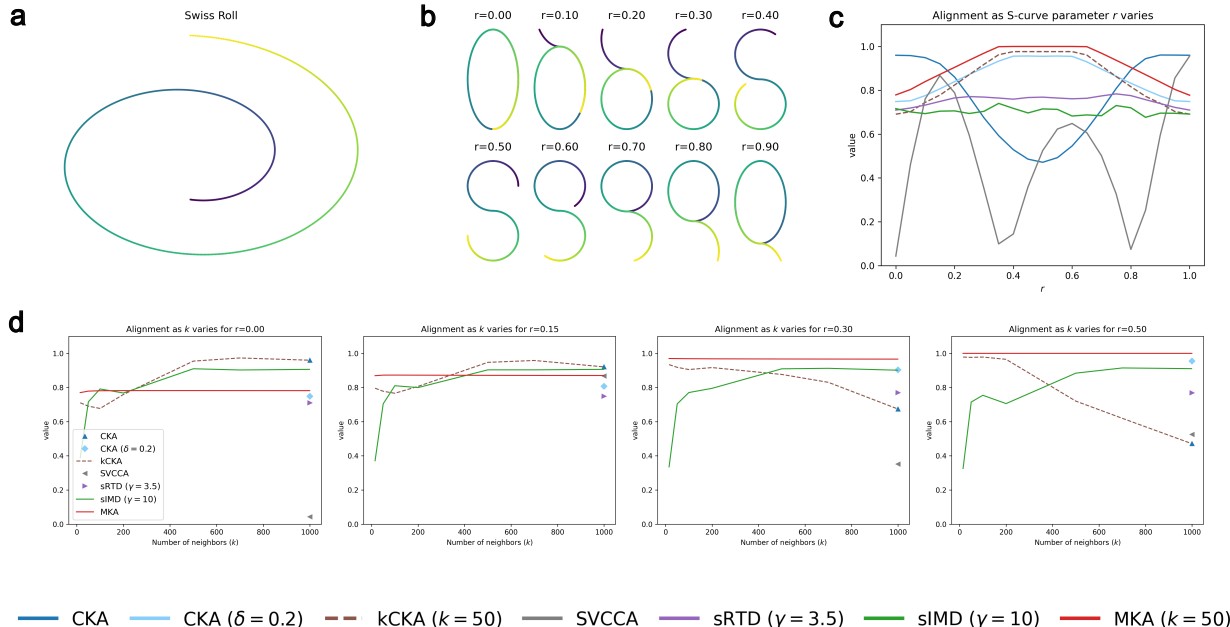

Figure 1: Equivalence of two different shapes with 1-D manifolds. (a) Swiss-roll. (b) S-curve by varying parameter $r$. (c) Alignment for the methods as S-curve parameter, $r$, varies. (d) Alignment for different methods as the number of nearest neighbors, $k$, varies. Note that CKA, RTD, and SVCCA do not have any notion of nearest neighbors; thus, we have plotted these values at the end of the x-axis.

*Further assume $\sigma_{i,k}, \sigma_{i,k+1} \in [\underline{\sigma}, \overline{\sigma}]$ for some $0 < \underline{\sigma} \leq \overline{\sigma} < \infty$, where $\sigma_{i,k}$ and $\sigma_{i,k+1}$ are the bandwidths with $k$ and $k+1$ nearest neighbors, respectively. Then for any $j$,*

$$\left| K_{ij}^{(k+1)} - K_{ij}^{(k)} \right| = \mathcal{O}\left( \frac{\log_2 k}{k} \right), \qquad k \geq 2, \tag{8}$$

*where the implicit constant depends only on $\delta, \underline{\sigma}, \overline{\sigma}$.*

It states that the changes in the kernel as $k$ varies are small, as their element-wise absolute difference is proportional to $log(k)/k$. As $k$ increases, the denominator dominates, making the kernel relatively unchanged. Thus, the MKA metric will be stable as $k$ varies.

## 5 Experiments

In this section, we empirically characterize MKA using various datasets and benchmarks. We compare MKA with several CKA variants with the RBF kernel: 1) CKA($\sigma = M$): $\sigma$ is set to the median, $M$, of the entries of the distance matrix, 2) CKA($\sigma = \delta M$): $\sigma$ is set to $\delta M$ for considering local relationships (we mostly use $\delta = 0.2$ or $0.45$), and 3) kCKA: sparsifying the kernel matrix by considering $k$-nearest neighbors of each sample and setting $\sigma$ to be median of the considered distances giving us a simple manifold approximation. Mathematically, kCKA has the same Markov kernel interpretation as CKA (Theorem 4.3), but uses a non-Mercer kernel like MKA. Along with the CKA variants, we consider Representational Topology Divergence (RTD), Intrinsic Multi-scale Distance (IMD), and Support Vector Canonical Correlation Analysis (SVCCA) metrics. RTD and IMD provide a metric within $[0, \infty]$, with a lower value corresponding to stronger alignment. We scale these values within $[0, 1]$ using the formulae sRTD $= \exp\left(-\text{RTD}/\gamma\right)$ and sIMD $= exp(-\text{IMD}/\gamma)$ for the respective methods and tune $\gamma$ for each experiment (for additional figures for RTD and IMD for the experiments, see Appendix G). In the figures, we explicitly differentiate between IMD (RTD) and sIMD (sRTD), while in the text, we use them interchangeably.

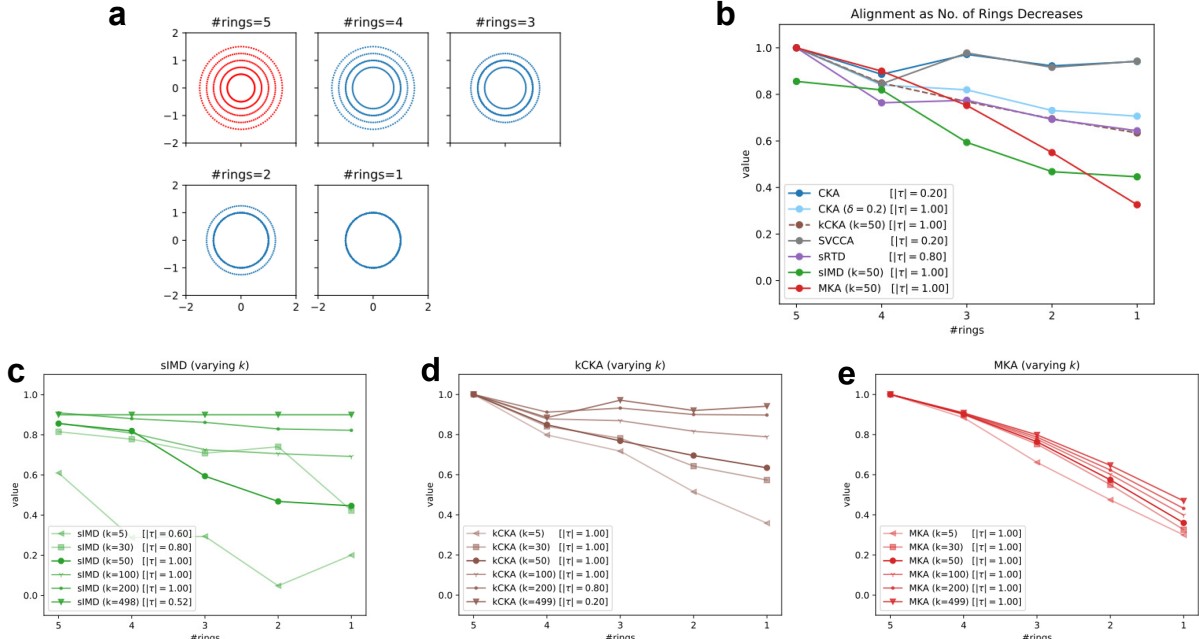

Figure 2: Alignment for the "rings" data. (a) Point clouds used in the clusters experiment. (b) Alignment using various methods, along with Kendall's rank correlation ($\tau$, higher is better). (c-e) Alignment by varying nearest neighbors, $k$, in (c) IMD, (d) kCKA, and (e) MKA. MKA shows the most robustness to the parameter $k$.

## 5.1 Equivalence of Shapes

We start the experiments by comparing two classic shapes: Swiss-roll (Fig. 1(a)) and S-curve (Fig. 1(b), $r = 0.5$). Although the Swiss roll and the S-curve look drastically different, they are topologically equivalent: both lie on a one-dimensional nonlinear manifold. Furthermore, the parameter $r$ in the S-curve can give it different shapes (Fig. 1(b), for details see Appendix C). A color map shows the correspondence among the shapes. For $r < 0.4$ and $r > 0.6$, the colors overlap, and the 1-D manifold disappears. For experiments, we sampled 1000 points from each of the shapes and computed the alignment between them.

CKA with $\sigma = M$ fails to align the manifold of Swiss-roll and S-curve ($r = 0.5$), giving a lower value (Fig. 1(c)). However, for cases where the 1-D manifold structure is absent (e.g., $r < 0.4$ and $r > 0.6$), CKA provides a higher value. On the contrary, CKA with $\delta = 0.2$, kCKA, and MKA properly capture the alignment of the two shapes. At $r = 0.5$, the alignment of the Swiss-roll and S-curve is highest and gets lower as the parameter moves away from this point. RTD and IMD do not show any trends, while SVCCA shows an unrelated oscillatory behavior (from curvature). However, kCKA is more sensitive to the number of nearest neighbors $k$ (Fig. 1(d)), while MKA is very robust to that.

## 5.2 Ranking Structures

In the second test, we reproduce the "rings" experiments that originally appeared in (Barannikov et al., 2022). This dataset consists of 500 points distributed over five concentric rings (radii varying from 0.5 to 1.5). Then, in each iteration, the number of rings decreases (as if one of the rings collapses onto another ring) until it reaches a singular ring (Fig. 2 (a)). Then, we use alignment metrics to compare these formations with the original structure (i.e., five rings). The target of the experiment is to check whether the metrics can track the collapsing rings structure. Kendall's rank correlation, $\tau$, can measure this ranking in a statistical sense (the absolute value is sufficient and thus higher is better).

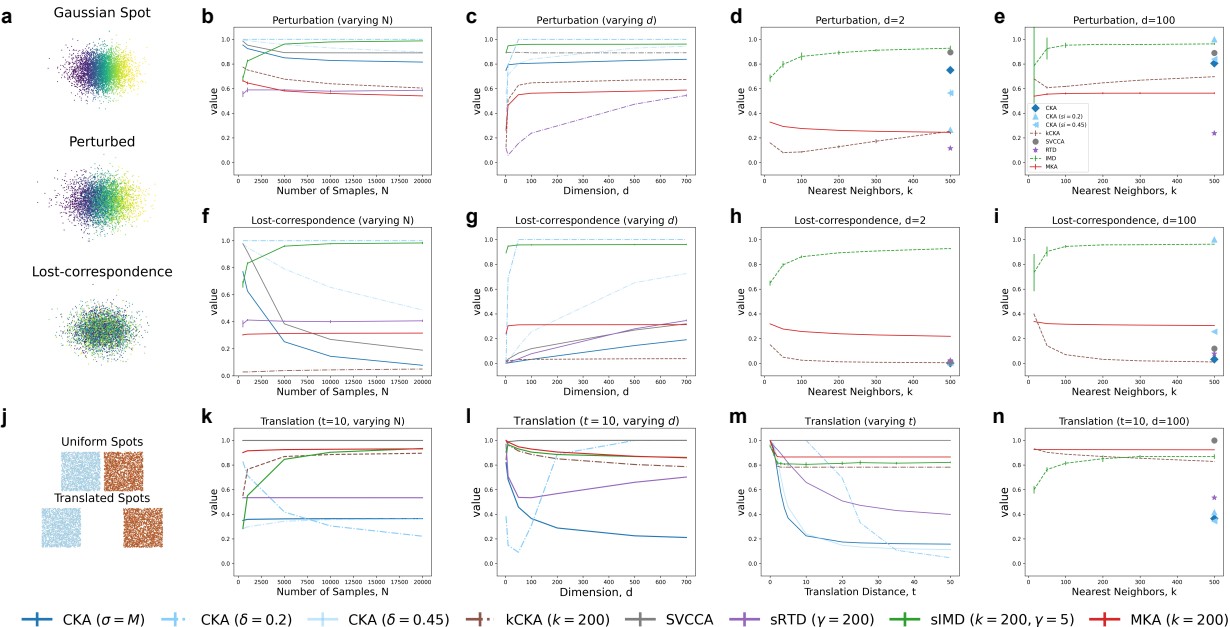

Figure 3: Characterizing MKA using synthetic datasets and comparison to other methods. **(a)** Top: A Gaussian spot; colors identify the position of the points on the x-axis. Middle: Perturbed Gaussian spot. We added noise to the points of the top figure so that the colors slightly overlap. Bottom: A Gaussian spot with no correspondence to the spot on the top. **(b-e)** Alignment between a Gaussian spot and when it is perturbed when (b) number of samples, $N$ ($d = 1000$), and (c) number of dimensions, $d$ ($N = 5000$), varies for various methods, and their performance as number of nearest neighbor, $k$, varies for (d) $d = 2$ and (e) $d = 100$ ($N = 5000$). **(f-i)** Alignment under lost correspondence when (b) number of samples, $N$ ($d = 1000$), and (c) number of dimensions, $d$ ($N = 5000$), varies for various methods, and their performance as number of nearest neighbor, $k$, varies for (d) $d = 2$ and (e) $d = 100$ ($N = 5000$). **(j)** Two uniform spots are located nearby (top) and translated far away (bottom). **(k-n)** Alignment under translation when (k) number of samples, $N$ ($d = 1000$), (l) number of dimensions, $d$ ($N = 5000$), (m) translation distance, $t$, and (n) number of nearest neighbors, $k$, varies. Error bars are drawn up to one standard deviation (5 trials for each experiment).

CKA and SVCCA fail to track this collapsing behavior, while RTD closely reflects the changes. CKA ($\delta = 0.2$), kCKA, IMD, and MKA capture the ranking quite well (Fig. 2(b)). However, varying the nearest neighbor parameter, $k$, causes different behaviors in different methods. IMD shows consistent behavior for $k = 50, 100$, and 200 (Fig. 2(c)). kCKA provides correct ranking only for lower values of $k$; at higher values $k \approx 200$ and above the method fails (Fig. 2(d)). MKA provides correct ranking for all possible values of $k$ (Fig. 2(e)). To the best of our knowledge, ours is the first paper that achieves correct ranking in this dataset.

We explored an additional dataset of similar interest: "clusters". The results are provided in Fig. 6(a) in Appendix B.

### 5.3 Characterizing The Algorithms

In this section, we characterize the algorithms using several synthetic datasets inspired by real-world scenarios. First, we consider the alignment between a d-dimensional Gaussian spot ($x_i \sim \mathcal{N}(\mathbf{0}, I_d)$, Fig. 3(a) top) and its perturbed version ($y_i = x_i + 0.5\mathcal{N}(\mathbf{0}, I_d)$, Fig. 3(a) middle). Such a scenario may occur when a representation learning algorithm runs repeatedly. This results in altered orders of the points in the point cloud (seen as colors slightly overlapping). As the number of samples in the spots increases ($d = 1000$, Fig. 3(e)), their alignment values using different methods decrease slightly (notable exceptions are IMD, which increases and then stabilizes, and CKA ($\delta = 0.2$), which saturates). This is expected, as the denser the spot gets, the higher the chance of orders within the point cloud. However, the dimensionality ($d$) of the data affects the

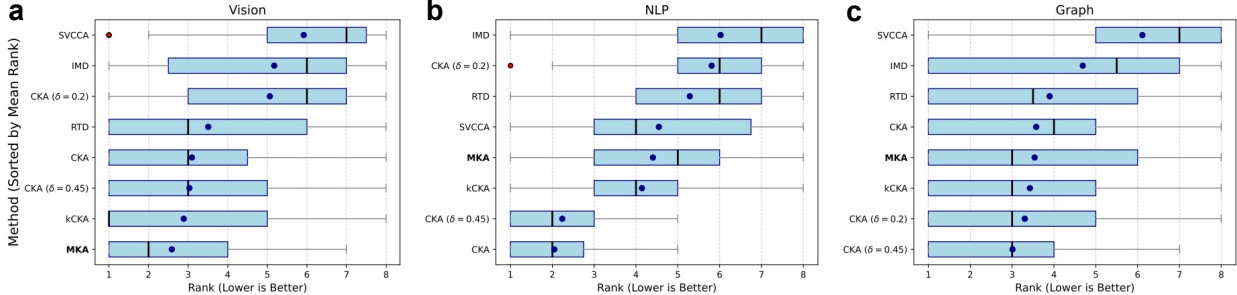

Figure 4: Aggregated ranks of alignment measures using the ReSi benchmark across different models and tests, separated by domains: (a) vision, (b) natural language processing, and (c) graph. Boxplots indicate quartiles of rank distributions; the whiskers extend up to 1.5 times the interquartile range. The black dots indicate the mean rank.

values differently ($N = 5000$, Fig. 3(c)). All methods, except CKA with $\delta = 0.2, 0.45$ and RTD, are fairly consistent as $d$ increases. CKA with $\delta = 0.2$ saturates rapidly, while with $\delta = 0.45$ it approaches saturation as $d$ increases. RTD starts with a lower value, and it increases with $d$. Additionally, $k$ CKA shows inconsistent behavior as the number of nearest neighbors ($k$) increases and sIMD shows high variance, while MKA values remain consistent across a wide range (Fig. 3(d,e)). Overall, MKA is more restrictive to perturbations in the features than other methods.

We can take this scenario to the extreme and make the colors completely overlap each other (Fig. 3(a), bottom). The orderings (based on some criterion) of both the Gaussian spots will not correspond to each other at all, and thus, we call it a lost-correspondence scenario. The CKA (and $\delta = 0.45$), SVCCA, measures are sensitive to the number of samples, while $k$ CKA, RTD, and MKA are fairly consistent ($d = 1000$, Fig. 3(f)). The CKA ($\delta = 0.45$ as well) measure tends to increase with higher data dimensionality, reflecting the effect of the curse of dimensionality ($N = 5000$, Fig. 3(g)). SVCCA and RTD also behave similarly. kCKA, IMD, and MKA, on the other hand, are fairly robust and less affected by the curse. However, like before, kCKA is highly sensitive to the number of nearest neighbors ($k$), which gets resolved at a higher value of $k \geq 200$ (Fig. 3(h,i)). Like previously, MKA is consistent for a wide range of $k$, even for values smaller than 200. Overall, MKA is more consistent with varying hyperparameters than other methods.

Finally, we consider two uniform spots separated by a small distance (Fig.3(j); this scenario is inspired by Davari et al. (2023)). Both spots ($N = 2500$ each) are drawn from uniform distribution by $x_i \sim \mathcal{U}(-0.5, 0.5)$ and $y_i \sim p + \mathcal{U}(-0.5, 0.5)$ with $p = [1.1 + t, 0, 0, \ldots, 0]$, where the translation distance, $t(> 0)$, controls the separation of the two spots. Regardless of the translation distance, the topology of the data remains the same, and alignment should be high. Surprisingly, most methods are consistent as the number of samples increases (except CKA with $\delta = 0.2$). CKA gives a low alignment score between the two representations, while kCKA and IMD stabilize as the number of samples increases. We get a more diverse result as the number of dimensions, $d$, (Fig. 3(l)) and the translation distance, $t$, (Fig. 3(m)) vary. CKA fails to capture this phenomenon. As $t$ increases, CKA value decreases; even using a smaller bandwidth $\delta = 0.2$ fails. Surprisingly, RTD also joins CKA and fails to capture the invariance of topology (RTD fuses individual graph of the two representations into a signle graph. In the fused graph, the large distances dominate and thus RTD follows CKA). SVCCA shows maximum alignment between the two representations under all circumstances. In contrast, kCKA, IMD, and MKA settle to a constant and higher number as $d$ and $t$ increase. As $k$ increases, the pattern mirrors the earlier experiments; by $k \simeq 100$ most methods stabilize, whereas MKA is already consistent at small $k$ (Fig. 3(m)).

### 5.4 Evaluation using Representation Similarity (ReSi) Benchmark

Representation Similarity (ReSi) Benchmark (Klabunde et al., 2025) is a collection of six different tests to assess the performance of representational similarity or alignment metrics. The tests are Correlation to Accuracy Difference (correlates the alignment score of a pair of models with the absolute difference in

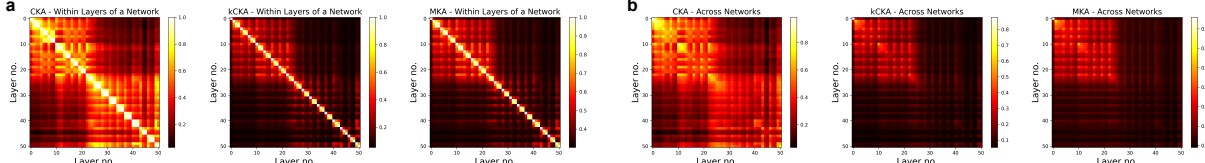

Figure 5: Alignment between features from different layers of ResNet-50 trained on the CIFAR-10 dataset. (a) Alignment between layers of a network using (left) CKA, (middle) kCKA, and (right) MKA. (b) Alignment between layers across different networks using (left) CKA, (middle) kCKA, and (right) MKA. The results are an average of 10 instances of ResNet-50 trained on CIFAR-10, each initialized randomly and using a subset of 10000 samples from the test set.

their accuracies), Correlation to Output Difference (correlates alignment metrics with the instance-wise disagreement and Jensen-Shannon divergence of the predictions), Label Randomization (evaluates whether alignment metrics can separate models trained with varying levels of label corruption), Shortcut Affinity (evaluates whether alignment metrics can distinguish models trained with spurious shortcut features at different shortcut–label correlation strengths), Augmentation (evaluates whether alignment metrics can stratify models trained with varying augmentation strengths, when all are tested on the same clean, non-augmented set), and Layer Monotonicity (evaluates whether alignment score decreases as the distance between layers increases within the same model). We used the ReSi tests on vision, natural language processing (NLP), and graph domain tasks. For the vision task, we used the ImageNet-100 dataset and seven representative networks from three different architectures: Residual Networks (ResNet-18, ResNet-34, ResNet-101) (He et al., 2016), Visual Geometry Group networks (VGG-11, VGG-19) (Simonyan & Zisserman, 2015), and Vision Transformers (ViT B32, ViT L32) (Dosovitskiy et al., 2021). For the language task, we used the MNLI dataset (Williams et al., 2018) and two language models: BERT (Devlin et al., 2019) and ALBERT (Lan et al., 2020). For the graph data, we explored three different datasets: Cora (Kipf & Welling, 2017), Flickr (Hamilton et al., 2017), and OGBN-Arxiv (Veličković et al., 2018), and four different graph networks: Graph Convolutional Network (GCN) (Yang et al., 2016), Graph Sample and Aggregate (SAGE) (Zeng et al., 2020), Graph Attention Network (GAT) (Hu et al., 2020), and Position-aware Graph Neural Networks (PGNN) (You et al., 2019). The original ReSi benchmark concluded that no method consistently outperforms others across domains. We expect to find a similar result here as well.

Figure 4 summarizes mean-rank distributions per domain (lower is better). We used $k = 100$ to compute the nearest neighbor graphs. In the vision domain, MKA attains the best central tendency with the tightest spread, edging out kCKA and clearly outperforming CKA and RTD (Fig. 4(a)). On the other hand, in the NLP domain, CKA (and with $\delta = 0.45$) is a clear winner (mean, median, and variance); however, MKA remains within striking distance while maintaining a compact dispersion, i.e., it is competitive without the heavy sensitivity to kernel bandwidths Fig. 4(b)). Finally, for graphs, the methods that focus on local geometry, i.e., MKA, kCKA, and CKA ($\delta = 0.2, 0.45$), cluster together (same median for all of them and the mean is within $\pm 1$). Overall, MKA delivers top performance in vision, matches the best local methods on graphs, and stays robustly competitive in NLP, making it a consistent, parameter-light choice when a single alignment metric must generalize across modalities.

MKA is particularly advantageous when curvature and local manifold structure matter (vision representations), while CKA can remain preferable in nearly linear regimes (NLP representations). For graphs, the methods perform comparably. Note that, ReSi benchmark doesn't probe dimensionality and sample numbers (as we have done in Section 5.3).

### 5.5 Neural Network Representations

In this section, we explore the representational similarity using ResNet-50 models trained on the CIFAR-10 dataset. First, we compute alignment between feature representations extracted from different layers (after activation) of the network to investigate how representational structure evolves across the depth of the model (Fig. 5(a)). We considered only CKA, kCKA, and MKA for this experiment (as other methods have been

explored elsewhere) and highlight how these three competing methods process information. Using CKA, we can reproduce the famous block structure (Kornblith et al., 2019; Nguyen et al., 2022). As we said previously, dominant clusters cause the block structure (Nguyen et al., 2022). However, when a k-nearest neighbor graph constrains the kernel, this block structure disappears. For kCKA, block structure appears in the early layers, but they less pronounced in the later layers. MKA exhibits a similar behavior; however, for adjacent layers ($\pm1$ around the diagonal), it shows higher alignment than kCKA, suggesting equivalence of the directed neighborhood graph, whereas kCKA additionally reflects within-neighborhood distances. Overall, CKA is sensitive to dominant high-density regions of large distances in the distance matrix compared to kCKA, and MKA is even less so. When we compare features of the networks to each other, the block structure is still present for CKA, less pronounced for kCKA and MKA (Fig. 5(b)). This suggests that the same architecture, under different random initializations, can converge to distinct internal orientations, i.e., manifold-level perturbations of the learned representation, despite similar test accuracy; this effect is often less apparent under CKA.

## 6   Discussion and Conclusions

In this paper, we introduced Manifold-approximated Kernel Alignment (MKA), provided a theoretical arguments to explain the algorithm, characterized it using several datasets, and compared it to different algorithms. We found that methods applying k-NN graph are suitable for comparing topological structures (MKA and kCKA in Figs. 1,2, and 6) and sometimes even better than their topological counterparts. By analyzing Gaussian distributions and their perturbations (Fig. 3), we showed that methods that rely on local neighborhoods show less sensitivity to intrinsic parameters of datasets (number of samples and dimensionality). However, most methods require hyperparameter tuning, while MKA is consistent with varying hyperparameter. When tested with uniform spots and their translation, we found MKA to be robust, even compared to other topological methods (Fig. 3(j-m)). We then show that MKA is competitive with contemporary methods across a wide range of tasks on the ReSi benchmark (Fig. 4). By analyzing representations of neural networks, we conclude that MKA perceives the neural network representations differently than CKA, with kCKA working as a conceptually intermediate step.

CKA is globally density-weighted: a single high-density region of large distances in the kernel matrix can dominate the score. kCKA mitigates this by restricting interactions to local k-NN neighborhoods, making it less susceptible to interactions from large distances. MKA goes further by ordering neighbors within each neighborhood and assigning weights that depend on rank and local density. In essence, vanilla CKA ignores ranks and depends solely on pairwise distances, while kCKA merely dichotomizes pairs into "within-k" vs "outside-k" and treats the k nearest neighbors essentially uniformly. RTD, a topology-based approach, sometimes aligns with the underlying manifold structure, but in other cases exhibits behavior closer to CKA. We attribute this to RTD's graph-fusion step, which can overweight long-range edges, i.e., large distances tend to persist across filtration scales and thus dominate the resulting barcodes. Accordingly, RTD primarily reflects multi-scale persistence, whereas kCKA and MKA are single-scale methods based on k-NN neighborhoods. At a fixed k, the k-NN graph is either faithful or not; persistence, by averaging over scales, can smooth away local structure, occasionally drifting toward density-driven behavior.

Future works could explore other kernel functions, e.g., effective resistance (Doyle & Snell, 1984), diffusion distance Coifman & Lafon (2006), and $\alpha$-decaying kernels Moon et al. (2019). This technique would find usage wherever alignment is beneficial, e.g., in neuroscience for monitoring brain activity, neural decoding, brain representation analysis, and graph learning for protein interactions.

## Software and Data

All the data used in this paper are publicly available. Codes have been submitted as supplementary material during review.

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

## Appendix

In the supplementary material, we provide some additional details and results. Apendix A provides the proofs of the theorems. Appendix B provides details of the "clusters" data experiment. Section C gives details of the Swiss-roll and S-curve. In Appendix D, we provide short comment on the computational complexity of the algorithms. In Appendix E, we discuss CKA with manifold approximation. In Appendix F, we discuss the linear kernel of kCKA. Appendix G gives supplementary figures for the experiments in the main text. We provide implementation details in Appendix H. Finally, we follow it by detailing the ReSi benchmark in Appendix I and corresponding supplementary results in Section J.

## A  Proofs

*(Proof of Theorem 4.1).* Let $KH = \bar{K}$ and $LH = \bar{L}$. Using the constant row sum property we show:

$$\bar{K}_{ij} = K_{ij} - \frac{1}{N}\sum_j K_{ij}$$

$$= K_{ij} - \frac{1}{N}D. \tag{9}$$

Now, we can compute the inner product,

$$\langle \bar{K}, \bar{K} \rangle = \sum_{i,j}(K_{ij} - \frac{1}{N}D)^2$$

$$= \sum_{i,j}\left(K_{ij}^2 - \frac{2}{N}DK_{ij} + \frac{1}{N^2}D^2\right)$$

$$= \sum_{i,j}K_{ij}^2 - \frac{2}{N}D\sum_{i,j}K_{ij} + \frac{1}{N^2}D^2\sum_{i,j}1$$

$$= \sum_{i,j}K_{ij}^2 - D^2$$

$$= \langle K, K \rangle - D^2 \tag{10}$$

We used the fact that $\sum_{i,j}K_{ij} = ND$ and $\sum_{i,j}1 = N^2$. Similarly, $\bar{L}_{ij} = L_{ij} - \frac{1}{N}D$ and $\langle \bar{L}, \bar{L} \rangle = \langle L, L \rangle - D^2$. Finally,

$$\langle \bar{K}, \bar{L} \rangle = \sum_{i,j}(K_{ij} - \frac{1}{N}D)(L_{ij} - \frac{1}{N}D)$$

$$= \sum_{i,j}K_{ij}L_{ij} - \frac{1}{N}D(K_{ij} + L_{ij}) - \frac{1}{N^2}D^2$$

$$= \sum_{i,j}K_{ij}L_{ij} - D^2$$

$$= \langle K_{ij}, L_{ij} \rangle - D^2 \tag{11}$$

$$\square$$

*(Proof of Corollary 4.2).* We start from the inner products,

$$\langle K, K \rangle - D^2 = \sum_{i,j}K_{ij}^2 - D^2$$

$$= \sum_{i,i}1 + \sum_{i,j,i\neq j}K_{ij}^2 - D^2$$

$$= N - D^2 + \sum_{i,j,i\neq j}K_{ij}^2. \tag{12}$$

Similarly,

$$\langle L, L \rangle - D^2 = N - D^2 + \sum_{i,j,i \neq j} L_{ij}^2 \tag{13}$$

And finally,

$$\langle K_U, L_U \rangle - D^2 = N - D^2 + \sum_{i,j,i \neq j} K_{ij} L_{ij} \tag{14}$$

The value $\sum_{i,j,i \neq j} K_{ij} L_{ij}$ can be zero if the nearest neighbors in the kernels do not overlap. Otherwise, this value is positive. Thus, the lower bound is guaranteed when $N > D^2$. The upper bound is due to Cauchy–Schwarz inequality. $\qquad \square$

*Proof of Theorem 4.3.* A simple calculation shows that

$$(KH)_{ij} = K_{ij} - \frac{1}{n} \sum_j K_{ij} \tag{15}$$

$$= K_{ij} - \frac{s_i}{n} \tag{16}$$

$$= s_i \Big( \frac{K_{ij}}{s_i} - \frac{1}{n} \Big) \tag{17}$$

$$= s_i \big( P_{ij} - U_{ij} \big), \tag{18}$$

Thus, in matrix form it becomes, $KH = S(P - U)$, and $LH = T(Q - U)$. Here, $P - U$ and $Q - U$ are the grand-mean-centered Markov kernels, as defined in Prothero et al. (2023). The derivation of the matrices $P$ and $Q$ follow from standard graph to Markov notations (Lovász, 1993; Coifman & Lafon, 2006). This makes

$$HKH = HS(P - U) \quad \text{and} \quad HLH = HT(Q - U). \tag{19}$$

$\qquad \square$

*Proof of Theorem 4.4.* For manifold-approximated kernels following property (A2), $\sum_j K_{ij} = \sum_j L_{ij} = D$. $\qquad \square$

**Lemma A.1** (Stability of the bandwidth under $k \to k+1$)**.** *Fix a point $x_i$, let the neighbor distances be*

$$d_1 \leq d_2 \leq \ldots, d_j = d(x_i, x_{(j)}) \tag{20}$$

*where $x_{(j)}$ is the $j^{th}$ nearest neighbor of $x_i$ and $\rho = d_1$, and define*

$$\Delta_j = d_j - \rho. \tag{21}$$

*Let $\sigma_k$ and $\sigma_{k+1}$ be two bandwidth parameters for two kernels obtained from Eq. (7) for $k$ and $k+1$ nearest neighbors. Assume the following non-degeneracy bounds hold for the bandwidths:*

$$\Delta_2 \geq \delta > 0, \ \text{and} \ \sigma_k, \sigma_{k+1} \in [\underline{\sigma}, \overline{\sigma}] \ \text{for some} \ 0 < \underline{\sigma} < \overline{\sigma} < \infty. \tag{22}$$

*Then $\forall \ k \geq 2$,*

$$|\sigma_{k+1} - \sigma_k| \leq \frac{\overline{\sigma}^2}{\delta} \exp\Big( \frac{\delta}{\underline{\sigma}} \Big) \Big( \log \Big( 1 + \frac{1}{k} \Big) + \frac{\log k - 1}{k - 1} \Big) = \mathcal{O}\Big( \frac{\log k}{k} \Big) \tag{23}$$

*Note: all the $\log$'s in the proofs use base 2. For notational simplicity we omit the base.*

*Proof.* Let

$$F_k(\sigma) = \sum_{j=2}^{k} \exp(-\Delta_j/\sigma) \tag{24}$$

Then, $F_k(\sigma_k) = \log(k) - 1$, $F_{k+1}(\sigma_{k+1}) = \log(k+1) - 1$ and $F_{k+1}(\sigma) = F_k(\sigma) + \exp(-\Delta_{k+1}/\sigma)$. By applying mean value theorem, $\exists\, \varepsilon$ between $\sigma_{k+1}$ and $\sigma_k$ such that,

$$F_{k+1}(\sigma_{k+1} - F_{k+1}(\sigma_k) = F'_{k+1}(\varepsilon)(\sigma_{k+1} - \sigma_k). \tag{25}$$

Thus, we obtain:

$$|\sigma_{k+1} - \sigma_k| = \frac{|F_{k+1}(\sigma_{k+1}) - F_{k+1}(\sigma_k)|}{F'_{k+1}(\varepsilon)} \tag{26}$$

Let's focus on the numerator of the equation (26) first:

$$|F_{k+1}(\sigma_{k+1}) - F_{k+1}(\sigma_k)| = \left| \log\left(1 + \frac{1}{k}\right) - \exp\left(-\frac{\Delta_{k+1}}{\sigma_k}\right) \right| \tag{27}$$

$$\leq \log\left(1 + \frac{1}{k}\right) + \exp\left(-\frac{\Delta_{k+1}}{\sigma_k}\right) \tag{28}$$

To shape the exponential into a more desirable form, we notice that,

$$\exp\left(-\frac{\Delta_{k+1}}{\sigma_k}\right) \leq \exp\left(-\frac{\Delta_k}{\sigma_k}\right) \tag{29}$$

$$= \min_{2 \leq j \leq k} \exp\left(-\frac{\Delta_j}{\sigma_k}\right) \tag{30}$$

$$\leq \frac{1}{k-1} F_k(\sigma_k) \tag{31}$$

$$= \frac{\log k - 1}{k - 1} \tag{32}$$

Thus, inequality (28) becomes:

$$|F_{k+1}(\sigma_{k+1}) - F_{k+1}(\sigma_k)| \leq \log\left(1 + \frac{1}{k}\right) + \frac{\log k - 1}{k - 1} \tag{33}$$

Now, we focus on the denominator of the equation (26). First, let's look at the derivative:

$$F'_{k+1}(\sigma) = \sum_{j=1}^{k+1} \frac{\Delta_j}{\sigma^2} \exp\left(-\frac{\Delta_j}{\sigma}\right) \geq \frac{\Delta_2}{\sigma^2} \exp\left(-\frac{\Delta_2}{\sigma}\right). \tag{34}$$

Under assumption (22), for any $\sigma \in [\underline{\sigma}, \overline{\sigma}]$, we can write

$$F'_{k+1}(\sigma) \geq \frac{\delta}{\overline{\sigma}^2} \exp\left(-\frac{\delta}{\underline{\sigma}}\right) =: m > 0. \tag{35}$$

$\varepsilon$ lies between $\sigma_{k+1}$ and $\sigma_k$ implies $\varepsilon \in [\underline{\sigma}, \overline{\sigma}]$. Thus, $F'_{k+1}(\varepsilon) \geq m$.

Plugging in (28) and (35) into (26), we obtain the desired result.

$\square$

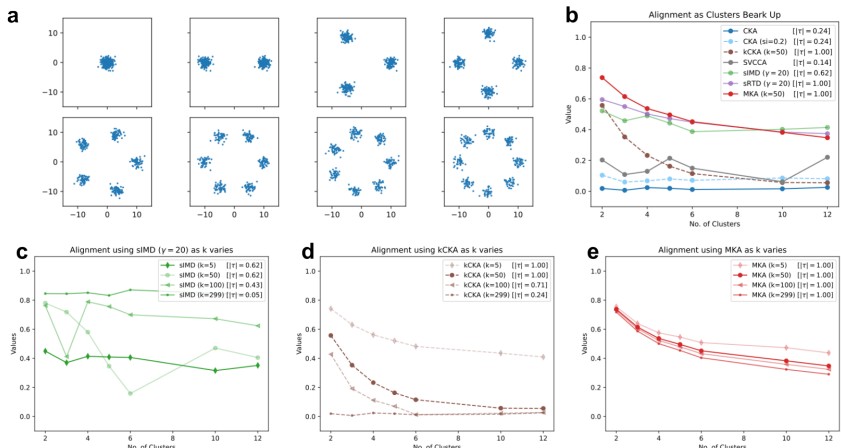

Figure 6: Alignment for the "clusters" data. (a) Point clouds used in the clusters experiment. (b) Alignment using various methods, along with Kendall's rank correlation (higher is better). (c-e) Alignment by varying nearest neighbors, $k$, in (c) IMD, (d) kCKA, and (e) MKA. MKA shows the most robustness to parameters.

*Proof of Theorem 4.5.* The diagonal and the first nearest neighbor stays unchanged. $K_{ii} = 1$ and $K_{i(1)} = \exp(0) = 1$ for all $k$. Entries that are zero in both kernels also remain unchanged.

Now, let's consider the neighbors that exist in both graphs (ranks $r \leq k$). If $x_j = x_{(r)}$ with $2 \leq r \leq k$, then

$$K_{i,(r)}^{(k)} = \exp\left(-\frac{\Delta_r}{\sigma_k}\right) \text{ and } K_{i,(r)}^{(k+1)} = \exp\left(-\frac{\Delta_r}{\sigma_{k+1}}\right) \tag{36}$$

These are the same functions with variables $\sigma_k$ and $\sigma_{k+1}$. Let, $w(\sigma) = \exp(-\Delta_r/\sigma)$. Then using the mean value theorem, we have

$$|w(\sigma_{k+1} - w(\sigma_k)| \leq \sup_{\sigma > 0} |w'(\sigma)| \cdot |\sigma_{k+1} - \sigma_k| \tag{37}$$

The supremum is attained at $\sigma = \Delta_r/2$. This result in

$$|K_{i,(r)}^{(k+1)} - K_{i,(r)}^{(k)}| \leq \frac{4}{e^2 \Delta_r} |\sigma_{k+1} - \sigma_k|. \tag{38}$$

By assumption, $\Delta_r \geq \Delta_2 \geq \delta > 0$ and using the result from Lemma A.1, we have

$$\left|K_{i,(r)}^{(k+1)} - K_{i,(r)}^{(k)}\right| \leq \frac{4}{e^2 \delta} |\sigma_{k+1} - \sigma_k| = \mathcal{O}\left(\frac{\log k}{k}\right) \tag{39}$$

Now we consider the new edge $(k+1)$-th neighbor:

$$K_{i,(k+1)}^{(k)} = 0, \quad K_{i,(k+1)}^{(k+1)} = \exp\left(-\frac{\Delta_{k+1}}{\sigma_{k+1}}\right) \tag{40}$$

Following the same logic from Eqs. (29)-(32), we can show that

$$\left|K_{i,(k+1)}^{(k+1)} - K_{i,(k+1)}^{(k)}\right| = \exp\left(-\frac{\Delta_{k+1}}{\sigma_{k+1}}\right) \leq \frac{\log(k+1) - 1}{k} = \mathcal{O}\left(\frac{\log k}{k}\right) \tag{41}$$

$\square$

## B   Clusters Data

Similar to the "rings" data, the "clusters" data was also compiled by Barannikov et al. (2022). The "clusters" set consists of 300 points sampled for a 2D normal distribution (mean=(0, 0)). Then the points are split into 2, 3, ..., 12 by moving them into a circle of radius 10 (Fig. 6). Then, we use alignment metrics to compare these formations with the original structure (i.e., one cluster). The target of the experiment is to check whether the metrics can track that the data breaks into multiple clusters. Kendall's rank correlation, $\tau$, can measure this in a statistical sense (for our case, the absolute value is sufficient and thus higher is better).

CKA, SVCCA, and IMD fail to track this clustering behavior, while kCKA, RTD, and MKA capture the ranking quite well (Fig. 6(b)). However, varying the nearest neighbor parameter, $k$, causes different behaviors in different methods. IMD shows inconsistent behavior (Fig. 6(c)). kCKA provides correct ranking only for lower values of $k$; at higher values $k \approx 100$ and above the method fails (Fig. 6(cd). MKA provides correct ranking for all possible values of $k$ (Fig. 6(e)).

## C   Details of Swiss-roll and S-Curve

Swiss-roll and S-curve are parameterized by variable $t \in [0, 1]$. S-curve contains an additional control parameter $r \in [0, 1]$ that determines the shape. $r = 0.5$ gives the familiar S-curve used in many studies. We only consider 2-D shapes in this study.

**Swiss-Roll:**

$$z = \frac{3\pi}{2}(1 + 2t) \tag{42}$$

$$x_1 = z \cos(z) \tag{43}$$

$$x_2 = z \sin(z) \tag{44}$$

**S-Curve:**

$$z = 3\pi(t - r) \tag{45}$$

$$y_1 = \sin(z) \tag{46}$$

$$y_2 = \text{sgn}(z)(\cos(z) - 1) \tag{47}$$

## D   Computational complexity

Let's assume the two representations have n samples each with $d_1$ and $d_2$ dimensions, respectively. Most algorithms discussed in the paper rely on nearest neighbor search and matrix multiplications. Particularly, constructing the k-nearest neighbor graphs ($O(n^2(d + \log k))$) is the costliest operation within many of them. Additionally, MKA relies on bisection method to compute the $\sigma_i$ values (Eq. 7) with a complexity $O(nk \log(\Delta/\epsilon))$, where $\Delta$ is the search range and $\epsilon$ is the tolerance. For MKA, $\log(\Delta/\epsilon) = \log(1000/10^{-12}) \simeq 50$ is a constant, which we ignore. Overall, the complexity of the MKA is $O(n^2(1 + d_1 + d_2 + \log k + nk))$. The complexity of the other algorithms are: kCKA - $O(n^3 + n^2(d_1 + d_2 + \log k)$, CKA - $O(n^3 + n^2(d_1 + d_2))$. Thus, all these methods have cubic complexity in n. The compelxity of SVCCA is $O(nd_1 \min(n, d_1) + nd_2 \min(n, d_2))$ (dominated by the singular value decomposition). RTD complexity depends on two factors: computing the distance matrix, which is the same as others, and computing the topological barcode, which is cubic in the number of simplexes (Barannikov et al., 2022). IMD is dominated by constructing the k-NN graph and performing m-steps of stochastic Lanczos quadrature algorithm with $n_v$ starting vectors ($O(n_v(m \log m + knm))$), giving an overall complexity of $O(n^2(d_1 + d_2 + \log k) + n_v(m \log m + knm))$ (Tsitsulin et al., 2020). On the other hand, the space complexity is roughly the same for all the algorithms, primarily to store the kernel matrices, and thus it is dominated by the $O(n^2)$ term (or $O(nk)$ if only k-NN graphs are stored).

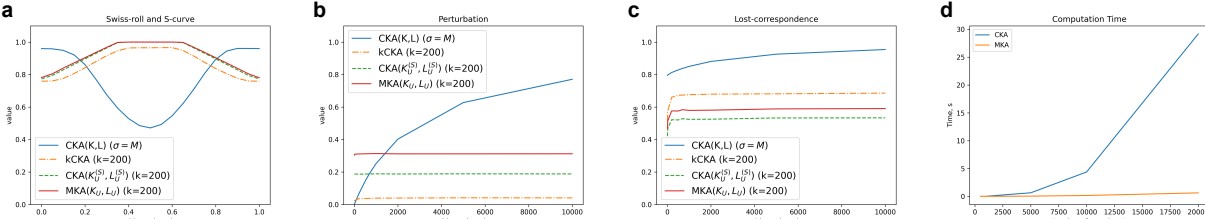

Figure 7: Effect of Kernel Approximation on the CKA algorithm. (a) Alignment between Swiss-roll and S-curve. (b,c) Gaussian spots under (b) perturbation and (c) lost-correspondence. CKA with manifold approximation (CKA($K_U^{(S)}, K_L^{(S)}$) behave similar to MKA, but with less bias. (d) Computation time for CKA and MKA. MKA require much less time than CKA (average of 5 runs). Note that we have excluded the computation time for the kernel matrix.

## E  CKA with Manifold Approximation

We can symmetrize the manifold-approximated kernel matrix, $K_U$ (we define it as $K_U$ here to differentiate from the RBF kernel matrix, $K$), using the probabilistic t-conorm given by

$$K_U^{(S)} = K_U + K_U^T - K_U \circ K_U^T, \tag{48}$$

where $\circ$ denotes element-wise multiplication. This operation does not guarantee a positive semidefinite kernel. However, we can now directly apply CKA on the approximated kernels $K_U^{(S)}$ and $L_U^{(S)}$. The CKA results obtained from this kernel matrix behave similarly to those of MKA but with less bias (Fig. 7(b-c)). However, computing MKA requires much less time compared to CKA (Fig. 7(d), using `NumPy` (Harris et al., 2020)).

## F  Linear vs Non-linear CKA, kCKA

Linear and non-linear CKA in its default form provides similar values and has been known empirically Kornblith et al. (2019); Davari et al. (2023) and recently, theoretically (Theorem 3.1, Alvarez (2022). Following this, we can claim the following for linear and non-linear kCKA:

**Corollary F.1** (Linear vs. Non-linear kCKA). kCKA($K_{\text{RBF}}, L$) = kCKA($K_{\text{LIN}}, L$) + $O(1/\sigma^2)$ *as* $\sigma \to \infty$. *Here,* $K_{\text{RBF}}$ *is the RBF kernel matrix with bandwidth* $\sigma$, $K_{\text{LIN}}$ *is the linear kernel matrix, and* $L$ *is any positive definite symmetric kernel matrix.*

In our implementation of kCKA, we constrained $\sigma$ to be the median of the distances within the k-NN set, which is often small compared to its CKA counterpart. As a result, while linear CKA and non-linear CKA can be equivalent by default implementation, it is hardly the case for kCKA. To make them equivalent, one has to arbitrarily set a large $\sigma$, which we consider an uncommon scenario. However, kCKA is more sensitive to the value of $k$, and as $k$ is increased

## G  Additional Details of Experiments

From Figs. 8-11 we show additional data for the experiment from Fig. 3. Figure 8 shows the dependence on the nearest neighbor parameter $k$ to obtain a stable result. Overall, MKA is stable in all scales, while others need a large value of $k$. Figure 9 shows additional results for $t = 50$ (in the main text, we only showed $t = 10$).

In the main text, we scaled RTD and IMD values to $[0, 1]$ using an exponential function so that it becomes easier to compare with MKA and CKA variants. Here we show (Figs. 10 and 11) the raw values of RTD and IMD for some of the experiments from Fig. 3. In many cases, these algorithms don't show any trends. Moreover, their raw values are all over the place.

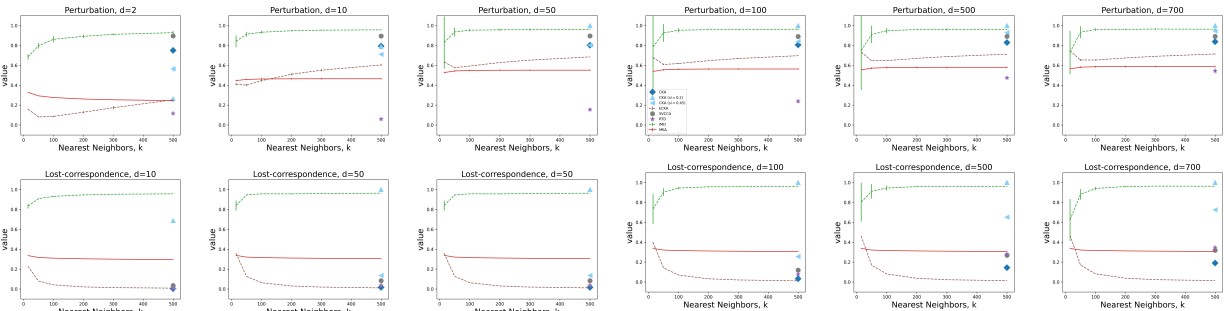

Figure 8: Dependency of the algorithms on nearest neighbor parameter $k$ for various algorithms.

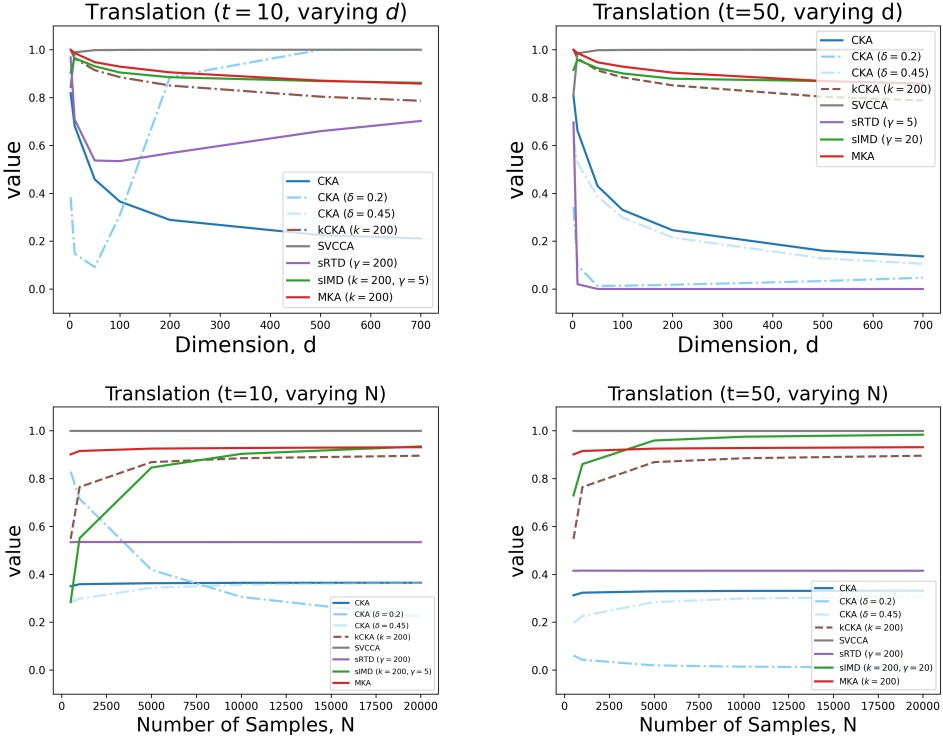

Figure 9: Additional Data for the Uniform Spots experiment. (left column) reproducing data from the main text for $t = 10$. (right column) data for $t = 50$.

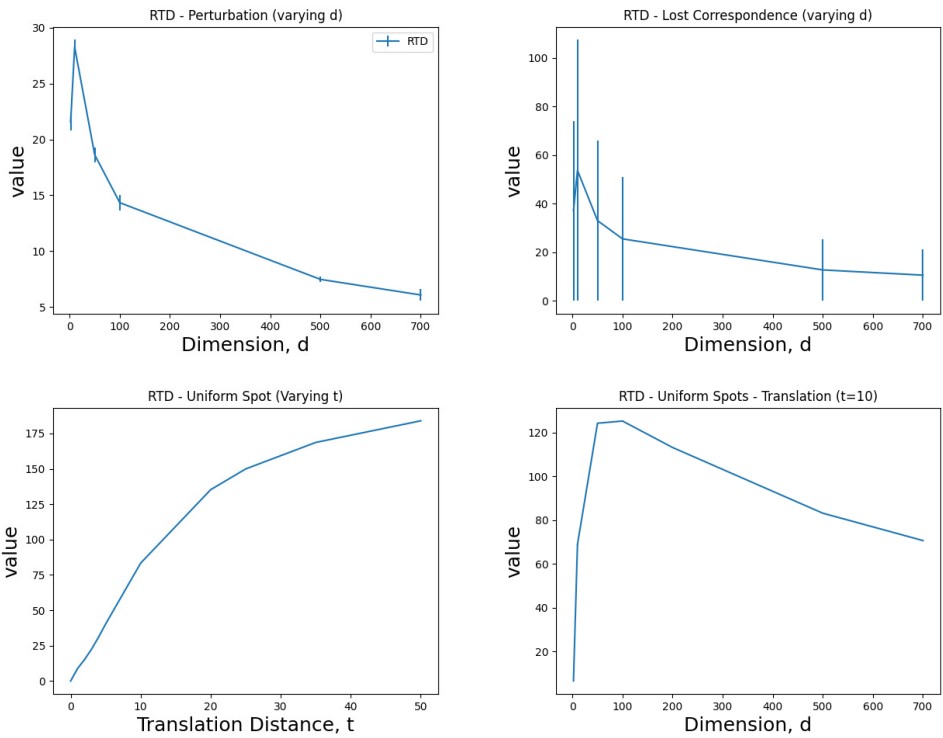

Figure 10: RTD values for a few experiments from Fig. 3. For lost correspondence, RTD exhibits higher variance.

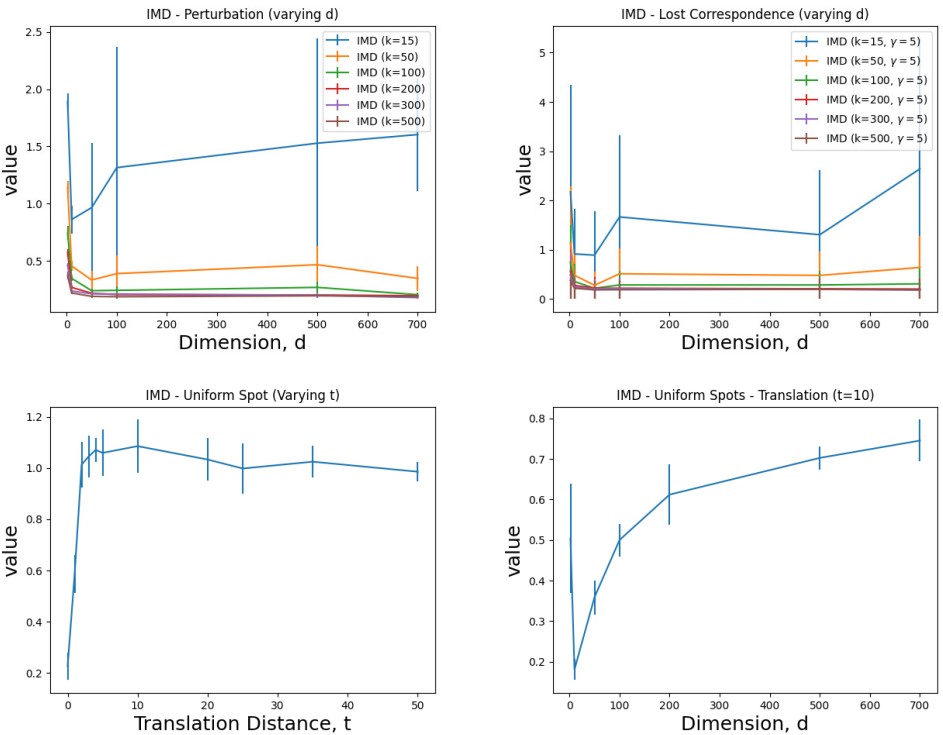

Figure 11: IMD values for a few experiments from Fig. 3. These show much larger variance than other method. However, as $k$ is increased the method show a more consistent behavior.

## H  Implementation Details

We used our own implementation of CKA, kCKA, and MKA algorithms. For RTD, IMD, and SVCCA, we used Barannikov et al. (2022)'s implementation of the algorithm following examples from the corresponding GitHub repository[1]. Additionally, Fig. 2 and 6 were also implemented reusing codes from the same repository.

The ReSi benchmark has been implemented from the publicly available repository. The full details of the benchmark are provided in Supplementary Section I.

The ResNet-50 networks have been trained using a standard training procedure (Adam Kingma & Ba (2015) optimizer with learning rate 0.001, 50 epochs, batch size 128, with a step learning rate schedule at epochs 30 and 40 with gamma 0.1).

All experiments were conducted on a workstation equipped with two NVIDIA RTX 4090 GPUs (24 GB memory each), an AMD Ryzen Threadripper 7960X processor with 24 cores, and 256 GB of system RAM.

Codes are attached as supplementary material for the review.

## I  Details of ReSi Benchmark

### I.1  Summary

We use the representational similarity measures benchmark ReSi (Klabunde et al., 2025) to evaluate MKA and compare with many other commonly used measures [2]. We adopt the ReSi benchmark design, which grounds representational similarity either by prediction (tests 1–2) or by design (tests 3–6). In each test, we construct a controlled set of models and compare layer-wise representations on held-out data. ReSi provides the training protocols, datasets, and reference implementations for 24 baseline similarity measures; we add three new variants - MKA, CKA with RBF kernel, and CKA with RBF kernel and k-NN. The benchmark evaluates measures per test/dataset/model and reports rank- and decision-based metrics accordingly.

Some measures, including PWCCA, Uniformity Difference, and Second-Order Cosine Similarity, are left blank in the result tables, and results for test 5 in the vision domain are also missing. These omissions arise from issues such as numerical instability, the occurrence of negative eigenvalues, prohibitively high runtime, or cases where the measures collapse to identical similarity values across comparisons.

### I.2  Datasets

#### I.2.1  Vision

ImageNet-100 is a balanced subset of 100 classes sampled from the full ImageNet-1k dataset (Russakovsky et al., 2015). The images are resized and center-cropped to 224×224 for both CNNs and ViTs.

#### I.2.2  Language

MNLI (Williams et al., 2018) is a large-scale natural language inference dataset with three labels: entailment, contradiction, and neutral. It consists of premise–hypothesis pairs sampled from ten text genres. We fine-tune BERT and ALBERT on MNLI and evaluate representations exclusively on the validation-matched split.

#### I.2.3  Graphs

For graph representation similarity tests, we use node classification datasets with fixed splits:

**Cora**  A citation network of 2,708 machine learning publications categorized into 7 classes. Each node represents a paper, edges denote citations, and input features are 1,433-dimensional bag-of-words vectors. (Yang et al., 2016)

---

[1] https://github.com/IlyaTrofimov/RTD
[2] https://github.com/mklabunde/resi

**Flickr**   A social network dataset where the nodes represent users, edges represent follow relationships, and node features are 500-dimensional vectors derived from user metadata. The classification task has 7 labels. We subsample 10,000 test nodes for representation extraction. (Zeng et al., 2020)

**OGBN-Arxiv**   A large-scale citation network from the Open Graph Benchmark (OGB). Nodes represent 169k CS papers, edges are citation links, and each node has a 128-dimensional feature vector. The classification task involves 40 subject areas. We subsample 10,000 nodes from the test split for representation extraction. (Hu et al., 2020)

### I.3   Models

To ensure broad coverage of architectural families, we adopt representative models from vision, language, and graph domains. All models are trained or fine-tuned under standardized protocols following the ReSi benchmark, and their hidden representations are extracted in a consistent manner for similarity evaluation.

#### I.3.1   Vision

We employ three canonical CNN families and a transformer-based architecture family. These four families allow us to test whether similarity measures generalize across convolutional, residual, and attention-based architectures.

**ResNets**   ResNet-18, ResNet-34, ResNet-101, trained from scratch on IN100 using cross-entropy loss and SGD with momentum. These models capture hierarchical convolutional features with residual connections. (He et al., 2016)

**VGGs**   VGG-11 and VGG-19 were trained from scratch under identical optimization schedules. Compared to ResNets, VGGs lack skip connections, providing a useful contrast in representational geometry. (Simonyan & Zisserman, 2015)

**Vision Transformers (ViTs)**   ViT-B/32 and ViT-L/32, initialized from ImageNet-21k pretraining and fine-tuned on IN100. Inputs are tokenized into $32 \times 32$ image patches with learnable positional embeddings. (Dosovitskiy et al., 2021)

#### I.3.2   Language

We fine-tune two Transformer encoder models on MNLI, and both models are evaluated on the validation-matched split of MNLI.

**BERT (base)**   Pre-trained BERT is fine-tuned with a linear learning rate schedule, 10% warm-up, and maximum learning rate $5 \times 10^{-5}$. (Devlin et al., 2019)

**ALBERT**   A parameter-reduced variant of BERT using factorized embeddings and cross-layer parameter sharing. Fine-tuning follows the same hyperparameter schedule as BERT. (Lan et al., 2020)

#### I.3.3   Graph

We use graph neural networks (GNNs) implemented in PyTorch Geometric, covering spectral, spatial, and attention-based designs.

**Graph Convolutional Network (GCN)**   A spectral GNN where each layer propagates node features by normalized adjacency matrix multiplication. We train GCNs with two hidden layers for most tests, and extend to five hidden layers for the Layer Monotonicity test to ensure sufficient depth. (Kipf & Welling, 2017)

**GraphSAGE**   A neighborhood-aggregation GNN that samples and aggregates neighbor features using mean aggregation. This model tests inductive generalization properties on large graphs such as Flickr and OGBN-Arxiv. (Hamilton et al., 2017)

**Graph Attention Network (GAT)**   A spatial GNN that computes attention coefficients over neighbors to weigh their contributions. We employ the standard configuration with 8 attention heads. (Veličković et al., 2018)

**Position-aware GNN (P-GNN)**   A positional-encoding GNN that incorporates relative distance features. Due to computational constraints, P-GNN is evaluated only on Cora and excluded from augmentation tests because DropEdge perturbations are incompatible with its positional encodings. (You et al., 2019)

### I.3.4   Representation Extraction

For all models across domains, we extract hidden representations in a standardized manner to ensure comparability of similarity measures. Unless otherwise required by a specific test (e.g., test 6: Layer Monotonicity), we always use the last hidden layer before the classifier head. For CNNs (ResNet, VGG), we take the post-global average pooling (GAP) feature vectors, and in the monotonicity test, we also extract intermediate convolutional blocks, with feature maps downsampled to a uniform $7 \times 7$ spatial resolution for memory control. For Vision Transformers, we use the [CLS] token from the final transformer block as the representation. For language models (BERT and ALBERT), we primarily use the final-layer [CLS] token embedding to represent each premise–hypothesis pair, while also including mean-pooled token embeddings as an alternative variant. For graph neural networks (GCN, GraphSAGE, GAT, P-GNN), we extract node embeddings from the last hidden layer, and in the monotonicity test, we additionally collect outputs from all intermediate layers. All representations are computed exclusively on held-out validation or test splits (IN100 validation set with 50 images per class, MNLI validation-matched set, and the test nodes of Cora/Flickr/OGBN-Arxiv) to prevent training leakage and to keep sample sizes fixed across similarity measures.

### I.4   Tests

### I.4.1   Test 1 — Correlation to Accuracy Difference

If two models differ in accuracy, their representations should differ accordingly. We train ten models per dataset, varying only random seeds, compute accuracies on the test split, and correlate pairwise representational similarity with the absolute accuracy difference.

### I.4.2   Test 2 — Correlation to Output Difference

Models with similar accuracy can still produce different instance-level predictions; we correlate representational similarity with (i) disagreement rate between hard labels and (ii) the mean Jensen–Shannon divergence (JSD) between probability vectors.

### I.4.3   Test 3 — Label Randomization

Distinguish models trained with different degrees of label corruption. Groups are defined by randomization rate (e.g., 0%, 25%, 50%, 75%, 100%), with five models per group. We then test if within-group similarities exceed between-group similarities.

### I.4.4   Test 4 — Shortcut Affinity

Detect reliance on artificial shortcut features. We add synthetic label-leaking features during training and form groups by shortcut "strength." Each group consists of five independently trained models with different random seeds. A good similarity measure should assign higher similarity within groups of models trained on shortcuts of the same strength than across groups trained with different strengths.

### I.4.5 Test 5 — Augmentation

Assess whether measures capture robustness to data augmentations. We train one "reference" group on standard data and additional groups with progressively stronger augmentation, but always evaluate on non-augmented test data. Each group consists of five independently trained models with different random seeds. It is expected that models of the same group should have more similarity than those trained on differently augmented data.

### I.4.6 Test 6 — Layer Monotonicity

Within a single model, nearby layers should be more similar than distant ones; we check whether similarity decreases with layer distance, and whether ordered pair constraints hold. We use the models from Tests 1–2 (for graphs, we increase the inner layers to five). We extract multiple intermediate layers and then compute (a) conformity to the ordinal constraints and (b) Spearman correlation between similarity and layer distance.

## I.5 Representational Similarity Measures

### I.5.1 Baseline Measures (from ReSi)

ReSi covers 24 measures spanning alignment/CCA-type scores, RSM-based distances, topology-based divergences, neighborhood statistics, and simple statistics; we use their official implementations and hyperparameters.

**CCA-based measures**
PWCCA — Projection-Weighted Canonical Correlation Analysis (Morcos et al., 2018)
SVCCA — Singular Vector Canonical Correlation Analysis (Raghu et al., 2017)

**Alignment-based measures**
AlignCos — Aligned Cosine Similarity (Hamilton et al., 2016b)
AngShape — Orthogonal Angular Shape Metric (Williams et al., 2021)
HardCorr — Hard Correlation Match (Li et al., 2016)
LinReg — Linear Regression Alignment (Kornblith et al., 2019)
OrthProc — Orthogonal Procrustes (Ding et al., 2021)
PermProc — Permutation Procrustes (Williams et al., 2021)
ProcDist — Procrustes Size-and-Shape Distance (Williams et al., 2021)
SoftCorr — Soft Correlation Match (Li et al., 2016)

**RSM-based measures**
CKA — Centered Kernel Alignment (Kornblith et al., 2019)
DistCorr — Distance Correlation (Székely et al., 2007)
EOS — Eigenspace Overlap Score (May et al., 2019)
GULP — Generalized Unsupervised Linear Prediction (Boix-Adsera et al., 2022)
RSA — Representational Similarity Analysis (Kriegeskorte et al., 2008)
RSMDiff — RSM Norm Difference (Yin & Shen, 2018)

**Neighbor-based measures**
2nd-Cos — Second-order Cosine Similarity (Hamilton et al., 2016a)
Jaccard — k-NN Jaccard Similarity (Wang et al., 2020)
RankSim — Rank Similarity (Wang et al., 2020)

**Topology-based measures**
IMD — Intrinsic Manifold Distance Tsitsulin et al. (2020)
RTD — Representation Topology Divergence Barannikov et al. (2022)

**Statistic-based measures**
ConcDiff — Concentricity Difference (Wang et al., 2020)
MagDiff — Magnitude Difference (Wang et al., 2020)
UnifDiff — Uniformity Difference (Wang & Isola, 2020)

### I.5.2 Additional Measures

In addition to the 24 baseline measures in the ReSi benchmark, we implemented three new kernel-based alignment variants (MKA, CKA with RBF kernel, and CKA with RBF kernel and k-NN).

**Manifold Approximated Kernel Alignment (MKA)** In our implementation, we evaluate MKA under four neighborhood sizes, namely $k = 15, 50, 100, 200$. These values allow us to probe the trade-off between local geometry (small $k$) and more global manifold structure (large $k$).

**CKA with RBF Kernel and $k$-Nearest Neighbors (kCKA)** In addition to the dense RBF kernel, we also evaluate a sparsified version that restricts non-zero entries to a fixed number of nearest neighbors. Given a representation set $X = \{x_1, \ldots, x_N\}$, we compute pairwise Euclidean distances $d(x_i, x_j) = \|x_i - x_j\|_2$. For each point $x_i$, we retain only its $k$ nearest neighbors, denoted $\text{KNN}(x_i, k)$. The sparsified RBF kernel matrix is then defined as

$$K_{ij} = \begin{cases} \exp\left(-\dfrac{d(x_i, x_j)}{2\sigma}\right), & \text{if } x_j \in \text{KNN}(x_i, k), \\ 0, & \text{otherwise,} \end{cases} \tag{49}$$

where the bandwidth parameter $\sigma$ is chosen as the median distance among all retained neighbor pairs.

The final CKA score between two representation sets $X$ and $Y$, with sparsified RBF kernels $K$ and $L$, is computed in the same way as standard CKA using the normalized HSIC formulation:

$$\text{CKA}(K, L) = \frac{\langle KH, LH \rangle}{\sqrt{\langle KH, KH \rangle \langle LH, LH \rangle}}, \tag{50}$$

where $H = I - \frac{1}{N}\mathbf{1}\mathbf{1}^\top$ is the centering matrix. In our experiments, we set the neighborhood size to $k = 100$, so that each instance is only connected to its 100 nearest neighbors in the kernel matrix.

## J ReSI Benchmark Scores

### J.1 Vision Task

Table 1: Results of Test 1 (Correlation to Accuracy Difference) for the vision domain on ImageNet-100

| Test
Dataset
Architecture | Accuracy Correlation
IN100 | | | | | | |
|---|---|---|---|---|---|---|---|
| | RNet18 | RNet34 | RNet101 | VGG11 | VGG19 | ViT B32 | ViT L32 |
| CKA | **0.33** | -0.09 | 0.08 | 0.02 | -0.22 | -0.23 | 0.02 |
| CKA ($\delta = 0.45$) | 0.29 | -0.06 | 0.01 | 0.00 | -0.10 | -0.28 | 0.09 |
| CKA ($\delta = 0.2$) | 0.02 | 0.19 | -0.08 | -0.14 | -0.23 | -0.15 | 0.02 |
| kCKA ($k = 100$) | 0.00 | -0.01 | **0.15** | 0.09 | **0.58** | -0.16 | **0.10** |
| SVCCA | 0.29 | **0.27** | 0.00 | -0.04 | -0.30 | -0.01 | -0.17 |
| RTD | 0.26 | -0.01 | -0.20 | **0.15** | 0.07 | -0.10 | **0.10** |
| IMD | 0.17 | -0.20 | 0.12 | 0.06 | -0.11 | -0.26 | -0.16 |
| MKA ($k = 100$) | 0.17 | -0.14 | -0.10 | -0.03 | -0.04 | **0.26** | 0.09 |
| CKA (linear) | 0.36 | -0.07 | 0.16 | 0.03 | -0.20 | -0.26 | 0.05 |
| MKA ($k = 15$) | 0.15 | -0.23 | -0.10 | -0.03 | -0.08 | 0.24 | 0.16 |
| MKA ($k = 50$) | 0.17 | -0.16 | -0.10 | -0.04 | -0.05 | 0.26 | 0.11 |
| MKA ($k = 200$) | 0.16 | -0.14 | -0.11 | -0.04 | -0.01 | 0.26 | 0.08 |
| AlignedCosineSimilarity | -0.08 | -0.35 | -0.01 | -0.13 | -0.12 | 0.07 | 0.05 |
| ConcentricityDifference | -0.11 | 0.34 | -0.04 | -0.11 | -0.13 | 0.00 | 0.18 |
| DistanceCorrelation | 0.31 | -0.08 | 0.08 | 0.03 | -0.21 | -0.26 | 0.03 |
| EigenspaceOverlapScore | 0.05 | -0.17 | 0.11 | -0.22 | 0.08 | 0.47 | 0.03 |
| Gulp | 0.02 | -0.18 | 0.12 | -0.17 | 0.10 | 0.28 | 0.04 |
| HardCorrelationMatch | 0.21 | 0.13 | -0.01 | -0.01 | -0.03 | 0.35 | -0.17 |
| JaccardSimilarity | -0.11 | -0.13 | -0.06 | -0.22 | 0.06 | -0.02 | 0.26 |
| LinearRegression | 0.19 | -0.11 | 0.09 | -0.04 | 0.09 | -0.01 | 0.05 |
| MagnitudeDifference | -0.16 | 0.02 | -0.08 | -0.07 | -0.12 | 0.07 | 0.15 |
| OrthogonalAngularShapeMetricCentered | 0.21 | -0.16 | 0.15 | -0.02 | 0.03 | 0.07 | 0.06 |
| OrthogonalProcrustesCenteredAndNormalized | 0.21 | -0.16 | 0.15 | -0.02 | 0.03 | 0.07 | 0.06 |
| PermutationProcrustes | 0.07 | 0.09 | 0.08 | 0.14 | -0.02 | -0.06 | -0.33 |
| ProcrustesSizeAndShapeDistance | 0.08 | 0.00 | 0.14 | 0.13 | 0.08 | 0.16 | 0.05 |
| RSA | 0.06 | -0.17 | 0.09 | 0.24 | -0.35 | -0.12 | -0.11 |
| RSMNormDifference | 0.09 | -0.10 | 0.11 | -0.04 | -0.08 | 0.01 | -0.06 |
| RankSimilarity | 0.09 | 0.03 | 0.13 | -0.01 | 0.05 | 0.18 | 0.36 |
| SecondOrderCosineSimilarity | -0.08 | -0.15 | 0.05 | -0.20 | -0.18 | -0.22 | 0.17 |
| SoftCorrelationMatch | 0.27 | 0.08 | 0.04 | -0.03 | -0.10 | 0.36 | -0.19 |
| UniformityDifference | -0.18 | —— | -0.02 | 0.17 | -0.04 | —— | —— |

Table 2: Results of Test 2 (Correlation to Output Difference) for the vision domain on ImageNet-100

| Test Dataset Architecture | JSD Correlation IN100 | | | | | | | Disagreement Correlation IN100 | | | | | | |
|---|---|---|---|---|---|---|---|---|---|---|---|---|---|---|
| | RNet18 | RNet34 | RNet101 | VGG11 | VGG19 | ViT B32 | ViT L32 | RNet18 | RNet34 | RNet101 | VGG11 | VGG19 | ViT B32 | ViT L32 |
| CKA | **0.26** | 0.02 | 0.30 | -0.09 | 0.04 | 0.02 | -0.12 | **0.36** | 0.00 | 0.29 | -0.01 | -0.25 | 0.00 | -0.05 |
| CKA ($\delta = 0.45$) | 0.20 | -0.13 | 0.30 | -0.08 | 0.19 | 0.03 | -0.21 | 0.33 | -0.14 | 0.35 | -0.00 | -0.21 | 0.04 | -0.16 |
| CKA ($\delta = 0.2$) | 0.07 | -0.03 | -0.26 | 0.02 | -0.13 | 0.32 | -0.61 | 0.35 | -0.18 | 0.08 | **0.39** | 0.13 | 0.35 | -0.48 |
| kCKA ($k = 100$) | -0.34 | **0.35** | **0.43** | -0.02 | **0.24** | 0.32 | -0.32 | -0.56 | **0.45** | 0.18 | -0.18 | -0.17 | 0.22 | -0.17 |
| SVCCA | 0.21 | -0.00 | 0.25 | -0.11 | 0.16 | 0.05 | 0.18 | 0.39 | 0.07 | 0.15 | 0.03 | 0.01 | -0.06 | 0.07 |
| RTD | -0.13 | -0.06 | 0.23 | -0.03 | -0.11 | 0.08 | -0.13 | 0.03 | -0.20 | 0.43 | -0.16 | -0.28 | -0.00 | -0.17 |
| IMD | -0.10 | 0.02 | 0.20 | **0.21** | 0.09 | **0.43** | **0.05** | -0.03 | -0.06 | 0.24 | 0.30 | -0.02 | 0.25 | **0.07** |
| MKA ($k = 100$) | 0.11 | 0.25 | 0.39 | 0.08 | 0.22 | 0.39 | -0.25 | 0.09 | 0.08 | 0.30 | 0.04 | -0.17 | 0.27 | -0.13 |
| CKA (linear) | 0.30 | 0.08 | 0.30 | -0.13 | -0.06 | 0.04 | -0.07 | 0.37 | 0.08 | 0.24 | 0.01 | -0.24 | 0.00 | -0.02 |
| MKA ($k = 15$) | 0.17 | 0.12 | 0.35 | 0.12 | 0.24 | 0.38 | -0.28 | 0.11 | -0.01 | 0.37 | 0.10 | -0.15 | 0.41 | -0.22 |
| MKA ($k = 50$) | 0.13 | 0.23 | 0.39 | 0.08 | 0.22 | 0.38 | -0.25 | 0.13 | 0.06 | 0.41 | 0.07 | -0.20 | 0.39 | -0.21 |
| MKA ($k = 200$) | 0.11 | 0.25 | 0.39 | 0.08 | 0.22 | 0.39 | -0.24 | 0.12 | 0.07 | 0.44 | 0.08 | -0.22 | 0.39 | -0.20 |
| AlignedCosineSimilarity | 0.08 | 0.05 | 0.38 | 0.10 | -0.20 | -0.22 | -0.06 | 0.20 | 0.50 | 0.17 | 0.16 | -0.13 | -0.08 | 0.00 |
| ConcentricityDifference | -0.29 | 0.24 | -0.11 | -0.17 | -0.13 | -0.11 | -0.37 | -0.08 | 0.00 | -0.20 | -0.11 | -0.06 | -0.08 | -0.29 |
| DistanceCorrelation | 0.26 | 0.05 | 0.31 | -0.10 | 0.04 | 0.05 | -0.12 | 0.36 | 0.01 | 0.30 | -0.00 | -0.25 | 0.02 | -0.05 |
| EigenspaceOverlapScore | 0.09 | 0.49 | 0.33 | -0.11 | 0.15 | -0.18 | -0.28 | 0.11 | 0.25 | 0.15 | -0.31 | -0.41 | 0.01 | -0.15 |
| Gulp | 0.07 | 0.49 | 0.35 | -0.05 | 0.15 | -0.09 | -0.28 | 0.09 | 0.27 | 0.13 | -0.27 | -0.41 | 0.07 | -0.15 |
| HardCorrelationMatch | 0.28 | 0.31 | 0.02 | -0.22 | -0.05 | 0.03 | -0.26 | 0.28 | -0.06 | -0.07 | -0.15 | -0.18 | 0.27 | -0.16 |
| JaccardSimilarity | 0.35 | 0.26 | 0.31 | 0.04 | 0.32 | 0.46 | -0.30 | 0.25 | 0.47 | 0.25 | 0.14 | -0.12 | 0.33 | -0.18 |
| LinearRegression | 0.21 | 0.21 | 0.41 | -0.01 | 0.25 | -0.09 | -0.14 | 0.19 | 0.25 | 0.30 | -0.17 | -0.23 | 0.04 | -0.07 |
| MagnitudeDifference | -0.38 | -0.20 | 0.01 | -0.16 | -0.28 | 0.02 | -0.32 | -0.17 | -0.22 | -0.04 | -0.09 | 0.04 | -0.01 | -0.22 |
| OrthogonalAngularShapeMetricCentered | 0.24 | 0.22 | 0.34 | -0.01 | 0.19 | -0.26 | -0.15 | 0.24 | 0.40 | 0.20 | -0.13 | -0.33 | -0.11 | -0.06 |
| OrthogonalProcrustesCenteredAndNormalized | 0.24 | 0.22 | 0.34 | -0.02 | 0.19 | -0.26 | -0.15 | 0.24 | 0.40 | 0.20 | -0.13 | -0.33 | -0.11 | -0.06 |
| PermutationProcrustes | 0.18 | 0.18 | 0.27 | -0.18 | 0.06 | 0.36 | -0.06 | 0.13 | -0.25 | -0.04 | 0.02 | 0.20 | 0.37 | 0.10 |
| ProcrustesSizeAndShapeDistance | 0.10 | 0.14 | 0.39 | -0.05 | 0.27 | -0.05 | 0.02 | 0.08 | -0.08 | 0.11 | -0.10 | -0.07 | -0.07 | 0.08 |
| RSA | 0.12 | 0.18 | 0.09 | -0.18 | -0.19 | -0.11 | -0.20 | 0.19 | 0.33 | 0.11 | -0.05 | 0.04 | 0.00 | -0.04 |
| RSMNormDifference | -0.41 | -0.22 | 0.30 | -0.27 | 0.07 | 0.02 | -0.28 | -0.18 | -0.20 | 0.19 | -0.01 | -0.03 | -0.21 | -0.17 |
| RankSimilarity | -0.13 | -0.01 | 0.24 | 0.03 | 0.05 | 0.25 | -0.09 | -0.09 | -0.04 | 0.05 | 0.03 | -0.34 | 0.15 | -0.30 |
| SecondOrderCosineSimilarity | -0.13 | 0.16 | 0.28 | 0.07 | -0.29 | 0.43 | -0.35 | -0.20 | 0.45 | 0.11 | 0.11 | -0.07 | 0.19 | -0.27 |
| SoftCorrelationMatch | 0.45 | 0.27 | 0.11 | -0.04 | -0.17 | 0.01 | -0.31 | 0.46 | -0.13 | -0.06 | -0.03 | -0.29 | 0.27 | -0.16 |
| UniformityDifference | -0.34 | —— | -0.40 | 0.04 | -0.17 | —— | -0.01 | —— | -0.27 | 0.17 | 0.39 | —— | —— | —— |

Table 3: Results of Test 3 (Label Randomization) for the vision domain on ImageNet-100

| Evaluation Dataset Architecture | AUPRC IN100 | | | | | | | Conformity Rate IN100 | | | | | | |
|---|---|---|---|---|---|---|---|---|---|---|---|---|---|---|
| | RNet18 | RNet34 | RNet101 | VGG11 | VGG19 | ViT B32 | ViT L32 | RNet18 | RNet34 | RNet101 | VGG11 | VGG19 | ViT B32 | ViT L32 |
| CKA | **1.00** | **1.00** | **1.00** | **1.00** | 0.77 | 0.52 | 0.81 | **1.00** | **1.00** | **1.00** | **1.00** | 0.77 | 0.65 | 0.83 |
| CKA ($\delta = 0.45$) | **1.00** | **1.00** | **1.00** | **1.00** | 0.77 | 0.78 | 0.86 | **1.00** | **1.00** | **1.00** | **1.00** | 0.75 | 0.81 | 0.87 |
| CKA ($\delta = 0.2$) | 0.65 | 0.70 | 0.82 | 0.80 | 0.76 | 0.77 | **0.97** | 0.71 | 0.74 | 0.87 | 0.84 | 0.73 | 0.75 | **0.99** |
| kCKA ($k = 100$) | **1.00** | **1.00** | **1.00** | 0.88 | 0.73 | 0.57 | 0.90 | **1.00** | **1.00** | **1.00** | 0.95 | 0.70 | 0.73 | 0.97 |
| SVCCA | 1.00 | 0.94 | 0.95 | 0.90 | 0.78 | 0.52 | 0.58 | 1.00 | 0.97 | 0.96 | 0.89 | 0.89 | 0.74 | 0.77 |
| RTD | **1.00** | **1.00** | **1.00** | **1.00** | **1.00** | **0.95** | 0.44 | **1.00** | **1.00** | **1.00** | **1.00** | **1.00** | **0.95** | 0.87 |
| IMD | **1.00** | **1.00** | **1.00** | **1.00** | **1.00** | 0.73 | 0.80 | **1.00** | **1.00** | **1.00** | **1.00** | **1.00** | 0.78 | 0.80 |
| MKA ($k = 100$) | **1.00** | **1.00** | **1.00** | 0.94 | 0.73 | 0.75 | 0.80 | **1.00** | **1.00** | **1.00** | 0.96 | 0.70 | 0.81 | 0.79 |
| CKA (linear) | 1.00 | 1.00 | 1.00 | 1.00 | 0.77 | 0.57 | 0.89 | 1.00 | 1.00 | 1.00 | 1.00 | 0.75 | 0.73 | 0.89 |
| MKA ($k = 15$) | 1.00 | 1.00 | 1.00 | 0.98 | 0.73 | 0.76 | 0.81 | 1.00 | 1.00 | 1.00 | 0.98 | 0.70 | 0.82 | 0.86 |
| MKA ($k = 50$) | 1.00 | 1.00 | 1.00 | 0.97 | 0.73 | 0.75 | 0.80 | 1.00 | 1.00 | 1.00 | 0.98 | 0.70 | 0.81 | 0.79 |
| MKA ($k = 200$) | 1.00 | 1.00 | 1.00 | 0.92 | 0.70 | 0.75 | 0.80 | 1.00 | 1.00 | 1.00 | 0.95 | 0.69 | 0.81 | 0.78 |
| AlignedCosineSimilarity | 0.72 | 0.72 | 0.85 | 0.72 | 0.46 | 0.58 | 1.00 | 0.83 | 0.83 | 0.94 | 0.83 | 0.55 | 0.75 | 1.00 |
| ConcentricityDifference | 0.99 | 1.00 | 1.00 | 1.00 | 0.57 | 0.73 | 1.00 | 1.00 | 1.00 | 1.00 | 1.00 | 0.84 | 0.90 | 1.00 |
| DistanceCorrelation | 1.00 | 1.00 | 1.00 | 1.00 | 0.77 | 0.57 | 0.90 | 1.00 | 1.00 | 1.00 | 1.00 | 0.77 | 0.76 | 0.90 |
| EigenspaceOverlapScore | 0.84 | 0.72 | 0.70 | 0.55 | 0.50 | 0.62 | 0.70 | 0.95 | 0.83 | 0.73 | 0.75 | 0.59 | 0.79 | 0.75 |
| Gulp | 0.89 | 0.72 | 0.60 | 0.88 | 0.44 | 0.53 | 0.66 | 0.97 | 0.84 | 0.91 | 0.96 | 0.64 | 0.85 | 0.93 |
| HardCorrelationMatch | 0.72 | 0.72 | 0.72 | 1.00 | 0.60 | 0.53 | 0.91 | 0.83 | 0.83 | 0.83 | 1.00 | 0.67 | 0.68 | 0.94 |
| JaccardSimilarity | 1.00 | 1.00 | 1.00 | 1.00 | 0.73 | 0.49 | 0.83 | 1.00 | 1.00 | 1.00 | 1.00 | 0.70 | 0.69 | 0.87 |
| MagnitudeDifference | 1.00 | 1.00 | 1.00 | 1.00 | 1.00 | 1.00 | 0.71 | 1.00 | 1.00 | 1.00 | 1.00 | 1.00 | 1.00 | 0.82 |
| OrthogonalAngularShapeMetricCentered | 0.72 | 0.72 | 1.00 | 1.00 | 0.80 | 0.55 | 0.91 | 0.83 | 0.83 | 1.00 | 1.00 | 0.82 | 0.70 | 0.94 |
| OrthogonalProcrustesCenteredAndNormalized | 0.72 | 0.72 | 1.00 | 1.00 | 0.80 | 0.55 | 0.91 | 0.83 | 0.83 | 1.00 | 1.00 | 0.82 | 0.70 | 0.94 |
| PermutationProcrustes | 0.70 | 0.70 | 0.71 | 1.00 | 0.72 | 0.42 | 0.70 | 0.67 | 0.67 | 0.75 | 1.00 | 0.83 | 0.50 | 0.78 |
| ProcrustesSizeAndShapeDistance | 0.72 | 0.71 | 0.73 | 1.00 | 0.72 | 0.70 | 0.75 | 0.83 | 0.76 | 0.85 | 1.00 | 0.83 | 0.67 | 0.77 |
| RSA | 0.75 | 0.72 | 1.00 | 1.00 | 0.76 | 0.49 | 0.86 | 0.89 | 0.83 | 1.00 | 1.00 | 0.75 | 0.63 | 0.86 |
| RSMNormDifference | 1.00 | 1.00 | 1.00 | 1.00 | 1.00 | 1.00 | 0.75 | 1.00 | 1.00 | 1.00 | 1.00 | 1.00 | 1.00 | 0.78 |
| RankSimilarity | 1.00 | 1.00 | 1.00 | 0.99 | 0.73 | 0.74 | 0.77 | 1.00 | 1.00 | 1.00 | 1.00 | 0.72 | 0.78 | 0.85 |
| SecondOrderCosineSimilarity | 0.71 | 0.71 | 0.71 | 0.70 | 0.70 | 0.70 | 0.69 | 0.79 | 0.75 | 0.74 | 0.67 | 0.67 | 0.75 | 0.67 |
| SoftCorrelationMatch | 0.72 | 0.72 | 0.72 | 0.85 | 0.46 | 0.52 | 0.91 | 0.83 | 0.83 | 0.83 | 0.96 | 0.56 | 0.62 | 0.94 |
| UniformityDifference | 0.42 | 0.42 | 0.53 | 0.75 | 0.57 | 0.52 | 0.20 | 0.62 | 0.57 | 0.72 | 0.90 | 0.81 | 0.67 | 0.32 |

Table 4: Results of Test 4 (Shortcut Affinity) for the vision domain on ImageNet-100

| Evaluation Dataset Architecture | AUPRC IN100 | | | | | | | Conformity Rate IN100 | | | | | | |
|---|---|---|---|---|---|---|---|---|---|---|---|---|---|---|
| | RNet18 | RNet34 | RNet101 | VGG11 | VGG19 | ViT B32 | ViT L32 | RNet18 | RNet34 | RNet101 | VGG11 | VGG19 | ViT B32 | ViT L32 |
| CKA | **1.00** | **1.00** | **1.00** | **1.00** | **1.00** | **1.00** | 0.86 | **1.00** | **1.00** | **1.00** | **1.00** | **1.00** | **1.00** | 0.96 |
| CKA ($\delta = 0.45$) | **1.00** | **1.00** | **1.00** | **1.00** | **1.00** | **1.00** | 0.90 | **1.00** | **1.00** | **1.00** | **1.00** | **1.00** | **1.00** | 0.97 |
| CKA ($\delta = 0.2$) | 0.77 | 0.62 | 0.66 | 0.80 | 0.74 | **1.00** | 0.79 | 0.94 | 0.85 | 0.88 | 0.93 | 0.94 | **1.00** | 0.93 |
| kCKA ($k = 100$) | 0.94 | 0.92 | 0.98 | **1.00** | **1.00** | **1.00** | **0.91** | 0.99 | 0.98 | **1.00** | **1.00** | **1.00** | **1.00** | 0.97 |
| SVCCA | 0.55 | 0.68 | 0.51 | 0.68 | 0.29 | 0.60 | 0.28 | 0.81 | 0.84 | 0.81 | 0.91 | 0.62 | 0.82 | 0.57 |
| RTD | **1.00** | **1.00** | **1.00** | **1.00** | **1.00** | 0.72 | 0.84 | **1.00** | **1.00** | **1.00** | **1.00** | **1.00** | 0.92 | 0.95 |
| IMD | 0.66 | 0.77 | 0.56 | 0.78 | 0.67 | 0.38 | 0.31 | 0.87 | 0.88 | 0.75 | 0.92 | 0.77 | 0.74 | 0.66 |
| MKA ($k = 100$) | **1.00** | **1.00** | **1.00** | **1.00** | **1.00** | **1.00** | **0.91** | **1.00** | **1.00** | **1.00** | **1.00** | **1.00** | **1.00** | **0.98** |
| CKA (linear) | 1.00 | 1.00 | 1.00 | 1.00 | 1.00 | 1.00 | 0.87 | 1.00 | 1.00 | 1.00 | 1.00 | 1.00 | 1.00 | 0.96 |
| MKA ($k = 15$) | 1.00 | 1.00 | 1.00 | 1.00 | 1.00 | 1.00 | 0.92 | 1.00 | 1.00 | 1.00 | 1.00 | 1.00 | 1.00 | 0.98 |
| MKA ($k = 50$) | 1.00 | 1.00 | 1.00 | 1.00 | 1.00 | 1.00 | 0.90 | 1.00 | 1.00 | 1.00 | 1.00 | 1.00 | 1.00 | 0.97 |
| MKA ($k = 200$) | 1.00 | 1.00 | 1.00 | 1.00 | 1.00 | 1.00 | 0.91 | 1.00 | 1.00 | 1.00 | 1.00 | 1.00 | 1.00 | 0.98 |
| AlignedCosineSimilarity | 1.00 | 1.00 | 1.00 | 1.00 | 1.00 | 1.00 | 0.99 | 1.00 | 1.00 | 1.00 | 1.00 | 1.00 | 1.00 | 1.00 |
| ConcentricityDifference | 0.53 | 0.70 | 0.50 | 0.78 | 0.27 | 0.28 | 0.25 | 0.83 | 0.86 | 0.81 | 0.95 | 0.67 | 0.58 | 0.64 |
| DistanceCorrelation | 1.00 | 1.00 | 1.00 | 1.00 | 1.00 | 1.00 | 0.88 | 1.00 | 1.00 | 1.00 | 1.00 | 1.00 | 1.00 | 0.96 |
| EigenspaceOverlapScore | 1.00 | 1.00 | 0.99 | 0.95 | 0.88 | 0.93 | 0.93 | 1.00 | 1.00 | 1.00 | 0.98 | 0.96 | 0.98 | 0.97 |
| Gulp | 1.00 | 1.00 | 1.00 | 0.96 | 0.88 | 0.97 | 0.93 | 1.00 | 1.00 | 1.00 | 0.98 | 0.96 | 1.00 | 0.97 |
| HardCorrelationMatch | 0.97 | 1.00 | 0.99 | 0.92 | 0.91 | 0.97 | 0.90 | 0.99 | 1.00 | 1.00 | 0.97 | 0.98 | 0.99 | 0.98 |
| JaccardSimilarity | 1.00 | 1.00 | 1.00 | 1.00 | 1.00 | 1.00 | 0.90 | 1.00 | 1.00 | 1.00 | 1.00 | 1.00 | 1.00 | 0.98 |
| LinearRegression | 1.00 | 1.00 | 1.00 | 1.00 | 0.98 | 0.52 | 0.92 | 1.00 | 1.00 | 1.00 | 1.00 | 0.99 | 0.83 | 0.98 |
| MagnitudeDifference | 0.37 | 0.37 | 0.44 | 0.53 | 0.23 | 0.23 | 0.47 | 0.62 | 0.75 | 0.79 | 0.85 | 0.51 | 0.57 | 0.79 |
| OrthogonalAngularShapeMetricCentered | 1.00 | 1.00 | 1.00 | 1.00 | 1.00 | 1.00 | 1.00 | 1.00 | 1.00 | 1.00 | 1.00 | 1.00 | 1.00 | 1.00 |
| OrthogonalProcrustesCenteredAndNormalized | 1.00 | 1.00 | 1.00 | 1.00 | 1.00 | 1.00 | 1.00 | 1.00 | 1.00 | 1.00 | 1.00 | 1.00 | 1.00 | 1.00 |
| PermutationProcrustes | 0.72 | 0.80 | 0.94 | 0.66 | 0.82 | 0.97 | 0.77 | 0.89 | 0.94 | 1.00 | 0.87 | 0.93 | 0.98 | 0.89 |
| ProcrustesSizeAndShapeDistance | 1.00 | 1.00 | 1.00 | 1.00 | 1.00 | 1.00 | 0.89 | 1.00 | 1.00 | 1.00 | 1.00 | 1.00 | 1.00 | 0.95 |
| RSA | 1.00 | 1.00 | 1.00 | 1.00 | 1.00 | 1.00 | 0.72 | 1.00 | 1.00 | 1.00 | 1.00 | 1.00 | 1.00 | 0.91 |
| RSMNormDifference | 0.57 | 0.42 | 0.59 | 0.87 | 0.59 | 0.50 | 0.47 | 0.81 | 0.71 | 0.83 | 0.97 | 0.80 | 0.82 | 0.69 |
| RankSimilarity | 0.99 | 0.99 | 1.00 | 1.00 | 1.00 | 1.00 | 0.89 | 1.00 | 1.00 | 1.00 | 1.00 | 1.00 | 1.00 | 0.97 |
| SecondOrderCosineSimilarity | 1.00 | 1.00 | 1.00 | 1.00 | 1.00 | 1.00 | 0.92 | 1.00 | 1.00 | 1.00 | 1.00 | 1.00 | 1.00 | 0.97 |
| SoftCorrelationMatch | 0.97 | 0.98 | 0.99 | 0.90 | 0.84 | 0.98 | 0.94 | 0.99 | 1.00 | 1.00 | 0.97 | 0.96 | 1.00 | 0.99 |
| UniformityDifference | 0.75 | 0.73 | 0.87 | 0.60 | 0.55 | 0.61 | 0.17 | 0.90 | 0.91 | 0.97 | 0.83 | 0.83 | 0.84 | 0.00 |

Table 5: Results of Test 6 (Layer Monotonicity) for the vision domain on ImageNet-100

| Evaluation Dataset Architecture | Spearman IN100 | | | | | | | Conformity Rate IN100 | | | | | | |
|---|---|---|---|---|---|---|---|---|---|---|---|---|---|---|
| | RNet18 | RNet34 | RNet101 | VGG11 | VGG19 | ViT B32 | ViT L32 | RNet18 | RNet34 | RNet101 | VGG11 | VGG19 | ViT B32 | ViT L32 |
| CKA | 0.97 | 0.79 | 0.97 | 0.88 | 0.93 | **1.00** | **1.00** | 0.95 | 0.98 | 0.99 | 0.90 | 0.92 | **1.00** | **1.00** |
| CKA ($\delta = 0.45$) | 0.90 | 0.73 | 0.91 | 0.97 | 0.94 | **1.00** | **1.00** | 0.94 | 0.95 | 0.97 | 0.95 | 0.93 | **1.00** | **1.00** |
| CKA ($\delta = 0.2$) | 0.97 | 0.89 | 0.84 | **1.00** | 0.84 | 0.95 | **1.00** | 0.96 | 0.96 | 0.96 | **1.00** | 0.94 | 0.93 | **1.00** |
| kCKA ($k = 100$) | **1.00** | **1.00** | **1.00** | **1.00** | **1.00** | **1.00** | **1.00** | **1.00** | **1.00** | **1.00** | **1.00** | **1.00** | **1.00** | **1.00** |
| SVCCA | 0.20 | 0.27 | 0.43 | 0.42 | 0.40 | 0.87 | 0.61 | 0.72 | 0.58 | 0.72 | 0.75 | 0.69 | 0.86 | 0.78 |
| RTD | 0.97 | 0.83 | 0.44 | 0.52 | 0.84 | 0.90 | **1.00** | **1.00** | 0.92 | 0.83 | 0.83 | 0.96 | 0.93 | **1.00** |
| IMD | -0.01 | 0.23 | 0.07 | -0.03 | 0.09 | 0.58 | 0.37 | 0.51 | 0.66 | 0.62 | 0.48 | 0.50 | 0.86 | 0.54 |
| MKA ($k = 100$) | **1.00** | **1.00** | 0.96 | 0.79 | **1.00** | **1.00** | **1.00** | **1.00** | **1.00** | 0.99 | 0.82 | **1.00** | **1.00** | **1.00** |
| CKA (linear) | 0.87 | 0.82 | 0.97 | 0.88 | 0.93 | 1.00 | 1.00 | 0.94 | 0.98 | 1.00 | 0.90 | 0.92 | 1.00 | 1.00 |
| MKA ($k = 15$) | 1.00 | 1.00 | 0.96 | 0.80 | 1.00 | 1.00 | 1.00 | 1.00 | 1.00 | 0.99 | 0.82 | 1.00 | 1.00 | 1.00 |
| MKA ($k = 50$) | 1.00 | 1.00 | 0.96 | 0.79 | 1.00 | 1.00 | 1.00 | 1.00 | 1.00 | 0.99 | 0.80 | 1.00 | 1.00 | 1.00 |
| MKA ($k = 200$) | 1.00 | 1.00 | 0.96 | 0.79 | 1.00 | 1.00 | 1.00 | 1.00 | 1.00 | 0.99 | 0.82 | 1.00 | 1.00 | 1.00 |
| AlignedCosineSimilarity | 0.52 | 0.63 | 0.52 | 0.93 | 0.12 | 1.00 | 1.00 | 0.84 | 0.86 | 0.80 | 0.90 | 0.68 | 1.00 | 1.00 |
| ConcentricityDifference | -0.78 | -0.05 | -0.14 | -0.25 | -0.27 | 0.65 | 1.00 | 0.20 | 0.46 | 0.46 | 0.32 | 0.43 | 0.90 | 1.00 |
| DistanceCorrelation | 0.97 | 0.79 | 0.97 | 0.88 | 0.93 | 1.00 | 1.00 | 0.95 | 0.98 | 0.99 | 0.90 | 0.92 | 1.00 | 1.00 |
| EigenspaceOverlapScore | 0.88 | 0.96 | 0.97 | 1.00 | 1.00 | 1.00 | 1.00 | 0.90 | 0.95 | 0.96 | 1.00 | 1.00 | 1.00 | 1.00 |
| Gulp | 0.53 | 0.48 | 0.64 | 1.00 | 1.00 | 1.00 | 1.00 | 0.70 | 0.70 | 0.80 | 1.00 | 1.00 | 1.00 | 1.00 |
| HardCorrelationMatch | 0.01 | 0.53 | 0.76 | 0.91 | 0.74 | 1.00 | 1.00 | 0.61 | 0.85 | 0.89 | 0.92 | 0.93 | 1.00 | 1.00 |
| JaccardSimilarity | 0.55 | 0.65 | 0.78 | 1.00 | 1.00 | 1.00 | 1.00 | 0.85 | 0.90 | 0.91 | 1.00 | 1.00 | 1.00 | 1.00 |
| LinearRegression | 0.55 | 0.96 | 0.93 | 0.55 | 0.78 | 0.99 | 1.00 | 0.85 | 0.95 | 0.91 | 0.85 | 0.91 | 0.99 | 1.00 |
| MagnitudeDifference | -0.37 | 0.13 | 0.14 | 0.21 | 0.28 | 0.84 | 1.00 | 0.35 | 0.46 | 0.55 | 0.64 | 0.65 | 0.86 | 1.00 |
| OrthogonalAngularShapeMetricCentered | 0.55 | 0.65 | 0.65 | 0.96 | 0.99 | 1.00 | 1.00 | 0.85 | 0.90 | 0.90 | 0.97 | 0.98 | 1.00 | 1.00 |
| OrthogonalProcrustesCenteredAndNormalized | 0.55 | 0.65 | 0.65 | 0.96 | 0.99 | 1.00 | 1.00 | 0.85 | 0.90 | 0.90 | 0.97 | 0.98 | 1.00 | 1.00 |
| PermutationProcrustes | 0.20 | 0.60 | 0.39 | 0.69 | 0.14 | 0.71 | 1.00 | 0.63 | 0.70 | 0.61 | 0.71 | 0.59 | 0.70 | 1.00 |
| ProcrustesSizeAndShapeDistance | 0.55 | 0.42 | 0.39 | 0.48 | 0.67 | 0.71 | 1.00 | 0.85 | 0.80 | 0.79 | 0.80 | 0.80 | 0.70 | 1.00 |
| RSA | 0.97 | 0.72 | 0.88 | 0.58 | 0.66 | 1.00 | 1.00 | 0.95 | 0.94 | 0.97 | 0.80 | 0.90 | 1.00 | 1.00 |
| RSMNormDifference | -0.33 | -0.09 | -0.21 | 0.85 | 0.64 | 0.75 | 1.00 | 0.45 | 0.50 | 0.48 | 0.85 | 0.70 | 0.75 | 1.00 |
| RankSimilarity | 0.55 | 0.65 | 0.67 | 1.00 | 1.00 | 1.00 | 1.00 | 0.85 | 0.90 | 0.90 | 1.00 | 1.00 | 1.00 | 1.00 |
| SecondOrderCosineSimilarity | 0.55 | 0.78 | 0.92 | 1.00 | 1.00 | 1.00 | 1.00 | 0.85 | 0.91 | 0.95 | 1.00 | 1.00 | 1.00 | 1.00 |
| SoftCorrelationMatch | 0.11 | 0.50 | 0.70 | 0.52 | 0.64 | 1.00 | 1.00 | 0.72 | 0.85 | 0.88 | 0.80 | 0.89 | 1.00 | 1.00 |
| UniformityDifference | 0.18 | 0.20 | 0.55 | -0.30 | -0.06 | —— | 0.65 | 0.68 | 0.81 | 0.38 | 0.50 | 1.00 | 1.00 | —— |

## J.2 NLP Task

Table 6: Results of Test 1 (Correlation to Accuracy Difference) on MNLI

| Representation Test Architecture | CLS Token Accuracy Correlation | |
|---|---|---|
| | BERT | ALBERT |
| CKA | **0.11** | **0.26** |
| CKA ($\delta = 0.45$) | 0.08 | 0.09 |
| CKA ($\delta = 0.2$) | -0.17 | -0.24 |
| kCKA ($k = 100$) | 0.03 | -0.24 |
| SVCCA | 0.32 | -0.00 |
| RTD | 0.11 | -0.23 |
| IMD | -0.26 | -0.08 |
| MKA ($k = 100$) | -0.17 | -0.26 |
| CKA (linear) | 0.18 | 0.17 |
| MKA ($k = 15$) | -0.16 | -0.24 |
| MKA ($k = 50$) | -0.16 | -0.26 |
| MKA ($k = 200$) | -0.16 | -0.27 |
| AlignedCosineSimilarity | 0.25 | 0.00 |
| ConcentricityDifference | -0.00 | -0.07 |
| DistanceCorrelation | 0.15 | 0.25 |
| EigenspaceOverlapScore | 0.03 | -0.10 |
| Gulp | 0.06 | -0.15 |
| HardCorrelationMatch | 0.04 | 0.21 |
| JaccardSimilarity | -0.21 | -0.25 |
| LinearRegression | 0.20 | 0.04 |
| MagnitudeDifference | 0.22 | -0.06 |
| OrthogonalAngularShapeMetricCentered | 0.28 | 0.12 |
| OrthogonalProcrustesCenteredAndNormalized | 0.27 | 0.12 |
| PWCCA | -0.61 | -0.27 |
| PermutationProcrustes | 0.09 | -0.02 |
| ProcrustesSizeAndShapeDistance | 0.28 | -0.04 |
| RSA | 0.00 | 0.18 |
| RSMNormDifference | 0.30 | -0.15 |
| RankSimilarity | -0.09 | -0.27 |
| SecondOrderCosineSimilarity | -0.26 | -0.25 |
| SoftCorrelationMatch | 0.11 | 0.18 |
| UniformityDifference | 0.14 | -0.16 |

Table 7: Results of Test 2 (Correlation to Output Difference) on MNLI

| Representation Test Architecture | CLS Token | | | | Mean-pooled Token | | | |
|---|---|---|---|---|---|---|---|---|
| | JSD Correlation | | Disagreement Correlation | | JSD Correlation | | Disagreement Correlation | |
| | BERT | ALBERT | BERT | ALBERT | BERT | ALBERT | BERT | ALBERT |
| CKA | 0.36 | **0.29** | 0.15 | **0.57** | 0.51 | 0.06 | 0.38 | 0.18 |
| CKA ($\delta = 0.45$) | 0.31 | 0.05 | **0.50** | 0.22 | **0.53** | 0.02 | **0.49** | 0.16 |
| CKA ($\delta = 0.2$) | -0.25 | -0.11 | 0.27 | -0.04 | 0.13 | -0.06 | -0.25 | 0.05 |
| kCKA ($k = 100$) | 0.27 | 0.03 | 0.33 | -0.00 | 0.45 | 0.05 | 0.41 | 0.10 |
| SVCCA | **0.47** | 0.12 | 0.00 | 0.33 | 0.46 | **0.12** | 0.13 | 0.14 |
| RTD | -0.06 | -0.31 | 0.06 | -0.19 | -0.02 | -0.14 | -0.08 | 0.00 |
| IMD | -0.39 | -0.30 | -0.07 | -0.16 | 0.14 | 0.13 | -0.04 | **0.27** |
| MKA ($k = 100$) | 0.22 | 0.05 | 0.22 | -0.02 | 0.24 | 0.01 | -0.21 | 0.07 |
| CKA (linear) | 0.30 | 0.28 | -0.01 | 0.57 | 0.45 | 0.07 | 0.33 | 0.16 |
| MKA ($k = 15$) | 0.21 | 0.02 | 0.19 | -0.05 | 0.22 | 0.01 | -0.22 | 0.07 |
| MKA ($k = 50$) | 0.22 | 0.05 | 0.22 | -0.02 | 0.24 | 0.01 | -0.21 | 0.07 |
| MKA ($k = 200$) | 0.23 | 0.06 | 0.23 | -0.01 | 0.24 | 0.01 | -0.21 | 0.08 |
| AlignedCosineSimilarity | 0.37 | 0.09 | -0.16 | -0.03 | -0.00 | 0.30 | -0.12 | 0.25 |
| ConcentricityDifference | 0.02 | -0.07 | -0.31 | 0.07 | -0.03 | 0.24 | -0.14 | 0.25 |
| DistanceCorrelation | 0.39 | 0.32 | 0.12 | 0.57 | 0.49 | 0.07 | 0.38 | 0.18 |
| EigenspaceOverlapScore | 0.36 | -0.10 | 0.01 | -0.16 | 0.35 | 0.02 | 0.10 | 0.03 |
| Gulp | 0.39 | -0.05 | 0.05 | -0.11 | 0.38 | 0.03 | 0.10 | 0.02 |
| HardCorrelationMatch | -0.27 | -0.03 | -0.43 | 0.00 | -0.03 | -0.10 | 0.29 | -0.02 |
| JaccardSimilarity | 0.12 | 0.02 | 0.16 | -0.05 | 0.33 | -0.03 | -0.02 | 0.04 |
| LinearRegression | 0.24 | -0.01 | 0.06 | -0.16 | -0.10 | 0.06 | 0.23 | 0.02 |
| MagnitudeDifference | 0.01 | 0.01 | -0.03 | 0.08 | -0.01 | -0.06 | 0.02 | 0.18 |
| OrthogonalAngularShapeMetricCentered | 0.26 | -0.08 | -0.02 | -0.01 | 0.27 | -0.09 | 0.36 | 0.06 |
| OrthogonalProcrustesCenteredAndNormalized | 0.26 | -0.08 | -0.02 | -0.01 | 0.27 | -0.09 | 0.36 | 0.06 |
| PWCCA | -0.32 | 0.32 | 0.13 | 0.32 | 0.35 | 0.45 | -0.20 | 0.38 |
| PermutationProcrustes | -0.06 | 0.04 | -0.30 | -0.04 | -0.05 | 0.07 | -0.28 | 0.16 |
| ProcrustesSizeAndShapeDistance | 0.07 | -0.00 | -0.38 | -0.07 | -0.05 | 0.05 | -0.18 | 0.12 |
| RSA | 0.27 | 0.23 | 0.19 | 0.47 | 0.43 | -0.03 | 0.38 | 0.10 |
| RSMNormDifference | -0.18 | -0.02 | -0.19 | 0.12 | -0.14 | -0.16 | -0.09 | -0.02 |
| RankSimilarity | 0.08 | -0.06 | 0.05 | -0.13 | 0.15 | -0.02 | -0.29 | 0.05 |
| SecondOrderCosineSimilarity | 0.16 | 0.03 | 0.55 | 0.11 | 0.34 | -0.04 | -0.04 | 0.05 |
| SoftCorrelationMatch | -0.23 | -0.02 | -0.42 | 0.01 | 0.00 | -0.03 | 0.31 | 0.02 |
| UniformityDifference | -0.02 | -0.30 | -0.14 | -0.24 | -0.02 | -0.17 | 0.14 | -0.10 |

Table 8: Results of Test 3 (Label Randomization) on MNLI

| Representation Evaluation Architecture | CLS Token | | | | Mean-pooled Token | | | |
|---|---|---|---|---|---|---|---|---|
| | AUPRC | | Conformity Rate | | AUPRC | | Conformity Rate | |
| | BERT | ALBERT | BERT | ALBERT | BERT | ALBERT | BERT | ALBERT |
| CKA | **0.75** | **0.80** | 0.89 | **0.93** | **0.66** | 0.45 | 0.86 | 0.71 |
| CKA ($\delta = 0.45$) | 0.74 | 0.79 | 0.89 | **0.93** | 0.58 | 0.45 | 0.81 | 0.68 |
| CKA ($\delta = 0.2$) | 0.56 | 0.54 | 0.82 | 0.75 | 0.45 | 0.31 | 0.66 | 0.53 |
| kCKA ($k = 100$) | 0.69 | 0.46 | 0.83 | 0.68 | 0.50 | 0.43 | 0.73 | 0.59 |
| SVCCA | 0.69 | 0.69 | 0.84 | 0.90 | **0.66** | 0.46 | **0.87** | 0.76 |
| RTD | 0.73 | 0.54 | 0.78 | 0.79 | 0.60 | **0.62** | 0.79 | **0.84** |
| IMD | 0.74 | 0.49 | **0.92** | 0.76 | 0.58 | 0.43 | 0.86 | 0.74 |
| MKA ($k = 100$) | 0.72 | 0.52 | 0.85 | 0.71 | 0.57 | 0.34 | 0.75 | 0.60 |
| CKA (linear) | 0.75 | 0.85 | 0.90 | 0.93 | 0.64 | 0.43 | 0.84 | 0.68 |
| MKA ($k = 15$) | 0.73 | 0.63 | 0.86 | 0.80 | 0.58 | 0.34 | 0.76 | 0.65 |
| MKA ($k = 50$) | 0.73 | 0.55 | 0.86 | 0.74 | 0.58 | 0.35 | 0.76 | 0.61 |
| MKA ($k = 200$) | 0.71 | 0.51 | 0.84 | 0.69 | 0.56 | 0.34 | 0.75 | 0.59 |
| AlignedCosineSimilarity | 1.00 | 0.68 | 1.00 | 0.91 | 0.80 | 0.65 | 0.94 | 0.80 |
| ConcentricityDifference | 1.00 | 0.81 | 1.00 | 0.89 | 0.76 | 0.52 | 0.90 | 0.78 |
| DistanceCorrelation | 0.75 | 0.79 | 0.89 | 0.93 | 0.66 | 0.50 | 0.86 | 0.72 |
| EigenspaceOverlapScore | 0.62 | 0.70 | 0.88 | 0.88 | 0.57 | 0.76 | 0.86 | 0.90 |
| Gulp | 0.62 | 0.39 | 0.90 | 0.70 | 0.53 | 0.43 | 0.83 | 0.73 |
| HardCorrelationMatch | 0.75 | 0.68 | 0.90 | 0.86 | 0.53 | 0.55 | 0.82 | 0.81 |
| JaccardSimilarity | 0.60 | 0.58 | 0.74 | 0.76 | 0.65 | 0.50 | 0.81 | 0.70 |
| LinearRegression | 0.43 | 0.29 | 0.80 | 0.67 | 0.43 | 0.43 | 0.68 | 0.68 |
| MagnitudeDifference | 0.33 | 0.56 | 0.75 | 0.76 | 0.39 | 0.48 | 0.78 | 0.81 |
| OrthogonalAngularShapeMetricCentered | 0.90 | 0.97 | 0.98 | 0.99 | 0.71 | 0.64 | 0.94 | 0.82 |
| OrthogonalProcrustesCenteredAndNormalized | 0.90 | 0.97 | 0.98 | 0.99 | 0.71 | 0.64 | 0.94 | 0.82 |
| PWCCA | 0.78 | —— | 0.95 | 1.00 | —— | 0.43 | 1.00 | 0.59 |
| PermutationProcrustes | 0.44 | 0.60 | 0.68 | 0.81 | 0.40 | 0.57 | 0.61 | 0.79 |
| ProcrustesSizeAndShapeDistance | 0.98 | 0.84 | 0.99 | 0.96 | 0.69 | 0.62 | 0.87 | 0.79 |
| RSA | 0.47 | 0.61 | 0.69 | 0.81 | 0.47 | 0.43 | 0.66 | 0.66 |
| RSMNormDifference | 1.00 | 1.00 | 1.00 | 1.00 | 0.86 | 0.59 | 0.94 | 0.83 |
| RankSimilarity | 0.57 | 0.48 | 0.73 | 0.65 | 0.57 | 0.36 | 0.78 | 0.63 |
| SecondOrderCosineSimilarity | 0.71 | —— | 0.72 | 1.00 | 0.73 | —— | 0.80 | 1.00 |
| SoftCorrelationMatch | 0.75 | 0.71 | 0.92 | 0.87 | 0.68 | 0.64 | 0.86 | 0.89 |
| UniformityDifference | 0.76 | 0.75 | 0.91 | 0.90 | 0.88 | 0.65 | 0.94 | 0.88 |

Table 9: Results of Test 4 (Shortcut Affinity) on MNLI

| Representation | CLS Token | | | | Mean-pooled Token | | | |
| Evaluation | AUPRC | | Conformity Rate | | AUPRC | | Conformity Rate | |
| Architecture | BERT | ALBERT | BERT | ALBERT | BERT | ALBERT | BERT | ALBERT |
|---|---|---|---|---|---|---|---|---|
| CKA | 0.59 | 0.63 | **0.88** | 0.67 | 0.55 | 0.56 | **0.85** | 0.64 |
| CKA ($\delta = 0.45$) | 0.58 | **0.67** | 0.87 | **0.81** | 0.55 | 0.56 | 0.84 | 0.66 |
| CKA ($\delta = 0.2$) | 0.50 | 0.43 | 0.79 | 0.58 | 0.55 | 0.32 | 0.74 | 0.52 |
| kCKA ($k = 100$) | 0.57 | 0.61 | 0.83 | 0.65 | 0.56 | **0.58** | 0.82 | 0.66 |
| SVCCA | 0.42 | 0.60 | 0.78 | 0.62 | 0.49 | 0.52 | 0.80 | 0.61 |
| RTD | **0.61** | 0.41 | 0.84 | 0.57 | **0.64** | 0.39 | 0.84 | 0.53 |
| IMD | 0.53 | 0.30 | 0.82 | 0.45 | 0.52 | 0.29 | 0.81 | 0.48 |
| MKA ($k = 100$) | 0.56 | 0.59 | 0.81 | 0.65 | 0.56 | 0.54 | 0.80 | **0.67** |
| CKA (linear) | 0.59 | 0.63 | 0.88 | 0.65 | 0.53 | 0.50 | 0.84 | 0.61 |
| MKA ($k = 15$) | 0.56 | 0.59 | 0.81 | 0.66 | 0.56 | 0.51 | 0.80 | 0.66 |
| MKA ($k = 50$) | 0.56 | 0.59 | 0.81 | 0.66 | 0.56 | 0.54 | 0.81 | 0.67 |
| MKA ($k = 200$) | 0.56 | 0.59 | 0.81 | 0.65 | 0.56 | 0.54 | 0.80 | 0.67 |
| AlignedCosineSimilarity | 0.58 | 0.37 | 0.89 | 0.58 | 0.53 | 0.45 | 0.83 | 0.58 |
| ConcentricityDifference | 0.38 | 0.42 | 0.75 | 0.45 | 0.35 | 0.40 | 0.65 | 0.48 |
| DistanceCorrelation | 0.58 | 0.62 | 0.88 | 0.67 | 0.54 | 0.54 | 0.85 | 0.63 |
| EigenspaceOverlapScore | 0.57 | 0.45 | 0.85 | 0.58 | 0.39 | 0.55 | 0.67 | 0.64 |
| Gulp | 0.62 | 0.46 | 0.87 | 0.58 | 0.45 | 0.56 | 0.77 | 0.65 |
| HardCorrelationMatch | 0.55 | 0.35 | 0.82 | 0.60 | 0.32 | 0.41 | 0.75 | 0.61 |
| JaccardSimilarity | 0.56 | 0.58 | 0.81 | 0.63 | 0.56 | 0.59 | 0.82 | 0.70 |
| LinearRegression | 0.36 | 0.43 | 0.69 | 0.48 | 0.29 | 0.45 | 0.64 | 0.60 |
| MagnitudeDifference | 0.48 | 0.56 | 0.81 | 0.77 | 0.30 | 0.30 | 0.58 | 0.47 |
| OrthogonalAngularShapeMetricCentered | 0.60 | 0.48 | 0.90 | 0.65 | 0.57 | 0.49 | 0.87 | 0.64 |
| OrthogonalProcrustesCenteredAndNormalized | 0.60 | 0.48 | 0.90 | 0.65 | 0.57 | 0.49 | 0.87 | 0.64 |
| PWCCA | —— | 0.42 | 1.00 | 0.61 | 0.43 | —— | 0.74 | 1.00 |
| PermutationProcrustes | 0.52 | 0.54 | 0.82 | 0.63 | 0.35 | 0.54 | 0.60 | 0.58 |
| ProcrustesSizeAndShapeDistance | 0.54 | 0.54 | 0.87 | 0.62 | 0.53 | 0.51 | 0.84 | 0.57 |
| RSA | 0.58 | 0.59 | 0.87 | 0.64 | 0.47 | 0.52 | 0.79 | 0.61 |
| RSMNormDifference | 0.28 | 0.57 | 0.66 | 0.69 | 0.36 | 0.44 | 0.68 | 0.61 |
| RankSimilarity | 0.58 | 0.61 | 0.82 | 0.66 | 0.50 | 0.45 | 0.73 | 0.61 |
| SecondOrderCosineSimilarity | 0.59 | 0.56 | 0.82 | 0.60 | 0.58 | 0.51 | 0.83 | 0.64 |
| SoftCorrelationMatch | 0.55 | 0.36 | 0.82 | 0.60 | 0.33 | 0.41 | 0.75 | 0.62 |
| UniformityDifference | 0.38 | 0.32 | 0.68 | 0.49 | 0.33 | 0.44 | 0.68 | 0.52 |

Table 10: Results of Test 5 (Augmentation) on MNLI

| Representation | CLS Token | | | | Mean-pooled Token | | | |
| Evaluation | AUPRC | | Conformity Rate | | AUPRC | | Conformity Rate | |
| Architecture | BERT | ALBERT | BERT | ALBERT | BERT | ALBERT | BERT | ALBERT |
|---|---|---|---|---|---|---|---|---|
| CKA | 0.44 | **0.85** | **0.84** | 0.84 | **0.36** | **0.74** | **0.79** | **0.78** |
| CKA ($\delta = 0.45$) | 0.45 | **0.85** | 0.80 | **0.86** | 0.34 | 0.72 | 0.77 | **0.78** |
| CKA ($\delta = 0.2$) | 0.43 | 0.69 | 0.71 | 0.74 | 0.35 | 0.42 | 0.56 | 0.55 |
| kCKA ($k = 100$) | 0.37 | 0.76 | 0.71 | 0.78 | 0.35 | 0.63 | 0.73 | 0.73 |
| SVCCA | 0.43 | 0.60 | 0.77 | 0.66 | 0.31 | 0.63 | 0.70 | 0.71 |
| RTD | **0.53** | 0.71 | 0.82 | 0.77 | 0.28 | 0.41 | 0.66 | 0.53 |
| IMD | 0.43 | 0.37 | 0.76 | 0.48 | 0.20 | 0.30 | 0.51 | 0.46 |
| MKA ($k = 100$) | 0.34 | 0.77 | 0.71 | 0.79 | 0.33 | 0.52 | 0.65 | 0.68 |
| CKA (linear) | 0.48 | 0.83 | 0.87 | 0.82 | 0.34 | 0.72 | 0.76 | 0.77 |
| MKA ($k = 15$) | 0.34 | 0.77 | 0.72 | 0.79 | 0.32 | 0.52 | 0.64 | 0.68 |
| MKA ($k = 50$) | 0.34 | 0.77 | 0.71 | 0.79 | 0.33 | 0.52 | 0.65 | 0.68 |
| MKA ($k = 200$) | 0.34 | 0.77 | 0.70 | 0.79 | 0.33 | 0.52 | 0.65 | 0.68 |
| AlignedCosineSimilarity | 0.35 | 0.77 | 0.80 | 0.80 | 0.28 | 0.55 | 0.63 | 0.68 |
| ConcentricityDifference | 0.28 | 0.55 | 0.61 | 0.67 | 0.27 | 0.44 | 0.65 | 0.52 |
| DistanceCorrelation | 0.45 | 0.84 | 0.85 | 0.86 | 0.35 | 0.73 | 0.78 | 0.77 |
| EigenspaceOverlapScore | 0.26 | 0.74 | 0.71 | 0.78 | 0.24 | 0.73 | 0.64 | 0.77 |
| Gulp | 0.27 | 0.73 | 0.68 | 0.79 | 0.25 | 0.73 | 0.65 | 0.77 |
| HardCorrelationMatch | 0.24 | 0.65 | 0.67 | 0.76 | 0.21 | 0.63 | 0.57 | 0.84 |
| JaccardSimilarity | 0.35 | 0.74 | 0.74 | 0.78 | 0.32 | 0.63 | 0.71 | 0.73 |
| LinearRegression | 0.33 | 0.48 | 0.74 | 0.71 | 0.34 | 0.61 | 0.64 | 0.76 |
| MagnitudeDifference | 0.16 | 0.60 | 0.45 | 0.79 | 0.28 | 0.68 | 0.65 | 0.86 |
| OrthogonalAngularShapeMetricCentered | 0.37 | 0.91 | 0.84 | 0.94 | 0.29 | 0.81 | 0.75 | 0.82 |
| OrthogonalProcrustesCenteredAndNormalized | 0.37 | 0.91 | 0.84 | 0.94 | 0.29 | 0.81 | 0.75 | 0.82 |
| PWCCA | 0.39 | 0.52 | 0.79 | 0.48 | —— | —— | —— | —— |
| PermutationProcrustes | 0.18 | 0.49 | 0.50 | 0.55 | 0.17 | 0.51 | 0.44 | 0.59 |
| ProcrustesSizeAndShapeDistance | 0.31 | 0.74 | 0.73 | 0.85 | 0.25 | 0.49 | 0.60 | 0.59 |
| RSA | 0.48 | 0.86 | 0.86 | 0.84 | 0.34 | 0.75 | 0.76 | 0.78 |
| RSMNormDifference | 0.36 | 0.85 | 0.67 | 0.84 | 0.24 | 0.42 | 0.58 | 0.52 |
| RankSimilarity | 0.33 | 0.73 | 0.71 | 0.77 | 0.35 | 0.50 | 0.60 | 0.66 |
| SecondOrderCosineSimilarity | 0.44 | 0.68 | 0.64 | 0.72 | 0.38 | 0.58 | 0.66 | 0.69 |
| SoftCorrelationMatch | 0.27 | 0.62 | 0.72 | 0.75 | 0.21 | 0.59 | 0.57 | 0.81 |
| UniformityDifference | 0.61 | 0.44 | 0.84 | 0.60 | 0.42 | 0.33 | 0.75 | 0.50 |

Table 11: Results of Test 6 (Layer Monotonicity) on MNLI

| Representation | CLS Token | | | | Mean-pooled Token | | | |
| Evaluation | Spearman | | Conformity Rate | | Spearman | | Conformity Rate | |
| Architecture | BERT | ALBERT | BERT | ALBERT | BERT | ALBERT | BERT | ALBERT |
|---|---|---|---|---|---|---|---|---|
| CKA | **0.99** | 0.98 | **0.98** | 0.97 | **1.00** | 0.99 | 0.99 | 0.99 |
| CKA ($\delta = 0.45$) | **0.99** | 0.98 | **0.98** | 0.97 | **1.00** | **1.00** | 0.99 | **1.00** |
| CKA ($\delta = 0.2$) | 0.92 | 0.95 | 0.90 | 0.93 | 0.95 | **1.00** | 0.96 | **1.00** |
| kCKA ($k = 100$) | 0.91 | 0.90 | 0.90 | 0.89 | **1.00** | **1.00** | **1.00** | **1.00** |
| SVCCA | 0.91 | 0.77 | 0.91 | 0.85 | 0.78 | 0.83 | 0.85 | 0.87 |
| RTD | 0.42 | 0.43 | 0.76 | 0.75 | 0.95 | 0.89 | 0.94 | 0.88 |
| IMD | 0.44 | 0.79 | 0.68 | 0.85 | 0.71 | 0.75 | 0.74 | 0.89 |
| MKA ($k = 100$) | **0.99** | **0.99** | 0.97 | **0.99** | **1.00** | **1.00** | **1.00** | **1.00** |
| CKA (linear) | 1.00 | 0.99 | 0.99 | 0.98 | 1.00 | 0.99 | 0.99 | 0.99 |
| MKA ($k = 15$) | 0.97 | 0.99 | 0.95 | 0.98 | 1.00 | 1.00 | 1.00 | 1.00 |
| MKA ($k = 50$) | 0.99 | 0.99 | 0.97 | 0.99 | 1.00 | 1.00 | 1.00 | 1.00 |
| MKA ($k = 200$) | 0.99 | 0.99 | 0.98 | 0.98 | 1.00 | 1.00 | 1.00 | 1.00 |
| AlignedCosineSimilarity | 1.00 | 0.95 | 1.00 | 0.95 | 1.00 | 1.00 | 1.00 | 0.99 |
| ConcentricityDifference | 0.99 | 0.87 | 0.99 | 0.90 | 0.50 | 0.34 | 0.62 | 0.62 |
| DistanceCorrelation | 0.97 | 0.99 | 0.96 | 0.99 | 1.00 | 0.99 | 0.99 | 0.99 |
| EigenspaceOverlapScore | 0.99 | 0.96 | 0.98 | 0.96 | 1.00 | 1.00 | 1.00 | 1.00 |
| Gulp | 0.89 | 0.91 | 0.90 | 0.91 | 1.00 | 0.94 | 1.00 | 0.94 |
| HardCorrelationMatch | 1.00 | 1.00 | 1.00 | 0.99 | 1.00 | 1.00 | 1.00 | 1.00 |
| JaccardSimilarity | 0.95 | 0.95 | 0.94 | 0.94 | 1.00 | 1.00 | 1.00 | 1.00 |
| LinearRegression | 0.57 | 0.66 | 0.82 | 0.83 | 0.40 | 0.40 | 0.71 | 0.75 |
| MagnitudeDifference | 0.52 | 0.90 | 0.64 | 0.93 | 0.66 | 0.60 | 0.81 | 0.88 |
| OrthogonalAngularShapeMetricCentered | 1.00 | 0.99 | 0.99 | 0.99 | 1.00 | 1.00 | 1.00 | 1.00 |
| OrthogonalProcrustesCenteredAndNormalized | 0.99 | 0.99 | 0.99 | 0.98 | 1.00 | 1.00 | 1.00 | 1.00 |
| PWCCA | —— | 1.00 | 1.00 | 1.00 | 1.00 | 1.00 | 1.00 | 1.00 |
| PermutationProcrustes | 0.73 | 0.96 | 0.80 | 0.95 | 0.90 | 0.95 | 0.90 | 0.93 |
| ProcrustesSizeAndShapeDistance | 0.92 | 0.94 | 0.96 | 0.94 | 0.99 | 0.99 | 0.99 | 0.99 |
| RSA | 1.00 | 1.00 | 0.99 | 1.00 | 1.00 | 1.00 | 1.00 | 1.00 |
| RSMNormDifference | 0.84 | 0.89 | 0.94 | 0.90 | 0.85 | 0.88 | 0.91 | 0.94 |
| RankSimilarity | 0.89 | 0.92 | 0.92 | 0.93 | 0.51 | 0.64 | 0.79 | 0.87 |
| SecondOrderCosineSimilarity | 0.94 | 0.91 | 0.94 | 0.92 | 1.00 | 1.00 | 1.00 | 1.00 |
| SoftCorrelationMatch | 0.99 | 0.99 | 0.98 | 0.98 | 1.00 | 1.00 | 0.99 | 0.99 |
| UniformityDifference | 0.81 | 0.87 | 0.94 | 0.91 | 0.83 | 0.91 | 0.95 | 0.98 |

### J.3 Graph Task

Table 12: Results of Test 1 (Correlation to Accuracy Difference) for the graph domain

| Evaluation Dataset Architecture | Spearman | | | | | | | | | |
| | Cora | | | | Flickr | | | OGBN-Arxiv | | |
| | GCN | SAGE | GAT | PGNN | GCN | SAGE | GAT | GCN | SAGE | GAT |
|---|---|---|---|---|---|---|---|---|---|---|
| CKA | 0.04 | -0.15 | 0.03 | -0.01 | 0.50 | **0.41** | -0.14 | -0.07 | 0.14 | **-0.03** |
| CKA ($\delta = 0.45$) | **0.09** | -0.22 | 0.00 | -0.06 | **0.52** | 0.17 | -0.16 | -0.12 | 0.18 | -0.13 |
| CKA ($\delta = 0.2$) | 0.01 | -0.29 | 0.07 | 0.03 | 0.43 | 0.14 | **0.11** | -0.22 | 0.17 | -0.28 |
| kCKA ($k = 100$) | 0.07 | -0.25 | -0.01 | **0.11** | 0.42 | -0.22 | -0.28 | **-0.06** | **0.23** | -0.16 |
| SVCCA | -0.03 | -0.12 | -0.16 | 0.08 | 0.01 | 0.01 | -0.18 | -0.27 | 0.09 | -0.10 |
| RTD | 0.16 | -0.26 | -0.02 | -0.30 | 0.24 | -0.02 | -0.16 | -0.32 | -0.07 | -0.27 |
| IMD | 0.03 | **0.00** | -0.02 | -0.04 | -0.10 | 0.36 | -0.09 | -0.24 | -0.02 | -0.15 |
| MKA ($k = 100$) | 0.01 | -0.18 | **0.11** | -0.15 | 0.32 | -0.05 | -0.19 | -0.27 | 0.08 | -0.26 |
| CKA (linear) | 0.03 | -0.18 | 0.03 | 0.09 | 0.03 | 0.27 | -0.16 | -0.17 | 0.11 | -0.05 |
| MKA ($k = 15$) | 0.00 | -0.21 | 0.13 | -0.13 | 0.32 | -0.04 | -0.21 | -0.31 | 0.05 | -0.26 |
| MKA ($k = 50$) | 0.00 | -0.17 | 0.13 | -0.12 | 0.31 | -0.04 | -0.19 | -0.28 | 0.08 | -0.25 |
| MKA ($k = 200$) | 0.02 | -0.16 | 0.11 | -0.12 | 0.32 | -0.06 | -0.18 | -0.26 | 0.09 | -0.28 |
| AlignedCosineSimilarity | -0.02 | 0.13 | -0.32 | -0.04 | 0.35 | 0.24 | -0.07 | -0.08 | 0.17 | -0.17 |
| ConcentricityDifference | 0.13 | -0.25 | -0.22 | 0.13 | -0.08 | -0.29 | -0.07 | -0.07 | -0.13 | -0.12 |
| DistanceCorrelation | -0.03 | -0.18 | 0.03 | 0.13 | 0.41 | 0.42 | -0.19 | -0.10 | 0.15 | -0.06 |
| EigenspaceOverlapScore | -0.19 | 0.07 | -0.05 | -0.06 | 0.15 | -0.27 | 0.29 | -0.21 | 0.05 | -0.32 |
| Gulp | -0.20 | 0.07 | -0.12 | 0.12 | 0.26 | -0.27 | -0.27 | -0.05 | 0.06 | -0.34 |
| HardCorrelationMatch | -0.00 | -0.11 | -0.14 | 0.16 | 0.31 | 0.35 | 0.06 | 0.36 | 0.02 | 0.04 |
| JaccardSimilarity | 0.05 | -0.12 | -0.16 | 0.02 | 0.32 | 0.28 | -0.18 | -0.32 | -0.12 | -0.15 |
| LinearRegression | 0.07 | -0.22 | -0.13 | 0.05 | -0.03 | 0.17 | -0.18 | 0.07 | -0.01 | -0.19 |
| MagnitudeDifference | 0.10 | -0.13 | -0.21 | 0.13 | 0.02 | -0.17 | 0.14 | -0.18 | -0.20 | 0.11 |
| OrthogonalAngularShapeMetricCentered | 0.03 | -0.29 | -0.13 | 0.23 | 0.39 | 0.28 | -0.15 | -0.04 | 0.09 | -0.09 |
| OrthogonalProcrustesCenteredAndNormalized | 0.03 | -0.29 | -0.13 | 0.23 | 0.39 | 0.28 | -0.15 | -0.04 | 0.09 | -0.09 |
| PWCCA | -0.16 | 0.06 | -0.26 | -0.15 | —— | -0.05 | -0.16 | -0.12 | 0.06 | -0.30 |
| PermutationProcrustes | 0.05 | 0.19 | -0.28 | 0.34 | 0.20 | -0.19 | 0.15 | -0.09 | 0.03 | 0.43 |
| ProcrustesSizeAndShapeDistance | 0.04 | 0.01 | -0.21 | 0.33 | 0.02 | -0.06 | 0.11 | -0.17 | 0.07 | 0.43 |
| RSA | 0.06 | 0.04 | -0.31 | 0.20 | 0.53 | 0.32 | -0.08 | -0.07 | 0.25 | 0.32 |
| RSMNormDifference | -0.06 | 0.08 | -0.14 | 0.28 | -0.18 | -0.16 | 0.13 | -0.05 | -0.19 | 0.02 |
| RankSimilarity | 0.00 | -0.10 | 0.34 | 0.11 | 0.35 | 0.31 | -0.19 | -0.26 | 0.05 | -0.10 |
| SecondOrderCosineSimilarity | 0.02 | 0.04 | -0.12 | 0.11 | 0.54 | -0.19 | 0.01 | -0.47 | 0.22 | -0.19 |
| SoftCorrelationMatch | 0.07 | -0.05 | 0.02 | 0.12 | 0.30 | 0.33 | -0.07 | 0.35 | 0.12 | 0.12 |
| UniformityDifference | -0.06 | -0.05 | -0.08 | -0.06 | -0.18 | 0.03 | -0.18 | -0.19 | -0.20 | -0.25 |

Table 13: Results of Test 2 (Correlation to Output Difference) for the graph domain

| Type | Grounding by Prediction | | | | | | | | | | | | | | | | | | | |
| Test | JSD Correlation | | | | | | | | | | Disagreement Correlation | | | | | | | | | |
| Evaluation | Spearman | | | | | | | | | | Spearman | | | | | | | | | |
| Dataset | Cora | | | | Flickr | | | OGBN-Arxiv | | | Cora | | | | Flickr | | | OGBN-Arxiv | | |
| Architecture | GCN | SAGE | GAT | PGNN | GCN | SAGE | GAT | GCN | SAGE | GAT | GCN | SAGE | GAT | PGNN | GCN | SAGE | GAT | GCN | SAGE | GAT |
|---|---|---|---|---|---|---|---|---|---|---|---|---|---|---|---|---|---|---|---|---|
| CKA | 0.73 | 0.12 | 0.59 | -0.23 | 0.54 | **0.35** | 0.12 | 0.18 | 0.19 | **0.38** | 0.65 | -0.05 | **0.44** | 0.01 | 0.26 | 0.32 | 0.02 | 0.13 | 0.14 | **0.21** |
| CKA ($\delta = 0.45$) | 0.73 | 0.30 | 0.52 | -0.08 | **0.56** | 0.18 | 0.15 | 0.22 | 0.22 | 0.25 | 0.64 | -0.06 | 0.34 | 0.14 | **0.29** | 0.15 | 0.01 | 0.21 | **0.18** | 0.09 |
| CKA ($\delta = 0.2$) | 0.61 | 0.49 | 0.52 | -0.04 | 0.41 | 0.25 | -0.42 | 0.16 | 0.24 | 0.11 | 0.51 | 0.17 | 0.41 | 0.03 | 0.14 | 0.22 | -0.43 | 0.15 | 0.17 | -0.01 |
| kCKA ($k = 100$) | 0.73 | 0.46 | **0.60** | **0.07** | 0.44 | -0.19 | 0.08 | **0.29** | 0.09 | 0.27 | 0.62 | 0.01 | 0.37 | **0.18** | 0.15 | -0.15 | -0.05 | **0.32** | 0.08 | 0.14 |
| SVCCA | 0.52 | 0.02 | 0.26 | -0.08 | -0.04 | 0.23 | 0.13 | 0.19 | 0.16 | -0.17 | 0.59 | 0.04 | 0.17 | -0.01 | -0.27 | 0.21 | **0.03** | 0.08 | 0.11 | -0.15 |
| RTD | 0.52 | **0.54** | 0.13 | -0.01 | 0.26 | -0.28 | 0.06 | 0.01 | 0.00 | 0.07 | 0.49 | **0.24** | 0.07 | -0.19 | 0.05 | -0.22 | **0.03** | 0.02 | -0.11 | -0.05 |
| IMD | -0.12 | -0.43 | -0.13 | -0.01 | -0.08 | 0.29 | 0.04 | -0.12 | -0.03 | -0.02 | -0.18 | -0.10 | -0.20 | -0.21 | -0.10 | **0.33** | -0.02 | -0.14 | 0.01 | 0.01 |
| MKA ($k = 100$) | **0.77** | 0.53 | 0.48 | -0.06 | 0.32 | 0.33 | **0.16** | 0.18 | **0.30** | 0.11 | **0.74** | 0.15 | 0.34 | -0.23 | 0.01 | **0.33** | 0.02 | 0.18 | 0.16 | 0.00 |
| CKA (linear) | 0.71 | -0.03 | 0.53 | -0.22 | 0.03 | 0.58 | 0.17 | 0.12 | -0.02 | 0.38 | 0.65 | 0.00 | 0.45 | -0.02 | -0.21 | 0.53 | 0.06 | 0.03 | -0.04 | 0.23 |
| MKA ($k = 15$) | 0.75 | 0.52 | 0.47 | -0.35 | 0.31 | 0.26 | 0.17 | 0.16 | 0.30 | 0.10 | 0.73 | 0.13 | 0.31 | -0.18 | 0.00 | 0.24 | 0.03 | 0.15 | 0.14 | 0.00 |
| MKA ($k = 50$) | 0.76 | 0.53 | 0.48 | -0.20 | 0.32 | 0.31 | 0.17 | 0.17 | 0.29 | 0.12 | 0.73 | 0.14 | 0.33 | -0.20 | 0.01 | 0.30 | 0.03 | 0.17 | 0.15 | 0.01 |
| MKA ($k = 200$) | 0.77 | 0.55 | 0.49 | -0.01 | 0.33 | 0.33 | 0.15 | 0.20 | 0.31 | 0.11 | 0.73 | 0.15 | 0.34 | -0.22 | 0.02 | 0.33 | 0.02 | 0.20 | 0.18 | -0.01 |
| AlignedCosineSimilarity | 0.32 | 0.38 | 0.17 | -0.14 | 0.31 | 0.44 | -0.01 | -0.03 | 0.05 | 0.28 | 0.27 | 0.27 | -0.05 | 0.14 | 0.15 | 0.37 | -0.08 | -0.10 | 0.00 | 0.17 |
| ConcentricityDifference | 0.41 | 0.04 | 0.01 | 0.26 | -0.17 | -0.03 | 0.03 | -0.13 | 0.02 | -0.22 | 0.31 | -0.10 | 0.03 | 0.03 | -0.21 | -0.04 | 0.03 | -0.25 | 0.07 | -0.16 |
| DistanceCorrelation | 0.71 | 0.05 | 0.60 | -0.23 | 0.46 | 0.43 | 0.03 | 0.16 | 0.12 | 0.36 | 0.63 | -0.08 | 0.46 | 0.08 | 0.17 | 0.40 | -0.03 | 0.11 | 0.08 | 0.20 |
| EigenspaceOverlapScore | -0.50 | 0.22 | -0.04 | -0.14 | -0.03 | 0.38 | 0.11 | -0.17 | 0.11 | 0.37 | -0.46 | 0.02 | -0.09 | 0.21 | 0.02 | 0.33 | 0.23 | -0.25 | -0.02 | 0.12 |
| Gulp | -0.58 | 0.18 | -0.04 | 0.33 | 0.48 | 0.38 | 0.12 | 0.13 | 0.11 | 0.35 | -0.54 | -0.09 | -0.13 | 0.34 | 0.29 | 0.33 | -0.01 | 0.11 | -0.04 | 0.10 |
| HardCorrelationMatch | 0.75 | 0.16 | 0.52 | -0.07 | 0.53 | 0.50 | 0.09 | -0.05 | -0.28 | 0.46 | 0.66 | 0.10 | 0.26 | 0.33 | 0.40 | 0.46 | -0.09 | 0.02 | -0.24 | 0.24 |
| JaccardSimilarity | 0.78 | 0.46 | 0.38 | -0.01 | 0.33 | 0.42 | 0.11 | 0.20 | 0.09 | 0.37 | 0.68 | 0.33 | 0.12 | 0.01 | 0.04 | 0.42 | 0.00 | 0.15 | 0.01 | 0.22 |
| LinearRegression | 0.39 | 0.33 | 0.19 | -0.21 | 0.05 | 0.48 | 0.18 | -0.10 | -0.17 | 0.47 | 0.35 | 0.09 | -0.06 | 0.10 | 0.01 | 0.46 | 0.06 | -0.12 | -0.19 | 0.22 |
| MagnitudeDifference | 0.44 | 0.06 | -0.15 | 0.23 | 0.03 | 0.06 | -0.26 | -0.13 | -0.13 | 0.08 | 0.32 | -0.25 | -0.07 | 0.27 | 0.06 | 0.07 | -0.20 | -0.25 | -0.19 | 0.22 |
| OrthogonalAngularShapeMetricCentered | 0.73 | 0.27 | 0.28 | -0.10 | 0.43 | 0.63 | 0.13 | 0.04 | -0.15 | 0.44 | 0.66 | 0.16 | 0.05 | 0.17 | 0.19 | 0.57 | 0.03 | 0.02 | -0.16 | 0.27 |
| OrthogonalProcrustesCenteredAndNormalized | 0.73 | 0.27 | 0.28 | -0.10 | 0.43 | 0.63 | 0.13 | 0.04 | -0.15 | 0.44 | 0.66 | 0.16 | 0.05 | 0.17 | 0.19 | 0.57 | 0.03 | 0.02 | -0.16 | 0.27 |
| PWCCA | -0.20 | 0.27 | 0.02 | -0.23 | —— | 0.38 | 0.27 | -0.21 | -0.04 | 0.29 | -0.20 | 0.36 | -0.14 | 0.23 | —— | 0.32 | 0.16 | -0.24 | -0.08 | 0.03 |
| PermutationProcrustes | 0.68 | 0.10 | 0.13 | 0.26 | 0.29 | -0.10 | -0.42 | 0.34 | -0.55 | 0.06 | 0.65 | 0.33 | -0.14 | 0.38 | 0.18 | -0.10 | -0.27 | 0.25 | -0.41 | 0.22 |
| ProcrustesSizeAndShapeDistance | 0.69 | 0.08 | 0.05 | 0.27 | 0.02 | -0.18 | -0.38 | 0.36 | -0.50 | 0.13 | 0.62 | 0.24 | -0.17 | 0.38 | -0.13 | -0.15 | -0.27 | 0.26 | -0.38 | 0.30 |
| RSA | 0.46 | 0.13 | 0.20 | 0.23 | 0.52 | 0.63 | 0.14 | 0.20 | 0.14 | 0.46 | 0.47 | 0.33 | -0.02 | 0.38 | 0.35 | 0.59 | 0.06 | 0.17 | 0.15 | 0.45 |
| RSMNormDifference | 0.33 | -0.01 | -0.07 | 0.20 | -0.22 | -0.04 | -0.37 | -0.20 | 0.11 | 0.33 | 0.34 | 0.14 | -0.15 | 0.30 | -0.30 | 0.00 | -0.26 | -0.14 | 0.09 | 0.36 |
| RankSimilarity | 0.39 | 0.55 | 0.54 | -0.04 | 0.33 | 0.30 | 0.22 | 0.03 | 0.36 | 0.13 | 0.46 | 0.21 | 0.46 | 0.21 | 0.06 | 0.24 | 0.05 | 0.04 | 0.27 | 0.09 |
| SecondOrderCosineSimilarity | 0.81 | 0.54 | 0.53 | -0.14 | 0.47 | 0.15 | 0.07 | 0.11 | 0.34 | 0.11 | 0.69 | 0.25 | 0.30 | 0.00 | 0.30 | 0.14 | -0.10 | 0.04 | 0.27 | 0.06 |
| SoftCorrelationMatch | 0.80 | -0.01 | 0.21 | -0.06 | 0.53 | 0.53 | 0.23 | 0.02 | -0.28 | 0.56 | 0.73 | 0.06 | 0.07 | 0.31 | 0.39 | 0.49 | 0.02 | 0.03 | -0.26 | 0.35 |
| UniformityDifference | -0.11 | 0.27 | -0.11 | -0.22 | -0.32 | 0.02 | 0.21 | -0.34 | 0.12 | 0.12 | -0.13 | 0.16 | -0.15 | 0.08 | -0.34 | 0.04 | 0.18 | -0.33 | 0.10 | 0.06 |

Table 14: Results of Test 3 (Label Randomization) for the graph domain

| Evaluation | AUPRC | | | | | | | | | | Conformity Rate | | | | | | | | | |
| Dataset | Cora | | | | Flickr | | | OGBN-Arxiv | | | Cora | | | | Flickr | | | OGBN-Arxiv | | |
| Architecture | GCN | SAGE | GAT | PGNN | GCN | SAGE | GAT | GCN | SAGE | GAT | GCN | SAGE | GAT | PGNN | GCN | SAGE | GAT | GCN | SAGE | GAT |
|---|---|---|---|---|---|---|---|---|---|---|---|---|---|---|---|---|---|---|---|---|
| CKA | 0.48 | 0.56 | 0.43 | 0.25 | 0.88 | 0.42 | 0.31 | 1.00 | 1.00 | 1.00 | 0.72 | 0.78 | 0.54 | 0.64 | 0.96 | 0.50 | 0.61 | 1.00 | 1.00 | 1.00 |
| CKA ($\delta = 0.45$) | 0.50 | 0.50 | 0.52 | 0.26 | **0.92** | 0.45 | **0.38** | 1.00 | 0.96 | 1.00 | 0.75 | 0.77 | 0.76 | **0.65** | **0.98** | 0.66 | **0.69** | 1.00 | 0.98 | 1.00 |
| CKA ($\delta = 0.2$) | **0.74** | 0.72 | 0.73 | 0.21 | 0.78 | 0.42 | 0.23 | 1.00 | 0.90 | 0.99 | **0.93** | 0.88 | 0.86 | 0.52 | 0.93 | 0.50 | 0.57 | 1.00 | 0.98 | 1.00 |
| kCKA ($k = 100$) | 0.43 | 0.42 | 0.42 | 0.24 | 0.74 | 0.43 | 0.37 | 1.00 | 0.74 | 0.92 | 0.59 | 0.52 | 0.52 | 0.58 | 0.88 | 0.61 | 0.66 | 1.00 | 0.88 | 0.98 |
| SVCCA | 0.37 | 0.31 | 0.42 | 0.19 | 0.33 | **0.80** | 0.27 | 1.00 | 1.00 | 0.93 | 0.63 | 0.56 | 0.59 | 0.39 | 0.69 | **0.91** | 0.54 | 1.00 | 1.00 | 0.96 |
| RTD | 0.58 | 0.84 | **0.96** | 0.22 | 0.86 | 0.63 | 0.21 | 0.98 | 0.85 | 0.93 | 0.82 | 0.97 | **0.99** | 0.51 | 0.97 | 0.84 | 0.47 | 1.00 | 0.95 | 0.99 |
| IMD | 0.66 | **0.98** | 0.83 | **0.27** | 0.22 | 0.35 | 0.23 | 1.00 | 1.00 | 0.89 | 0.90 | **0.99** | 0.96 | 0.53 | 0.47 | 0.70 | 0.54 | 1.00 | 1.00 | 0.97 |
| MKA ($k = 100$) | 0.45 | 0.43 | 0.45 | 0.21 | 0.73 | 0.43 | 0.30 | 0.94 | 0.43 | 0.81 | 0.66 | 0.54 | 0.67 | 0.52 | 0.85 | 0.51 | 0.56 | 0.99 | 0.57 | 0.95 |
| CKA (linear) | 0.43 | 0.42 | 0.42 | 0.24 | 0.73 | 0.66 | 0.27 | 1.00 | 1.00 | 1.00 | 0.55 | 0.51 | 0.51 | 0.58 | 0.91 | 0.86 | 0.54 | 1.00 | 1.00 | 1.00 |
| MKA ($k = 15$) | 0.44 | 0.42 | 0.44 | 0.24 | 0.73 | 0.43 | 0.31 | 0.94 | 0.43 | 0.80 | 0.63 | 0.51 | 0.62 | 0.55 | 0.84 | 0.58 | 0.57 | 0.99 | 0.54 | 0.94 |
| MKA ($k = 50$) | 0.44 | 0.42 | 0.44 | 0.23 | 0.73 | 0.43 | 0.31 | 0.94 | 0.43 | 0.81 | 0.64 | 0.52 | 0.64 | 0.53 | 0.85 | 0.52 | 0.56 | 0.99 | 0.55 | 0.95 |
| MKA ($k = 200$) | 0.46 | 0.43 | 0.46 | 0.20 | 0.73 | 0.43 | 0.31 | 0.94 | 0.44 | 0.81 | 0.67 | 0.57 | 0.69 | 0.51 | 0.85 | 0.51 | 0.56 | 0.99 | 0.60 | 0.95 |
| AlignedCosineSimilarity | 0.50 | 0.48 | 0.43 | 0.24 | 0.84 | 0.42 | 0.29 | 0.98 | 0.70 | 0.93 | 0.71 | 0.66 | 0.60 | 0.60 | 0.94 | 0.50 | 0.58 | 1.00 | 0.67 | 0.97 |
| ConcentricityDifference | 0.29 | 0.39 | 0.39 | 0.20 | 0.37 | 0.57 | 0.21 | 0.96 | 1.00 | 1.00 | 0.64 | 0.78 | 0.74 | 0.46 | 0.72 | 0.85 | 0.48 | 0.99 | 1.00 | 1.00 |
| DistanceCorrelation | 0.43 | 0.44 | 0.42 | 0.25 | 0.86 | 0.43 | 0.22 | 1.00 | 1.00 | 1.00 | 0.64 | 0.63 | 0.52 | 0.66 | 0.95 | 0.56 | 0.52 | 1.00 | 1.00 | 1.00 |
| EigenspaceOverlapScore | 0.48 | 0.26 | 0.28 | 0.26 | 0.41 | 0.42 | 0.22 | 0.25 | 0.43 | 0.34 | 0.72 | 0.55 | 0.66 | 0.59 | 0.50 | 0.52 | 0.58 | 0.50 | 0.54 | 0.51 |
| Gulp | 0.48 | 0.26 | 0.28 | 0.24 | 0.20 | 0.42 | 0.23 | 0.29 | 0.43 | 0.30 | 0.74 | 0.56 | 0.66 | 0.56 | 0.50 | 0.51 | 0.57 | 0.51 | 0.55 | 0.56 |
| HardCorrelationMatch | 0.42 | 0.42 | 0.42 | 0.22 | 0.77 | 0.46 | 0.33 | 0.83 | 0.83 | 0.54 | 0.53 | 0.51 | 0.51 | 0.54 | 0.94 | 0.68 | 0.67 | 0.97 | 0.97 | 0.77 |
| JaccardSimilarity | 0.42 | 0.42 | 0.43 | 0.25 | 0.56 | 0.43 | 0.29 | 0.83 | 0.43 | 0.78 | 0.52 | 0.51 | 0.56 | 0.57 | 0.77 | 0.58 | 0.57 | 0.97 | 0.53 | 0.93 |
| LinearRegression | 0.49 | 0.43 | 0.46 | 0.24 | 0.22 | 0.45 | 0.23 | 0.45 | 0.68 | 0.47 | 0.72 | 0.58 | 0.69 | 0.54 | 0.48 | 0.66 | 0.52 | 0.64 | 0.81 | 0.63 |
| MagnitudeDifference | 0.24 | 0.24 | 0.37 | 0.27 | 0.66 | 0.72 | 0.18 | 0.55 | 0.49 | 0.34 | 0.56 | 0.60 | 0.74 | 0.61 | 0.89 | 0.93 | 0.47 | 0.83 | 0.86 | 0.75 |
| OrthogonalAngularShapeMetricCentered | 0.43 | 0.42 | 0.42 | 0.23 | 0.88 | 0.43 | 0.27 | 0.83 | 0.73 | 0.77 | 0.54 | 0.51 | 0.52 | 0.60 | 0.95 | 0.60 | 0.53 | 0.97 | 0.84 | 0.88 |
| OrthogonalProcrustesCenteredAndNormalized | 0.43 | 0.42 | 0.42 | 0.23 | 0.88 | 0.43 | 0.27 | 0.83 | 0.73 | 0.77 | 0.54 | 0.51 | 0.52 | 0.60 | 0.95 | 0.60 | 0.53 | 0.97 | 0.84 | 0.88 |
| PWCCA | 0.45 | 0.33 | 0.28 | 0.26 | —— | 0.44 | 0.24 | 0.24 | 0.44 | 0.36 | 0.67 | 0.52 | 0.64 | 0.58 | 1.00 | 0.64 | 0.53 | 0.44 | 0.61 | 0.56 |
| PermutationProcrustes | 0.45 | 0.39 | 0.44 | 0.27 | 0.77 | 0.90 | 0.19 | 0.68 | 0.72 | 0.93 | 0.66 | 0.71 | 0.61 | 0.52 | 0.94 | 0.97 | 0.50 | 0.84 | 0.88 | 0.98 |
| ProcrustesSizeAndShapeDistance | 0.46 | 0.45 | 0.43 | 0.27 | 0.79 | 0.62 | 0.19 | 0.92 | 0.98 | 1.00 | 0.70 | 0.68 | 0.60 | 0.51 | 0.93 | 0.88 | 0.51 | 0.98 | 1.00 | 1.00 |
| RSA | 0.47 | 0.44 | 0.43 | 0.23 | 0.74 | 0.42 | 0.33 | 0.96 | 0.43 | 0.49 | 0.71 | 0.61 | 0.54 | 0.57 | 0.89 | 0.52 | 0.64 | 0.99 | 0.58 | 0.63 |
| RSMNormDifference | 0.53 | 0.53 | 0.78 | 0.29 | 0.71 | 0.92 | 0.19 | 1.00 | 1.00 | 1.00 | 0.78 | 0.83 | 0.86 | 0.62 | 0.91 | 0.97 | 0.51 | 1.00 | 1.00 | 1.00 |
| RankSimilarity | 0.45 | 0.42 | 0.50 | 0.20 | 0.73 | 0.43 | 0.33 | 0.85 | 0.55 | 0.78 | 0.64 | 0.53 | 0.66 | 0.49 | 0.66 | 0.53 | 0.58 | 0.97 | 0.63 | 0.93 |
| SecondOrderCosineSimilarity | 0.56 | 0.73 | 0.73 | 0.22 | 0.61 | 0.42 | 0.37 | 0.99 | 0.96 | 0.95 | 0.80 | 0.87 | 0.86 | 0.52 | 0.82 | 0.50 | 0.66 | 1.00 | 0.99 | 0.99 |
| SoftCorrelationMatch | 0.43 | 0.42 | 0.42 | 0.23 | 0.60 | 0.45 | 0.33 | 0.83 | 0.82 | 0.55 | 0.53 | 0.51 | 0.50 | 0.55 | 0.85 | 0.67 | 0.65 | 0.97 | 0.96 | 0.71 |
| UniformityDifference | 0.29 | 0.50 | 0.32 | 0.24 | 0.53 | 0.90 | 0.40 | 0.53 | 0.54 | 0.33 | 0.65 | 0.81 | 0.67 | 0.58 | 0.84 | 0.96 | 0.75 | 0.78 | 0.81 | 0.66 |

Table 15: Results of Test 4 (Shortcut Affinity) for the graph domain

| Evaluation Dataset Architecture | AUPRC | | | | | | | | | | Conformity Rate | | | | | | | | | |
|---|---|---|---|---|---|---|---|---|---|---|---|---|---|---|---|---|---|---|---|---|
| | Cora | | | | Flickr | | | OGBN-Arxiv | | | Cora | | | | Flickr | | | OGBN-Arxiv | | |
| | GCN | SAGE | GAT | PGNN | GCN | SAGE | GAT | GCN | SAGE | GAT | GCN | SAGE | GAT | PGNN | GCN | SAGE | GAT | GCN | SAGE | GAT |
| CKA | 0.67 | 0.82 | 0.80 | 0.42 | 0.41 | **1.00** | 0.37 | **1.00** | **1.00** | **1.00** | 0.85 | 0.92 | 0.95 | 0.62 | 0.78 | **1.00** | 0.75 | **1.00** | **1.00** | **1.00** |
| CKA ($\delta = 0.45$) | 0.73 | 0.81 | 0.82 | 0.42 | 0.75 | **1.00** | 0.53 | **1.00** | **1.00** | **1.00** | 0.91 | 0.92 | 0.96 | 0.55 | 0.94 | **1.00** | 0.80 | **1.00** | **1.00** | **1.00** |
| CKA ($\delta = 0.2$) | **0.79** | 0.85 | 0.78 | 0.42 | 0.97 | **1.00** | 0.29 | **1.00** | **1.00** | **1.00** | **0.95** | 0.96 | 0.93 | 0.55 | 0.99 | **1.00** | 0.60 | **1.00** | **1.00** | **1.00** |
| kCKA ($k = 100$) | 0.77 | 0.86 | **0.84** | 0.42 | **1.00** | **1.00** | 0.55 | **1.00** | **1.00** | **1.00** | 0.94 | 0.95 | **0.97** | 0.56 | **1.00** | **1.00** | **0.84** | **1.00** | **1.00** | **1.00** |
| SVCCA | 0.23 | 0.36 | 0.46 | 0.24 | 0.24 | 0.93 | 0.32 | **1.00** | 0.97 | 0.83 | 0.46 | 0.60 | 0.69 | 0.53 | 0.57 | 0.97 | 0.66 | **1.00** | 0.99 | 0.91 |
| RTD | 0.72 | **0.89** | 0.73 | 0.34 | 0.78 | **1.00** | 0.31 | **1.00** | **1.00** | **1.00** | 0.88 | 0.96 | 0.93 | **0.65** | 0.91 | **1.00** | 0.63 | **1.00** | **1.00** | **1.00** |
| IMD | 0.69 | 0.80 | 0.60 | 0.23 | 0.75 | 0.97 | 0.36 | 0.61 | 0.93 | 0.92 | 0.92 | 0.96 | 0.89 | **0.65** | 0.94 | 0.99 | 0.62 | 0.84 | 0.98 | 0.98 |
| MKA ($k = 100$) | 0.76 | 0.87 | 0.83 | **0.43** | **1.00** | 0.99 | 0.54 | **1.00** | **1.00** | **1.00** | 0.92 | **0.97** | 0.96 | 0.58 | **1.00** | **1.00** | 0.83 | **1.00** | **1.00** | **1.00** |
| CKA (linear) | 0.61 | 0.78 | 0.78 | 0.34 | 0.28 | 1.00 | 0.33 | 1.00 | 0.98 | 1.00 | 0.75 | 0.90 | 0.94 | 0.63 | 0.57 | 1.00 | 0.65 | 1.00 | 1.00 | 1.00 |
| MKA ($k = 15$) | 0.75 | 0.83 | 0.82 | 0.42 | 1.00 | 0.86 | 0.55 | 1.00 | 1.00 | 1.00 | 0.91 | 0.95 | 0.96 | 0.58 | 1.00 | 0.96 | 0.84 | 1.00 | 1.00 | 1.00 |
| MKA ($k = 50$) | 0.76 | 0.87 | 0.83 | 0.43 | 1.00 | 0.95 | 0.54 | 1.00 | 1.00 | 1.00 | 0.92 | 0.97 | 0.96 | 0.59 | 1.00 | 0.99 | 0.84 | 1.00 | 1.00 | 1.00 |
| MKA ($k = 200$) | 0.76 | 0.88 | 0.83 | 0.43 | 1.00 | 1.00 | 0.54 | 1.00 | 1.00 | 1.00 | 0.92 | 0.97 | 0.96 | 0.58 | 1.00 | 1.00 | 0.83 | 1.00 | 1.00 | 1.00 |
| AlignedCosineSimilarity | 0.64 | 0.60 | 0.59 | 0.41 | 0.89 | 1.00 | 0.48 | 1.00 | 1.00 | 1.00 | 0.84 | 0.88 | 0.89 | 0.63 | 0.98 | 1.00 | 0.77 | 1.00 | 1.00 | 1.00 |
| ConcentricityDifference | 0.51 | 0.17 | 0.20 | 0.19 | 0.18 | 0.18 | 0.32 | 0.81 | 0.96 | 1.00 | 0.77 | 0.43 | 0.54 | 0.45 | 0.50 | 0.46 | 0.61 | 0.96 | 0.99 | 1.00 |
| DistanceCorrelation | 0.66 | 0.82 | 0.80 | 0.42 | 0.33 | 1.00 | 0.32 | 1.00 | 0.99 | 1.00 | 0.83 | 0.92 | 0.95 | 0.62 | 0.69 | 1.00 | 0.72 | 1.00 | 1.00 | 1.00 |
| EigenspaceOverlapScore | 0.30 | 0.50 | 0.20 | 0.37 | 0.46 | 0.43 | 0.54 | 1.00 | 0.72 | 0.97 | 0.71 | 0.74 | 0.46 | 0.50 | 0.87 | 0.60 | 0.81 | 1.00 | 0.83 | 0.99 |
| Gulp | 0.29 | 0.50 | 0.20 | 0.37 | 0.23 | 0.43 | 0.45 | 0.48 | 0.72 | 0.96 | 0.67 | 0.74 | 0.46 | 0.50 | 0.45 | 0.60 | 0.81 | 0.88 | 0.83 | 0.99 |
| HardCorrelationMatch | 0.28 | 0.35 | 0.29 | 0.35 | 0.55 | 1.00 | 0.52 | 0.80 | 0.72 | 0.83 | 0.65 | 0.77 | 0.70 | 0.69 | 0.83 | 1.00 | 0.81 | 0.96 | 0.83 | 0.97 |
| JaccardSimilarity | 0.73 | 0.78 | 0.87 | 0.42 | 1.00 | 0.83 | 0.54 | 1.00 | 1.00 | 1.00 | 0.91 | 0.93 | 0.97 | 0.53 | 1.00 | 0.96 | 0.84 | 1.00 | 1.00 | 1.00 |
| LinearRegression | 0.70 | 0.74 | 0.50 | 0.35 | 0.23 | 0.61 | 0.36 | 1.00 | 1.00 | 1.00 | 0.88 | 0.95 | 0.75 | 0.63 | 0.56 | 0.81 | 0.72 | 1.00 | 1.00 | 1.00 |
| MagnitudeDifference | 0.37 | 0.15 | 0.18 | 0.22 | 0.27 | 0.78 | 0.20 | 0.55 | 0.53 | 1.00 | 0.74 | 0.41 | 0.49 | 0.58 | 0.67 | 0.89 | 0.48 | 0.82 | 0.82 | 1.00 |
| OrthogonalAngularShapeMetricCentered | 0.63 | 0.79 | 0.62 | 0.35 | 0.58 | 1.00 | 0.31 | 1.00 | 1.00 | 1.00 | 0.79 | 0.91 | 0.90 | 0.63 | 0.80 | 1.00 | 0.72 | 1.00 | 1.00 | 1.00 |
| OrthogonalProcrustesCenteredAndNormalized | 0.63 | 0.79 | 0.62 | 0.35 | 0.58 | 1.00 | 0.31 | 1.00 | 1.00 | 1.00 | 0.79 | 0.91 | 0.90 | 0.63 | 0.80 | 1.00 | 0.72 | 1.00 | 1.00 | 1.00 |
| PWCCA | 0.33 | 0.47 | 0.22 | 0.37 | —— | 0.43 | 0.33 | 0.99 | 0.72 | 1.00 | 0.73 | 0.71 | 0.56 | 0.51 | 1.00 | 0.60 | 0.73 | 1.00 | 0.83 | 1.00 |
| PermutationProcrustes | 0.23 | 0.27 | 0.25 | 0.25 | 0.42 | 1.00 | 0.27 | 0.65 | 0.43 | 0.77 | 0.57 | 0.67 | 0.64 | 0.56 | 0.72 | 1.00 | 0.58 | 0.86 | 0.60 | 0.91 |
| ProcrustesSizeAndShapeDistance | 0.60 | 0.81 | 0.68 | 0.25 | 0.60 | 1.00 | 0.33 | 1.00 | 1.00 | 1.00 | 0.77 | 0.91 | 0.92 | 0.58 | 0.82 | 1.00 | 0.61 | 1.00 | 1.00 | 1.00 |
| RSA | 0.49 | 0.43 | 0.73 | 0.42 | 0.89 | 1.00 | 0.52 | 0.97 | 0.98 | 0.90 | 0.72 | 0.78 | 0.91 | 0.70 | 0.97 | 1.00 | 0.81 | 0.99 | 0.99 | 0.98 |
| RSMNormDifference | 0.46 | 0.50 | 0.69 | 0.37 | 0.48 | 0.92 | 0.36 | 1.00 | 1.00 | 1.00 | 0.70 | 0.76 | 0.91 | 0.64 | 0.75 | 0.97 | 0.59 | 1.00 | 1.00 | 1.00 |
| RankSimilarity | 0.72 | 0.64 | 0.85 | 0.42 | 1.00 | 0.77 | 0.54 | 1.00 | 1.00 | 1.00 | 0.86 | 0.89 | 0.96 | 0.56 | 1.00 | 0.92 | 0.83 | 1.00 | 1.00 | 1.00 |
| SecondOrderCosineSimilarity | 0.85 | 0.91 | 0.81 | 0.33 | 1.00 | 1.00 | 0.52 | 1.00 | 1.00 | 1.00 | 0.95 | 0.97 | 0.95 | 0.68 | 1.00 | 1.00 | 0.82 | 1.00 | 1.00 | 1.00 |
| SoftCorrelationMatch | 0.31 | 0.40 | 0.30 | 0.39 | 0.66 | 1.00 | 0.57 | 0.82 | 0.72 | 0.83 | 0.68 | 0.79 | 0.71 | 0.67 | 0.87 | 1.00 | 0.82 | 0.96 | 0.83 | 0.97 |
| UniformityDifference | 0.30 | 0.64 | 0.71 | 0.30 | 0.59 | 0.50 | 0.48 | 0.99 | 0.72 | 0.97 | 0.62 | 0.91 | 0.91 | 0.66 | 0.83 | 0.77 | 0.67 | 1.00 | 0.86 | 0.99 |

Table 16: Results of Test 5 (Augmentation) for the graph domain

| Evaluation Dataset Architecture | AUPRC Cora GCN | SAGE | GAT | Flickr GCN | SAGE | GAT | OGBN-Arxiv GCN | SAGE | Conformity Rate Cora GCN | SAGE | GAT | Flickr GCN | SAGE | GAT | OGBN-Arxiv GCN | SAGE |
|---|---|---|---|---|---|---|---|---|---|---|---|---|---|---|---|---|
| CKA | 0.13 | 0.76 | 0.91 | 0.64 | 0.80 | 0.57 | **1.00** | **1.00** | 0.37 | 0.92 | 0.97 | 0.85 | 0.94 | 0.77 | **1.00** | **1.00** |
| CKA ($\delta = 0.45$) | 0.13 | 0.80 | **0.98** | 0.69 | 0.94 | 0.58 | **1.00** | **1.00** | 0.38 | 0.93 | **0.99** | 0.86 | 0.99 | 0.80 | **1.00** | **1.00** |
| CKA ($\delta = 0.2$) | 0.13 | **0.93** | 0.93 | **0.75** | 0.42 | 0.55 | 1.00 | 0.99 | 0.38 | **0.99** | 0.97 | 0.90 | 0.52 | **0.84** | **1.00** | **1.00** |
| kCKA ($k = 100$) | 0.13 | 0.76 | 0.96 | 0.74 | 0.82 | 0.60 | **1.00** | 0.96 | 0.37 | 0.92 | **0.99** | 0.89 | 0.94 | 0.83 | **1.00** | 0.99 |
| SVCCA | **0.14** | 0.29 | 0.34 | 0.64 | 0.67 | 0.49 | 0.89 | 0.81 | 0.39 | 0.55 | 0.58 | 0.84 | 0.86 | 0.68 | 0.89 | 0.85 |
| RTD | **0.14** | 0.81 | 0.95 | 0.72 | **1.00** | 0.54 | **1.00** | **1.00** | **0.42** | 0.95 | **0.99** | **0.91** | **1.00** | 0.75 | **1.00** | **1.00** |
| IMD | **0.14** | 0.36 | 0.21 | 0.59 | 0.59 | 0.28 | **1.00** | **1.00** | **0.42** | 0.65 | 0.48 | 0.87 | 0.87 | 0.53 | **1.00** | **1.00** |
| UMAP ($k = 100$) | 0.13 | 0.84 | 0.93 | **0.75** | 0.93 | **0.59** | **1.00** | **1.00** | 0.35 | 0.95 | 0.98 | 0.89 | 0.98 | **0.84** | **1.00** | **1.00** |
| CKA (linear) | 0.12 | 0.73 | 0.91 | 0.57 | 0.75 | 0.51 | 1.00 | 1.00 | 0.33 | 0.92 | 0.97 | 0.83 | 0.91 | 0.74 | 1.00 | 1.00 |
| MKA ($k = 15$) | 0.13 | 0.82 | 0.92 | 0.75 | 0.93 | 0.57 | 1.00 | 1.00 | 0.35 | 0.94 | 0.98 | 0.89 | 0.97 | 0.84 | 1.00 | 1.00 |
| MKA ($k = 50$) | 0.13 | 0.83 | 0.93 | 0.75 | 0.93 | 0.59 | 1.00 | 1.00 | 0.36 | 0.95 | 0.98 | 0.89 | 0.97 | 0.84 | 1.00 | 1.00 |
| MKA ($k = 200$) | 0.13 | 0.84 | 0.95 | 0.74 | 0.93 | 0.60 | 1.00 | 1.00 | 0.37 | 0.95 | 0.98 | 0.89 | 0.98 | 0.84 | 1.00 | 1.00 |
| AlignedCosineSimilarity | 0.13 | 0.63 | 0.91 | 0.69 | 0.70 | 0.54 | 0.74 | 0.43 | 0.37 | 0.88 | 0.98 | 0.87 | 0.90 | 0.81 | 0.87 | 0.54 |
| ConcentricityDifference | 0.13 | 0.50 | 0.62 | 0.41 | 0.35 | 0.53 | 0.43 | 0.53 | 0.35 | 0.78 | 0.87 | 0.78 | 0.68 | 0.74 | 0.76 | 0.80 |
| DistanceCorrelation | 0.13 | 0.73 | 0.89 | 0.60 | 0.79 | 0.61 | 1.00 | 1.00 | 0.34 | 0.91 | 0.97 | 0.84 | 0.94 | 0.81 | 1.00 | 1.00 |
| EigenspaceOverlapScore | 0.13 | 0.62 | 0.34 | 0.68 | 0.53 | 0.46 | 0.82 | 0.49 | 0.36 | 0.90 | 0.75 | 0.88 | 0.80 | 0.70 | 0.96 | 0.75 |
| Gulp | 0.13 | 0.61 | 0.35 | 0.21 | 0.54 | 0.55 | 0.53 | 0.48 | 0.39 | 0.89 | 0.76 | 0.53 | 0.81 | 0.84 | 0.80 | 0.75 |
| HardCorrelationMatch | 0.13 | 0.63 | 0.51 | 0.71 | 0.72 | 0.57 | 0.47 | 0.51 | 0.35 | 0.89 | 0.81 | 0.89 | 0.92 | 0.85 | 0.59 | 0.77 |
| JaccardSimilarity | 0.13 | 0.80 | 0.95 | 0.74 | 0.88 | 0.58 | 1.00 | 0.99 | 0.36 | 0.95 | 0.98 | 0.89 | 0.97 | 0.84 | 1.00 | 1.00 |
| LinearRegression | 0.13 | 0.85 | 0.87 | 0.25 | 0.81 | 0.34 | 0.72 | 0.72 | 0.38 | 0.96 | 0.95 | 0.53 | 0.93 | 0.69 | 0.83 | 0.83 |
| MagnitudeDifference | 0.13 | 0.39 | 0.54 | 0.59 | 0.17 | 0.43 | 0.77 | 1.00 | 0.35 | 0.74 | 0.83 | 0.87 | 0.45 | 0.66 | 0.93 | 1.00 |
| OrthogonalAngularShapeMetricCentered | 0.13 | 0.78 | 0.83 | 0.63 | 0.76 | 0.54 | 0.72 | 0.72 | 0.37 | 0.93 | 0.95 | 0.85 | 0.94 | 0.77 | 0.83 | 0.83 |
| OrthogonalProcrustesCenteredAndNormalized | 0.13 | 0.78 | 0.83 | 0.63 | 0.76 | 0.54 | 0.72 | 0.72 | 0.36 | 0.93 | 0.95 | 0.85 | 0.94 | 0.77 | 0.83 | 0.83 |
| PWCCA | 0.14 | 0.65 | 0.42 | —— | 0.57 | 0.51 | —— | 0.48 | 0.43 | 0.89 | 0.83 | 1.00 | 0.83 | 0.75 | 1.00 | 0.74 |
| PermutationProcrustes | 0.13 | 0.61 | 0.42 | 0.65 | 0.69 | 0.40 | 0.88 | 0.73 | 0.35 | 0.89 | 0.51 | 0.87 | 0.90 | 0.62 | 0.97 | 0.87 |
| ProcrustesSizeAndShapeDistance | 0.13 | 0.69 | 0.75 | 0.70 | 0.81 | 0.40 | 1.00 | 1.00 | 0.35 | 0.90 | 0.90 | 0.89 | 0.95 | 0.63 | 1.00 | 1.00 |
| RSA | 0.13 | 0.57 | 0.78 | 0.68 | 0.72 | 0.59 | 0.75 | 0.49 | 0.38 | 0.84 | 0.94 | 0.86 | 0.92 | 0.84 | 0.89 | 0.78 |
| RSMNormDifference | 0.13 | 0.79 | 1.00 | 0.64 | 0.93 | 0.40 | 1.00 | 1.00 | 0.35 | 0.93 | 1.00 | 0.87 | 0.97 | 0.62 | 1.00 | 1.00 |
| RankSimilarity | 0.13 | 0.61 | 0.95 | 0.75 | 0.88 | 0.55 | 1.00 | 1.00 | 0.36 | 0.88 | 0.98 | 0.89 | 0.96 | 0.83 | 1.00 | 1.00 |
| SecondOrderCosineSimilarity | 0.13 | 0.86 | 0.97 | 0.78 | 0.92 | 0.62 | 1.00 | 0.99 | 0.38 | 0.97 | 0.99 | 0.91 | 0.98 | 0.84 | 1.00 | 1.00 |
| SoftCorrelationMatch | 0.13 | 0.70 | 0.50 | 0.73 | 0.58 | 0.55 | 0.43 | 0.46 | 0.34 | 0.92 | 0.80 | 0.87 | 0.85 | 0.84 | 0.51 | 0.71 |
| UniformityDifference | 0.13 | 0.53 | 1.00 | 0.68 | 0.24 | 0.25 | 0.76 | 0.53 | 0.38 | 0.86 | 1.00 | 0.90 | 0.52 | 0.66 | 0.88 | 0.81 |

Table 17: Results of Test 6 (Layer Monotonicity) for the graph domain

| Evaluation Dataset Architecture | Conformity Rate | | | | | | | | | | Spearman | | | | | | | | | |
| --- | --- | --- | --- | --- | --- | --- | --- | --- | --- | --- | --- | --- | --- | --- | --- | --- | --- | --- | --- | --- |
| | Cora | | | | Flickr | | | OGBN-Arxiv | | | Cora | | | | Flickr | | | OGBN-Arxiv | | |
| | GCN | SAGE | GAT | PGNN | GCN | SAGE | GAT | GCN | SAGE | GAT | GCN | SAGE | GAT | PGNN | GCN | SAGE | GAT | GCN | SAGE | GAT |
| CKA | 0.99 | **1.00** | **1.00** | 0.99 | 0.81 | **0.99** | 0.63 | 0.88 | 0.70 | 0.90 | 0.99 | **1.00** | **1.00** | 0.99 | 0.73 | **0.99** | 0.48 | 0.86 | 0.58 | 0.91 |
| CKA ($\delta = 0.45$) | **1.00** | **1.00** | **1.00** | **1.00** | 0.86 | **0.99** | 0.80 | 0.84 | 0.80 | 0.93 | **1.00** | **1.00** | **1.00** | **1.00** | 0.79 | **0.99** | 0.66 | 0.75 | 0.73 | 0.93 |
| CKA ($\delta = 0.2$) | **1.00** | **1.00** | **1.00** | **1.00** | 0.97 | **0.99** | 0.85 | 0.91 | 0.96 | **1.00** | **1.00** | **1.00** | **1.00** | **1.00** | 0.98 | **0.99** | 0.64 | 0.84 | 0.91 | **1.00** |
| kCKA ($k = 100$) | **1.00** | **1.00** | **1.00** | 0.91 | 0.95 | 0.95 | 0.96 | 0.88 | 0.98 | **1.00** | **1.00** | **1.00** | **1.00** | 0.91 | 0.90 | 0.97 | 0.95 | 0.85 | 0.99 | **1.00** |
| SVCCA | 0.41 | 0.78 | 0.61 | **1.00** | 0.75 | 0.69 | 0.57 | 0.49 | 0.41 | 0.56 | 0.63 | 0.80 | 0.81 | **1.00** | 0.53 | 0.44 | 0.47 | -0.17 | -0.10 | 0.33 |
| RTD | **1.00** | **1.00** | **1.00** | 0.90 | 0.55 | **0.99** | 0.61 | **1.00** | 0.98 | **1.00** | **1.00** | **1.00** | **1.00** | 0.96 | 0.19 | 0.98 | 0.31 | **1.00** | 0.99 | **1.00** |
| IMD | **1.00** | **1.00** | **1.00** | 0.97 | **1.00** | 0.94 | **0.97** | 0.85 | **1.00** | **1.00** | **1.00** | **1.00** | **1.00** | 0.97 | **1.00** | 0.82 | **0.97** | 0.55 | **1.00** | **1.00** |
| MKA ($k = 100$) | **1.00** | **1.00** | **1.00** | 0.94 | 0.99 | 0.94 | 0.96 | 0.96 | 0.99 | **1.00** | **1.00** | **1.00** | **1.00** | 0.95 | 0.98 | 0.95 | 0.94 | 0.94 | 0.99 | **1.00** |
| CKA (linear) | 0.98 | 1.00 | 1.00 | 1.00 | 0.79 | 0.92 | 0.59 | 0.87 | 0.64 | 0.95 | 0.98 | 1.00 | 1.00 | 1.00 | 0.60 | 0.89 | 0.39 | 0.85 | 0.46 | 0.96 |
| MKA ($k = 15$) | 1.00 | 1.00 | 1.00 | 1.00 | 0.98 | 0.92 | 0.99 | 0.96 | 0.99 | 1.00 | 1.00 | 1.00 | 1.00 | 1.00 | 0.98 | 0.93 | 0.98 | 0.94 | 0.99 | 1.00 |
| MKA ($k = 50$) | 1.00 | 1.00 | 1.00 | 0.99 | 0.98 | 0.93 | 0.97 | 0.96 | 0.99 | 1.00 | 1.00 | 1.00 | 1.00 | 0.98 | 0.98 | 0.94 | 0.95 | 0.94 | 0.99 | 1.00 |
| MKA ($k = 200$) | 1.00 | 1.00 | 1.00 | 0.93 | 0.99 | 0.94 | 0.96 | 0.96 | 1.00 | 1.00 | 1.00 | 1.00 | 1.00 | 0.94 | 0.98 | 0.95 | 0.94 | 0.94 | 1.00 | 1.00 |
| AlignedCosineSimilarity | 0.98 | 1.00 | 1.00 | 0.48 | 0.59 | 0.77 | 0.46 | 0.83 | 0.93 | 0.89 | 0.97 | 1.00 | 1.00 | 0.67 | 0.33 | 0.68 | 0.04 | 0.84 | 0.93 | 0.84 |
| ConcentricityDifference | 0.85 | 0.25 | 0.40 | 0.15 | 0.57 | 0.36 | 0.58 | 0.70 | 0.74 | 0.67 | 0.85 | 0.52 | 0.62 | 0.53 | 0.38 | -0.27 | 0.24 | 0.73 | 0.39 | 0.40 |
| DistanceCorrelation | 1.00 | 1.00 | 1.00 | 0.93 | 0.81 | 0.99 | 0.63 | 0.88 | 0.65 | 0.92 | 1.00 | 1.00 | 1.00 | 0.99 | 0.63 | 0.99 | 0.47 | 0.81 | 0.52 | 0.93 |
| EigenspaceOverlapScore | 1.00 | 1.00 | 1.00 | 0.99 | 1.00 | 1.00 | 0.92 | 1.00 | 1.00 | 1.00 | 1.00 | 1.00 | 1.00 | 0.98 | 1.00 | 1.00 | 0.92 | 1.00 | 1.00 | 1.00 |
| Gulp | 0.98 | 1.00 | 1.00 | 0.92 | 0.74 | 1.00 | 0.74 | 0.88 | 1.00 | 1.00 | 0.98 | 1.00 | 1.00 | 0.92 | 0.43 | 1.00 | 0.38 | 0.80 | 1.00 | 1.00 |
| HardCorrelationMatch | 0.83 | 0.98 | 0.54 | 0.80 | 0.81 | 0.84 | 0.68 | 0.86 | 0.72 | 1.00 | 0.91 | 0.99 | 0.76 | 0.84 | 0.63 | 0.80 | 0.21 | 0.83 | 0.62 | 1.00 |
| JaccardSimilarity | 1.00 | 1.00 | 1.00 | 1.00 | 0.94 | 0.96 | 0.97 | 0.96 | 0.98 | 1.00 | 1.00 | 1.00 | 1.00 | 1.00 | 0.95 | 0.97 | 0.96 | 0.97 | 0.99 | 1.00 |
| LinearRegression | 1.00 | 1.00 | 1.00 | 0.92 | 0.63 | 1.00 | 0.58 | 0.98 | 1.00 | 1.00 | 1.00 | 1.00 | 1.00 | 0.95 | 0.19 | 1.00 | 0.01 | 0.99 | 1.00 | 1.00 |
| MagnitudeDifference | 0.55 | 0.49 | 0.89 | 0.91 | 0.65 | 0.63 | 0.58 | 0.52 | 0.71 | 0.87 | 0.63 | 0.72 | 0.92 | 0.97 | 0.58 | 0.50 | 0.38 | 0.06 | 0.29 | 0.63 |
| OrthogonalAngularShapeMetricCentered | 1.00 | 1.00 | 1.00 | 0.91 | 0.80 | 0.99 | 0.66 | 0.95 | 0.99 | 0.99 | 1.00 | 1.00 | 1.00 | 0.97 | 0.62 | 0.98 | 0.45 | 0.97 | 0.99 | 0.99 |
| OrthogonalProcrustesCenteredAndNormalized | 1.00 | 1.00 | 1.00 | 0.91 | 0.80 | 0.99 | 0.66 | 0.95 | 0.99 | 0.99 | 1.00 | 1.00 | 1.00 | 0.97 | 0.62 | 0.98 | 0.45 | 0.97 | 0.99 | 0.99 |
| PWCCA | 1.00 | 1.00 | 1.00 | 1.00 | 1.00 | 1.00 | 0.75 | 1.00 | 1.00 | 1.00 | 1.00 | 1.00 | 1.00 | 0.98 | 1.00 | 0.37 | 1.00 | 1.00 | 1.00 | —— |
| PermutationProcrustes | 0.92 | 1.00 | 1.00 | 1.00 | 0.68 | 0.72 | 0.75 | 0.69 | 0.87 | 1.00 | 0.91 | 1.00 | 1.00 | 1.00 | 0.60 | 0.68 | 0.63 | 0.24 | 0.88 | 1.00 |
| ProcrustesSizeAndShapeDistance | 0.99 | 1.00 | 1.00 | 0.91 | 0.75 | 1.00 | 0.78 | 0.93 | 0.96 | 1.00 | 0.99 | 1.00 | 1.00 | 0.97 | 0.66 | 1.00 | 0.62 | 0.83 | 0.85 | 1.00 |
| RSA | 0.84 | 0.91 | 1.00 | 0.61 | 0.70 | 0.98 | 0.70 | 0.81 | 0.97 | 0.94 | 0.90 | 0.97 | 1.00 | 0.74 | 0.58 | 0.99 | 0.44 | 0.52 | 0.97 | 0.94 |
| RSMNormDifference | 0.99 | 1.00 | 1.00 | 0.91 | 0.66 | 0.68 | 0.92 | 0.85 | 0.93 | 1.00 | 0.99 | 1.00 | 1.00 | 0.97 | 0.54 | 0.65 | 0.81 | 0.85 | 0.93 | 1.00 |
| RankSimilarity | 1.00 | 1.00 | 1.00 | 1.00 | 0.95 | 0.95 | 0.98 | 0.99 | 0.98 | 1.00 | 1.00 | 1.00 | 1.00 | 1.00 | 0.97 | 0.97 | 0.99 | 0.99 | 0.99 | 1.00 |
| SecondOrderCosineSimilarity | 1.00 | 1.00 | 1.00 | 0.90 | 0.91 | 0.96 | 0.96 | 0.95 | 1.00 | 1.00 | 1.00 | 1.00 | 1.00 | 0.90 | 0.78 | 0.96 | 0.88 | 0.97 | 1.00 | 1.00 |
| SoftCorrelationMatch | 0.96 | 0.99 | 0.50 | 0.70 | 0.89 | 0.88 | 0.64 | 0.91 | 0.73 | 1.00 | 0.95 | 0.99 | 0.78 | 0.80 | 0.79 | 0.89 | 0.11 | 0.91 | 0.62 | 1.00 |
| UniformityDifference | 0.67 | 0.67 | 0.59 | 0.58 | 0.94 | 0.92 | 0.94 | 0.60 | 0.41 | 0.83 | 0.70 | 0.70 | 0.69 | 0.70 | 0.82 | 0.89 | 0.82 | 0.33 | -0.45 | 0.68 |

