# OpenReview forum: "Manifold Approximation leads to Robust Kernel Alignment"
_TMLR — Under review for TMLR_

### Review · Reviewer_XwjP · 2026-06-20

**Summary Of Contributions:**

The paper introduces Manifold-approximated Kernel Alignment (MKA), a representation-similarity measure that replaces CKA’s dense kernel with a directed, non-symmetric, non-Mercer k-nearest-neighbor kernel whose rows are normalized to a constant sum, intended to reflect local manifold structure. The paper also provides a closed-form simplification under the constant-row-sum condition, a Markov-transition interpretation of CKA and MKA. Empirically it reports synthetic experiments, ReSi benchmark results across vision/NLP/graph, and a ResNet-50/CIFAR-10 representation analysis.

Strengths:
- Markov-kernel interpretation
- Empirical evidence that MKA is relatively insensitive to k compared with kCKA
- Experiments are well chosen to expose failure modes of dense CKA
- ReSi benchmark gives a broader view across vision, NLP, and graph settings

Weaknesses:
- Some claims are stronger than the evidence supports, e.g., topology/manifold superiority.
- RBF kernel definition inconsistency

(See below for details)

**Audience:**

Yes

**Audience Explanation:**

Representation similarity metrics are widely used in interpretability, model comparison, transfer analysis, and neuroscience-adjacent work,  so a method that shifts CKA from dense kernels to locally normalized k-NN transition kernels is of interest.

**Broader Impact Concerns:**

No major ethical risks; the work is primarily methodological and concerns representation comparison.

**Claims And Evidence:**

No

**Claims Explanation:**

While most of the core claims are well supported, but still not all the claims are fully supported.
- Topology/manifold superiority. While the paper does show that MKA is a promising local kNN-based alignment measure whose behavior is more stable with respect to k than kCKA in the tested settings, the evidence does not fully support the stronger claim that MKA captures topology or manifold structure better in a broad sense. The empirical topology evidence is limited to low-dimensional synthetic examples.
- Stability claim. The paper defines A3 as a row-wise ℓ1​ stability condition, but Theorem 4.5 bounds only individual entries by O(logk/k), since many entries in a row can change with k, the conclusion that the metric is stable is not obvious.
- RBF kernel. The main text writes the RBF kernel with an unsquared norm, while the cited large-bandwidth equivalence between RBF CKA and linear CKA normally uses the Gaussian RBF with squared Euclidean distance.
- ReSi results are mixed. The evidence supports competitiveness, but not a "best" claim.

**Requested Changes:**

Critical:
- Soften or justify the topology claims. The paper should avoid claiming that MKA captures topology unless it adds formal assumptions or stronger topology-grounded experiments.
- Fix stability argument. The authors should make the Theorem 4.5 to A3 connection rigorous, or weaken the claim.
- Clarify RBF definition. The paper should specify whether the RBF kernel uses squared Euclidean distance and ensure that the use of the large-bandwidth CKA equivalence is mathematically consistent.
- Soften the cross-domain ReSi conclusion. The paper could add confidence intervals, paired tests, or equivalent uncertainty estimates for the rank comparisons, especially MKA versus kCKA.
- Address the row-only centering and its residual bias. Discuss row-only centering and possible bias/hubness effects in the main text

Would strengthen the work:
- Include end-to-end timing and memory in the main text (distance computation, k-NN search, bandwidth solving, sparse kernel operations)
- Strengthen ResNet interpretation with diagnostic experiments, or present it explicitly as a hypothesis rather than a conclusion.

---

### Review · Reviewer_Dp51 · 2026-07-06

**Summary Of Contributions:**

This paper proposes Manifold-approximated Kernel Alignment (MKA), a representation similarity metric that extends Centered Kernel Alignment (CKA) by incorporating local manifold geometry through a manifold-based kernel. The authors provide a mathematical formulation of MKA and evaluate it extensively on both synthetic datasets with controlled geometric properties and real-world representation learning benchmarks spanning vision, NLP, and graph domains. Through these experiments, the paper tries to demonstrate that MKA is more robust than existing alignment metrics to changes in data geometry, dimensionality, and hyperparameter choices, suggesting it provides a more reliable measure of representation similarity for downstream analysis.

**Additional Comments:**

In Section 4, the authors mention "However, unlike CKA, which performs both row- and column-wise centering, we opted for only row-wise centering. This leaves additional bias terms in the estimation, however, we empirically show in Appendix E that this slight oversight does not make MKA less meaningful."

This phrasing is quite ambiguous. Is the row-wise only centering not a purposeful design choice, rather than an oversight?

**Audience:**

Yes

**Audience Explanation:**

Yes, I think the metric can easily be applied to different representation learning problems in various domains when manifold comparison is involved. The authors have chosen a variety of applications from the ReSI benchmark spanning multiple domains already and make a case for broad applicability of their method. The stability shown by the proposed MKA is quite promising, therefore general machine learning audience would benefit from access to the paper/code.

**Claims And Evidence:**

Yes

**Claims Explanation:**

The paper presents a theoretical exposition as well as extensive comparisons against existing methods developed for centered kernel alignment (CKA). Multiple ablation studies and applications are considered for evaluating the method- I think the validation is quite thorough and the results present a convincing case for adopting their methodology. Code is also provided for reproducibility of experiments.The experiments also carefully examine different parameters controlling the representation, which is very valuable to practitioners considering using this work.

**Requested Changes:**

- I would suggest the authors rework the introduction section to explicitly mention which problems of CKA and existing methods exactly the proposed metric is solving, rather than having readers infer it from the theoretical exposition in the methods section.
My understanding is that the reason MKA works better than traditional CKA is due to this specific design change addressing conditions for maximal alignment more correctly, as mentioned in Section 4.1

    "CKA is dependent on the row sum of the kernel matrices, which in turn depends on the pairwise distances and the value of σ. MKA, on the other hand, is independent of the row sum (by design) and therefore it compares how the two representations deviate from the uniform distribution in their local transition structure, without requiring the kernels to be symmetric or positive semidefinite. In particular, if the two representations induce identical neighborhood distributions, $P= Q = U$, then $MKA(K,L) = 1$. Thus, the maximum alignment score corresponds exactly to perfect agreement of the directed neighborhood structure on the manifold."

    I think having a prelude to this in the introduction would improve the readability of the paper. In general, the positioning of the MKA approach (specific benefits it provides over existing approaches in literature) in the introduction is quite cryptic and hard to follow. Related works also reads more as a laundry list of what's relevant, but provides insufficient detail about the motivation for the proposed methodology.

- In 4.2 the authors mention "This manifold approximation method is similar to that of UMAP, but unlike UMAP, we avoid the symmetrization step to ensure it follows property (A2) and provides some computational efficiency."

   Could the authors elaborate on what they mean by computational efficiency beyond the kNN approximation? What additional computational gain does avoiding symmetrisation provide over UMAP? Please justify clearly (and point to relevant experiments)

- Can the authors provide more explanation for why kCKA is more sensitive to the number of neighbours than the proposed metric? (Section 5.1)

---

### Review · Reviewer_nigW · 2026-07-06

**Summary Of Contributions:**

This paper proposes Manifold-approximated Kernel Alignment (MKA), a representation similarity measure that combines a directed k-nearest-neighbor manifold kernel with a kernel-alignment objective. The kernel uses row-wise bandwidths chosen so that each row has constant mass, which leads to a simplified MKA expression, a Markov-kernel interpretation, and an element-wise stability result with respect to the neighborhood size k. The paper evaluates MKA on several synthetic topology-preserving settings, the ReSi benchmark across multiple domains, and layer-wise ResNet-50 representations.

**Additional Comments:**

None

**Audience:**

Yes

**Audience Explanation:**

Representation similarity is relevant to a broad TMLR audience, including researchers working on representation learning, interpretability, model comparison, and neuroscience-inspired analysis. A manifold-aware version of kernel alignment is likely to be useful to readers who currently use CKA or related metrics. Even though the methodological novelty is somewhat incremental, the Markov-kernel view, the synthetic characterization, and the broad ReSi evaluation make the findings useful to a meaningful subset of the community.

**Broader Impact Concerns:**

I do not see major broader-impact concerns. The paper proposes an analysis metric for representation similarity and does not introduce a new deployment setting, data-collection procedure, or obvious misuse pathway.

**Claims And Evidence:**

Yes

**Claims Explanation:**

The main empirical claim that MKA is less sensitive to k than kCKA/IMD is well supported by the synthetic experiments. The ReSi evaluation is broad and supports the claim that MKA is a competitive representation similarity measure, especially in vision and to some extent in graph settings. However, several claims should be stated more carefully. First, the novelty should be positioned more precisely, since the kernel is very close to a UMAP-style kNN membership kernel without symmetrization, followed by kernel alignment. Second, the paper does not compare against the closest same-family method, namely mutual-nearest-neighbor CKA. Third, Theorem 4.5 bounds element-wise kernel changes, but the paper does not propagate this result to the normalized MKA score itself. Finally, the ReSi results do not support a uniform claim that MKA outperforms CKA or contemporary methods across domains; in NLP, CKA remains clearly stronger in several tests.

**Requested Changes:**

Weakness
1. Sharpen the positioning relative to prior work. The paper should explicitly state what MKA contributes beyond combining a UMAP-style kNN kernel with a kernel-alignment objective. In my view, the key new ingredients are the constant-row-sum construction, the omission of symmetrization, and the resulting Markov-kernel interpretation/stability analysis. This should be made explicit.

2. Add the closest same-family baseline. The paper should include a direct comparison to Huh et al. (2024)’s mutual-nearest-neighbor CKA, at least on the synthetic topology-preserving experiments, and ideally also in the ReSi benchmark if feasible. This is important because it is the most natural baseline for a kNN-based extension of CKA.

3. Ablate the core design choices. The paper should evaluate MKA with the symmetrized manifold kernel (K^{(S)}), and MKA with full (HKH)-style centering, at least on the main synthetic experiments. Appendix E studies CKA on the symmetrized kernel, but this does not directly answer whether the proposed non-symmetric MKA design is necessary.

4. Calibrate the claims in the abstract/introduction. The empirical results suggest that MKA is strongest in vision, competitive with local methods on graphs, and weaker than CKA in some NLP settings. The paper should avoid broad claims that MKA is generally better than CKA and contemporary methods across domains.

Minors:

1. The robustness result should either be propagated from element-wise kernel stability to a bound on (|\mathrm{MKA}(K^{(k+1)},L)-\mathrm{MKA}(K^{(k)},L)|), or the text should explicitly limit the claim to kernel-level stability.

2. The choice of the row-sum target (1+\log_2 k) should be better justified. Testing alternatives such as a constant row sum, (\sqrt{k}), or (k^{1/3}) would help clarify how much of the observed robustness comes from this design choice.

3. The implementation details should be made more reproducible. Please add pseudocode for computing (\sigma_i), including the bisection bracket, tolerance, initialization, and tie handling for kNN. The paper should also clarify the “median distance” convention used for kCKA.

4. Please fix presentation issues: the graph-model citations in the main text appear inconsistent with the appendix, and the “UMAP (k=100)” row in Table 16 appears to be a mislabeled MKA row.